# Dynamic targeting enables domain-general inhibitory control over action and thought by the prefrontal cortex

Dace Apšvalka [1,6✉], Catarina S. Ferreira [2,6], Taylor W. Schmitz[3], James B. Rowe [1,4,5] & Michael C. Anderson [1,5✉]

Over the last two decades, inhibitory control has featured prominently in accounts of how humans and other organisms regulate their behaviour and thought. Previous work on how the brain stops actions and thoughts, however, has emphasised distinct prefrontal regions supporting these functions, suggesting domain-specific mechanisms. Here we show that stopping actions and thoughts recruits common regions in the right dorsolateral and ventrolateral prefrontal cortex to suppress diverse content, via dynamic targeting. Within each region, classifiers trained to distinguish action-stopping from action-execution also identify when people are suppressing their thoughts (and vice versa). Effective connectivity analysis reveals that both prefrontal regions contribute to action and thought stopping by targeting the motor cortex or the hippocampus, depending on the goal, to suppress their task-specific activity. These findings support the existence of a domain-general system that underlies inhibitory control and establish Dynamic Targeting as a mechanism enabling this ability.

---

[1] MRC Cognition and Brain Sciences Unit, University of Cambridge, 15 Chaucer Road, Cambridge CB2 7EF, UK. [2] School of Psychology, University of Birmingham, Birmingham B15 2TT, UK. [3] Department of Physiology and Pharmacology, University of Western Ontario, London, ON N6A 5B7, Canada. [4] Department of Clinical Neurosciences and Cambridge University Hospitals NHS Trust, University of Cambridge, Cambridge CB2 0SZ, UK. [5] Behavioural and Clinical Neuroscience Institute, University of Cambridge, Cambridge CB2 3EB, UK. [6] These authors contributed equally: Dace Apšvalka, Catarina S. Ferreira. ✉email: dace.apsvalka@mrc-cbu.cam.ac.uk; michael.anderson@mrc-cbu.cam.ac.uk

Well-being during difficult times requires the ability to stop unwelcome thoughts. This vital ability may be grounded in inhibitory control mechanisms that also stop physical actions[1–5]. According to this hypothesis, the right lateral prefrontal cortex (rLPFC) supports self-control, allowing people to regulate their thoughts and behaviours when fears, ruminations, or impulsive actions might otherwise hold sway[6–8]. This proposal rests on the concept of inhibitory control, a putative domain-general control mechanism that has attracted much interest in psychology and neuroscience over the last two decades[9–19] (for early work, see ref. [20]). Despite the widespread and enduring interest, direct evidence for the neural basis of domain-general inhibitory control is missing: no study has shown a control region that dynamically shifts its connectivity to suppress local processing in diverse cortical areas depending on the stopping goal—a fundamental capability of this putative mechanism. Stopping actions and memories, for example, requires that an inhibitory control region target disparate specialised brain areas to suppress motoric or mnemonic processing, respectively. We term this predicted capability dynamic targeting. Here, we tested the existence of dynamic targeting by asking participants to stop unwanted actions or thoughts. Using functional magnetic resonance imaging (fMRI) and pattern classification, we identified prefrontal regions that contribute to successful stopping in both domains. Critically, we then tested whether people's intentions to stop actions or thoughts were reflected in altered effective connectivity between the domain-general inhibition regions in the prefrontal cortex with memory or motor-cortical areas. By tracking the dynamic targeting of inhibitory control in the brain, we provide a window into humans' capacity for self-control over their thoughts and behaviours[21].

Our analysis builds on evidence that two regions of the rLPFC may contribute to stopping both actions and thoughts: the right ventrolateral prefrontal cortex (rVLPFC) and the right dorsolateral prefrontal cortex (rDLPFC). For example, stopping motor actions activates rVLPFC (especially in BA44/45, pars opercularis), rDLPFC, and anterior insula[10,22–26]. Disrupting rVLPFC impairs motor inhibition, whether via lesions[27], transcranial magnetic stimulation[28], intracranial simulation in humans[29] or monkeys[30], establishing its causal role in stopping. RVLPFC thus could promote top-down inhibitory control over actions, and possibly inhibitory control more broadly[3,10,31–33]. Within-subjects comparisons also have identified shared activations in rDLPFC (BA 9/46) that could support a domain-general mechanism that stops both actions and thoughts[5].

If these rLPFC regions play a causal role in how domain-general inhibitory control achieves stopping, the question arises as to how inhibition is directed at actions or thoughts. In our dynamic targeting hypothesis, this function is achieved by domain-general sources of inhibitory control in the LPFC that interact with specialised domain-specific target regions, the activity of which may require stopping. Here we tested whether any regions within the rLPFC had the dynamic targeting capacity needed to support domain-general inhibitory control.

Dynamic targeting requires that a candidate inhibitory control system exhibit five core attributes during stopping (see Fig. 1). First, stopping in diverse domains should engage the proposed source of control, with activation patterns within this region generalising over the specific demands of each stopping type. Consequently, activation patterns during any one form of stopping should contain information shared with inhibition in other domains. Second, the engagement of the proposed prefrontal source should track indices of inhibitory control in diverse domains, demonstrating its behavioural relevance. Third, stopping-related activity in the prefrontal sources should co-

occur with interrupted functioning in domain-specific target sites representing thoughts or actions. Fourth, the prefrontal source should exert top-down inhibitory coupling with these target sites, providing the causal basis of their targeted suppression. Finally, dynamic targeting requires that inhibitory coupling between prefrontal source and domain-specific target regions be selective to current goals. Note that domain-general inhibitory control does not require direct monosynaptic connections between the source(s) and target(s) of control.

These five attributes of dynamic targeting remain unproven, despite the fundamental importance of inhibitory control. Research on response inhibition and thought suppression instead has focused on how the prefrontal cortex contributes to stopping within each domain[9,34–36]. For example, research on thought suppression has revealed top-down inhibitory coupling from the anterior rDLPFC to the hippocampus, and to several cortical regions representing specific mnemonic content[8,37–41]. Moreover, suppressing thoughts down-regulates hippocampal activity, with the down-regulation linked to hippocampal GABA and forgetting of the suppressed content[8]. Top-down modulation of actions by rVLPFC suggests that premotor and primary motor cortex are target sites[42–44]. Action stopping engages local intracortical inhibition within M1 to achieve stopping[45–48], with a person's stopping efficacy related to local GABAergic inhibition[49]. Reinforcing this domain-specific focus, research has posited that control originates from different prefrontal regions in these two domains suggesting separate control abilities: whereas the right anterior DLPFC has received attention in work on thought suppression[2], the right VLPFC has been the focus in work on response inhibition[10,11], despite both regions often arising in both stopping tasks[22]. To integrate research from these separate domains, we sought to determine which of these candidate sources of domain-general inhibitory control participate in stopping both actions and thoughts and which exhibit the key attributes of dynamic targeting.

Although dynamic inhibitory targeting has not been tested, some large-scale networks flexibly shift their coupling with diverse brain regions that support task performance. Diverse tasks engage a frontoparietal network[50–53], which exhibits greater cross-task variability in coupling with other regions than other networks[51,54]. Variable connectivity may index this network's ability to reconfigure flexibly and coordinate multiple task elements in the interests of cognitive control[51]. A cingulo-opercular network, including aspects of rDLPFC and rVLPFC, also is tied to cognitive control, including conflict and attentional processing[55–61], with the prefrontal components exhibiting high connectivity variability over differing tasks[54]. However, previous analyses of these networks do not address dynamic inhibitory targeting: Dynamic targeting requires not merely that the prefrontal cortex exhibits connectivity to multiple regions, but that the connectivity includes a top-down component that suppresses target regions.

We sought to test the presence of dynamic targeting through the properties of prefrontal, motor and hippocampal networks (see Fig. 1 for an overview of our approach). We combined, within one fMRI session, a cognitive manipulation to suppress unwanted thoughts, the Think/No-Think paradigm[6,62], with motor action stopping in a stop-signal task[63,64]. This design provided the opportunity to identify co-localised activations of domain-general inhibitory control in prefrontal sources and observe their changes in effective connectivity with motor cortical and hippocampal targets. For the thought suppression task, prior to scanning, participants learned word pairs, each composed of a reminder and a paired thought (Fig. 2). During thought-stopping scanning blocks, on each trial, participants viewed one of these reminders. For each reminder, we cued participants either to

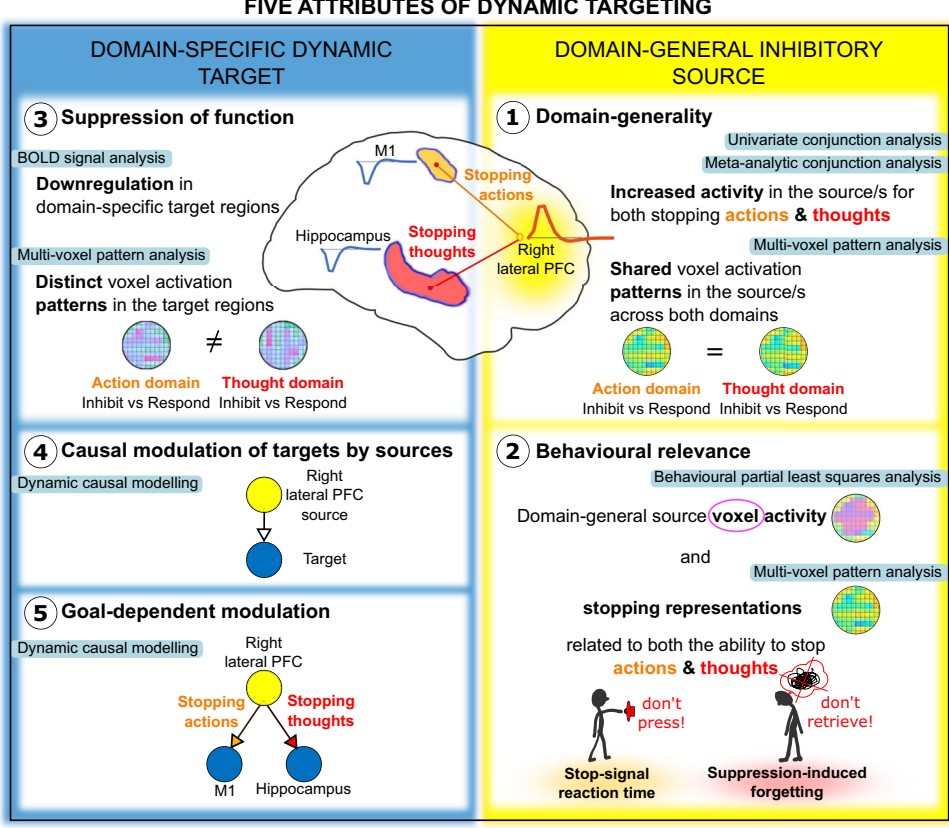

**Fig. 1 The five attributes of dynamic targeting.** Schematic of the five attributes of domain-general inhibitory control by dynamic targeting and methods employed (teal colour boxes) to test the attributes. Attributes 1–2 relate to the existence of domain-general inhibitory sources. The predicted location of such sources was in the right lateral PFC. We present the two attributes on the right side to match the visualised location of the expected sources. To test the domain-generality of inhibitory sources (attribute 1), we performed univariate and meta-analytic conjunction analysis of the No-Think > Think and Stop > Go contrasts, and cross-task multi-voxel pattern analysis (MVPA). To test the behavioural relevance (attribute 2), we related inhibitory activations within the identified domain-general regions to individual variation in inhibition ability (stop-signal reaction time and suppression-induced forgetting) using behavioural partial-least squares and MVPA. Attributes 3–5 relate to the existence of domain-specific target sites that are dynamically modulated by the domain-general sources. Our a priori assumption was that suppressing actions and thoughts would target M1 and hippocampus, respectively. To test the suppression of function within the target sites (attribute 3) we performed a region of interest (ROI) analysis expecting down-regulation within the target sites, and cross-task MPVA expecting distinct activity patterns across the two task domains. To test whether the prefrontal domain-general sources exert top-down modulation of the target sites (attribute 4) dynamically targeting M1 or the hippocampus depending on the process being stopped (attribute 5), we performed dynamic causal modelling.

retrieve their associated thought (Think trials) or instead to suppress its retrieval, stopping the thought from coming to mind (No-Think trials). For the action stopping a task, prior to scanning, participants were trained to press one of two buttons in response to differently coloured circles[8]. During the action stopping scanning blocks, participants engaged in a speeded motor response task that, on a minority of trials, required them to stop their key-press following an auditory stop signal. Action and thought-stopping blocks alternated, to enable quantification of domain-general and domain-specific activity and connectivity.

The dynamic targeting hypothesis predicts that stopping actions and thoughts call upon a common inhibition mechanism. For thought suppression, we predicted that the reminder would activate the associated thought, triggering inhibitory control to suppress hippocampal retrieval[1,65]. We predicted that this disruption would hinder later retrieval of the thought, causing suppression-induced forgetting. To verify this, we tested all pairs (both Think and No-Think pairs) after scanning, including a group of pairs that had been learned, but that was omitted during the Think/No-Think task, to estimate baseline memory performance (Baseline pairs). Suppression-induced forgetting occurs when final recall of No-Think items is lower than Baseline

items[62]. For action stopping, we proposed that the Go stimulus would rapidly initiate action preparation, with the presentation of the stop signal triggering inhibitory control to suppress motor processes in M1[63,64]. If the capacities to stop actions and thoughts are related, more efficient action stopping, as measured by stop-signal reaction time, should correlate with greater suppression-induced forgetting, at least in healthy samples.

Our primary goal was to determine whether prefrontal source regions meet the five core attributes for dynamic targeting. To test this, we first identified candidate regions that could serve as sources of control. We isolated prefrontal regions that were more active during the action and thought stopping, compared to their respective control conditions (e.g., "Go" trials, wherein participants made the cued action; or Think trials, wherein they retrieved the cued thought) and then performed a within-subjects conjunction analysis on these activations. We performed a parallel conjunction analysis on independent data from two quantitative meta-analyses of fMRI studies that used the Stop-signal or the Think/No-Think tasks, to confirm the generality of the regions identified. We next tested whether activation patterns within these potential source regions generalised over the particular stopping domains. We used multi-voxel activation patterns

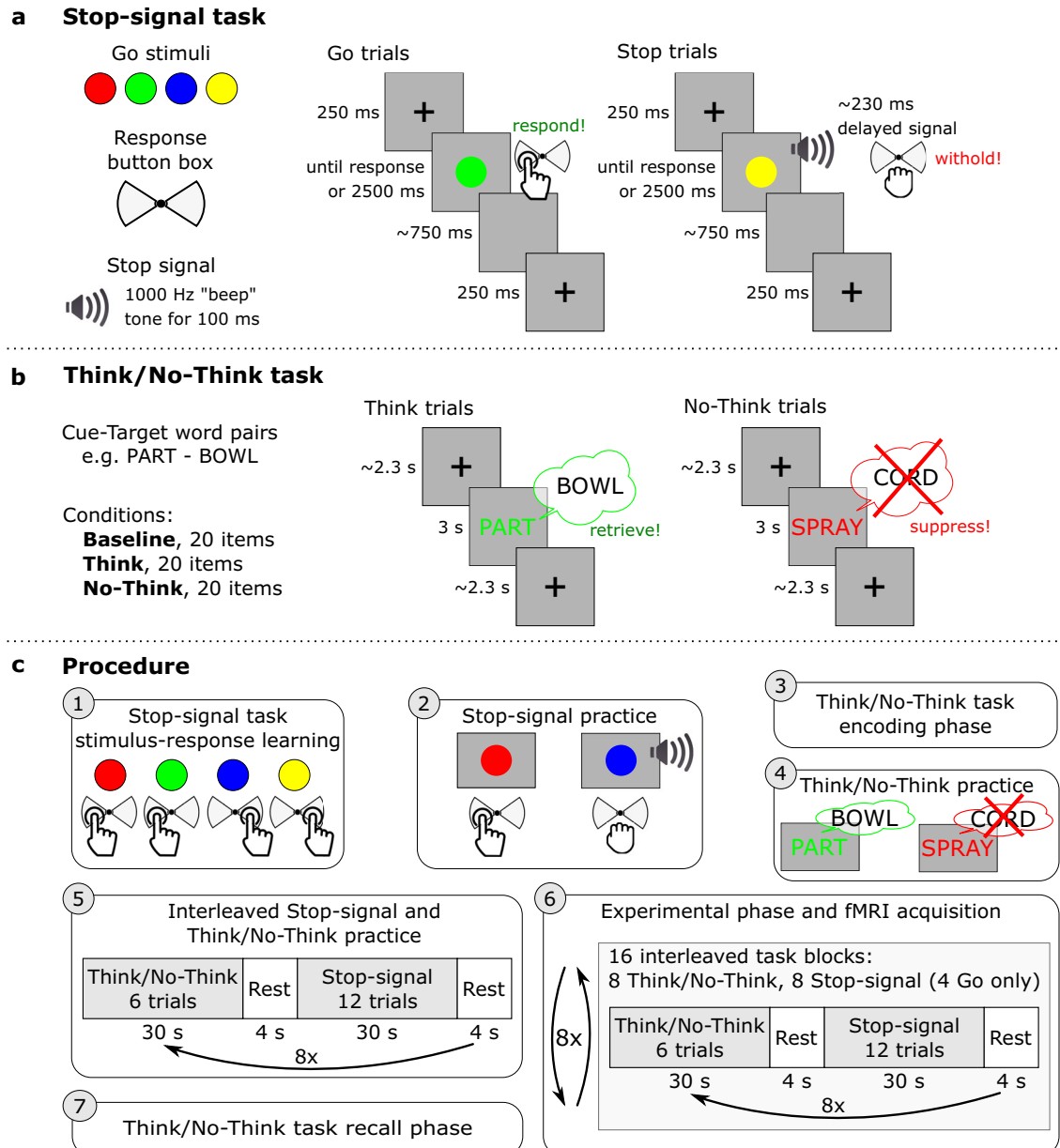

**Fig. 2 Schematic of the experimental paradigm and procedure. a** In the Stop-signal task, the Go stimuli were red, green, blue, and yellow coloured circles. On Go trials, participants responded by pressing one of the two buttons on a button box according to learned stimulus–response associations. On Stop trials, shortly after the Go stimulus, an auditory "beep" tone would signal participants to withhold the button press. The stop-signal delay varied dynamically in 50 ms steps to achieve approximately a 50% success-to-stop rate for each participant. **b** In the Think/No-Think task, participants learned 78 cue-target word pair associations. Sixty of the word pairs were then divided into three lists composed of 20 items each and allocated to the three experimental conditions: Think, No-Think, and Baseline. During Think trials, a cue word appeared in green, and participants had 3 s to retrieve and think of the associated target word. On No-Think trials, a cue word appeared in red and participants were asked to suppress the retrieval of the associated target word and push it out of awareness if it popped into their mind. **c** The procedure consisted of 7 steps: (1) stimulus–response learning for the Stop-signal task: (2) Stop-signal task practice; (3) encoding phase of the Think/No-Think task; (4) Think/No-Think practice; (5) practice of interleaved Stop-signal and Think/No-Think tasks; (6) the main experimental phase during fMRI acquisition where participants performed interleaved 30 s blocks of Stop-signal and Think/No-Think tasks; (7) recall phase of the Think/No-Think task.

to train a classifier to discriminate stopping from going in one modality (e.g., action stopping), to test whether it could identify stopping in the other modality (e.g., thought suppression). Finally, to examine behavioural relevance, we related inhibitory activations within these meta-analytic conjunction areas to individual variation in inhibition ability (e.g., suppression-induced forgetting and stop-signal reaction time) using behavioural partial least squares and multi-voxel pattern analysis. Any region surviving these constraints was considered a strong candidate for a

hub of inhibitory control. We hypothesised that these analyses would identify the right anterior DLPFC[5,6,22,37], and right VLPFC[10,24].

To verify that inhibitory control targets goal-relevant brain regions during stopping, we next confirmed that a priori target sites are suppressed in a goal-specific manner. Specifically, stopping retrieval should down-regulate hippocampal activity[1,4,37,39–41,65], more than does action stopping. In contrast, stopping actions should inhibit motor cortex more than does

**Table 1 Within-subjects and meta-analysis domain-general inhibition-induced activations (Stop > Go and No-Think > Think).**

| Hemisphere | Region | ~BA | Network | MNI of the peak | | | Volume (mm³) |
|---|---|---|---|---|---|---|---|
| | | | | x | y | z | |
| *Within-subjects, Stop > Go & No-Think > Think* | | | | | | | |
| Right | Inferior frontal gyrus (VLPFC) Insula | 44, 45 | CON, FPN | 45 | 18 | 8 | 5366 |
| Right | Inferior parietal lobule | 40 | CON, FPN, PMM | 63 | −42 | 41 | 3611 |
| Right | Supplementary motor area | 6, 8 | CON, FPN, LAN | 15 | 18 | 64 | 2498 |
| Right | Middle frontal gyrus (DLPFC) | 9, 10, 46 | CON | 33 | 42 | 23 | 1654 |
| | Superior frontal gyrus (DLPFC) | | | | | | |
| Right | Precentral gyrus | 6 | CON, FPN, LAN | 42 | 3 | 41 | 945 |
| Left | Inferior parietal lobule | 40 | CON, FPN | −60 | −48 | 41 | 641 |
| *Meta-analysis, Stop > Go & No-Think > Think* | | | | | | | |
| Right | Inferior frontal gyrus (VLPFC) Insula | 44, 45 | CON, FPN | 36 | 26 | 0 | 4523 |
| Right/Left | Supplementary motor area | 6, 8 | CON, FPN, LAN | 14 | 14 | 60 | 3071 |
| Left | Inferior frontal gyrus Insula | 44, 45 | CON, FPN | −44 | 18 | 0 | 2970 |
| Right | Inferior parietal lobule | 40 | CON, FPN, PMM | 58 | −46 | 34 | 2633 |
| Right | Anterior cingulate cortex | 24, 32 | CON, FPN | 6 | 22 | 38 | 1620 |
| Right | Middle frontal gyrus (DLPFC) | 9, 10, 46 | CON | 36 | 50 | 22 | 844 |
| | Superior frontal gyrus (DLPFC) | | | | | | |
| Right | Basal ganglia | | | 16 | 8 | 8 | 776 |
| Left | Inferior parietal lobule | 40 | CON, FPN | −60 | −50 | 34 | 608 |
| Right | Precentral gyrus | 6 | CON, LAN | 44 | 2 | 46 | 270 |
| Right | Superior parietal lobule | 7 | FPN, DAN | 34 | −48 | 46 | 176 |
| *Within-subjects & Meta-analysis, Stop > Go & No-Think > Think* | | | | | | | |
| Right | Inferior frontal gyrus (VLPFC) Insula | 44, 45 | CON, FPN | 45 | 18 | 8 | 2666 |
| Right | Inferior parietal lobule | 40 | CON, FPN, PMM | 63 | −42 | 38 | 1620 |
| Right | Supplementary motor area | 6, 8 | CON, FPN, LAN | 15 | 18 | 64 | 1418 |
| Right | Middle frontal gyrus (DLPFC) | 9, 10, 46 | CON | 33 | 39 | 26 | 338 |
| Left | Inferior parietal lobule | 40 | CON, FPN | −60 | −48 | 41 | 270 |
| Right | Precentral gyrus | 6 | CON, LAN | 42 | 3 | 41 | 135 |

thought stopping[8]. To determine whether these differences in modulation arise from inhibitory targeting by our putative domain-general prefrontal control regions, we used dynamic causal modelling[66]. If both DLPFC and VLPFC are involved, as prior work suggests, we sought to evaluate whether one or both regions are critical sources of inhibitory control.

Here, we show that stopping unwanted thoughts and actions engages common regions in the rDLPFC and rVLPFC. Critically, these regions did not merely share common activation during these forms of stopping, but also exhibited the five core attributes needed to infer dynamic targeting, shifting their connectivity to domain-specific target regions to suppress their regional activity. These findings confirm central predictions of a domain-general inhibitory control mechanism and establish the joint role of both rDLPFC and rVLPFC in achieving this function.

## Results

**Stopping actions and thoughts recruits right DLPFC and VLPFC.** We first identified brain regions that could provide a source of inhibitory control during action and thought stopping (Establishing Attribute 1: domain-generality). The whole-brain voxel-wise conjunction analysis of the Stop > Go and the No-Think > Think contrasts revealed that both motor and thought inhibition evoked conjoint activations in the right prefrontal cortex (PFC), specifically, the rDLPFC (middle frontal and superior frontal gyri), rVLPFC (ventral aspects of inferior frontal gyrus, including BA44/45, extending into insula), precentral gyrus, and supplementary motor area (see Table 1 and Fig. 3). These findings suggest a role of the right PFC in multiple stopping domains[5,10,67], necessary for dynamic targeting.

The observation that rDLPFC contributes to inhibitory control might seem surprising, given the published emphasis on the rVLPFC in motor stopping studies[10,11]. It could be that rDLPFC

activation arises from the need to alternate between the Stop-signal and Think/No-Think tasks, or from carryover effects between tasks. We, therefore, compared the activations observed in our within-subjects conjunction analysis to a meta-analytic conjunction analysis of independent Stop-signal ($N = 40$) and Think/No-Think ($N = 16$) studies (see the "Methods" section) conducted in many different laboratories with different variations on the two procedures (see ref. [22] for an earlier version with fewer studies). The meta-analytic conjunction results were highly similar to our within-subjects results, with conjoint clusters in matched regions of DLPFC, VLPFC (BA44/45, extending into insula), right anterior cingulate cortex, and right basal ganglia (see Table 1 and Fig. 3). Notably, in both the within-subjects and meta-analytic conjunctions, the domain-general activation in rDLPFC did not spread throughout the entire right middle frontal gyrus but was confined to the anterior portion of the rDLPFC, spanning BA9/46 and BA10. The convergence of these conjunction analyses suggests that the involvement of the rDLPFC, and our findings of conjoint activations across the two stopping domains more broadly, do not arise from the specific procedures of the stopping tasks or to carryover effects arising from our within-subjects design; rather, they indicate a pattern that converges across laboratories and different experimental procedures.

The domain-general stopping activations included areas outside of the prefrontal cortex (see Table 1 and Fig. 3). Although not the focus on the current investigation, we characterised these activations in relation to large-scale brain networks, using a publicly available Cole-Anticevic brain-wide network partition (CAB-NP)[68]. We used the Connectome Workbench software[69] to overlay our activations over the CAB-NP to estimate the parcel and network locations of our clusters. Domain-general clusters primarily were in the Cingulo-Opercular (CON) and Frontoparietal (FPN) networks (86% of parcels fell within these two

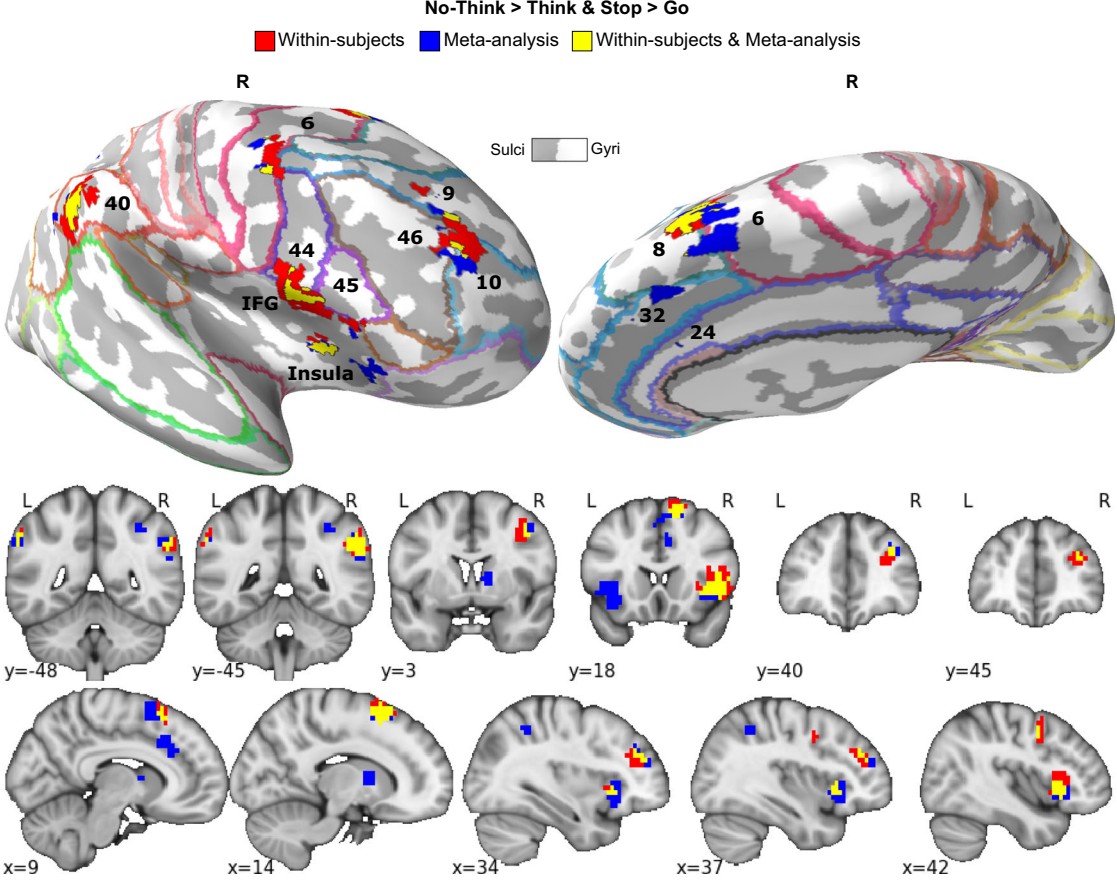

**Fig. 3 Domain-general inhibition-induced activations.** Red: within-subjects ($N = 24$) conjunction of the Stop > Go and the No-Think > Think contrasts thresholded at $p < 0.05$ FDR corrected for whole-brain multiple comparisons. Blue: meta-analytic conjunction of Stop > Go and the No-Think > Think contrasts from independent 40 Stop-signal and 16 Think/No-Think studies. Yellow: overlap of the within-subjects and meta-analytic conjunctions. Results are displayed on an inflated MNI-152 surface with outlined and numbered Brodmann areas (top panel), as well as on MNI-152 volume slices (bottom panel). R: right hemisphere; L: left hemisphere. The brain images were generated using FreeSurfer software (http://surfer.nmr.mgh.harvard.edu), and PySurfer (https://pysurfer.github.io) and Nilearn (https://nilearn.github.io) Python (Python Software Foundation, DE, USA) packages. Conjunction maps and visualisation notebook are available on the GitHub repository[104] (https://github.com/dcdace/Domain-general/).

networks in the within-subjects conjunction), but also included Posterior-Multimodal and Language networks parcels (see Supplementary Table 1 and Supplementary Fig. 1). Of the 21 cortical parcels identified for the within-subjects conjunction (see Supplementary Table 1), the majority (57%) participated in the CON, whereas 29% were involved in the FPN; the independent meta-analysis yielded similar findings (56% vs. 30%; see Supplementary Table 2 and Supplementary Fig. 2). Our main right prefrontal regions both featured parcels from the CON; however, whereas rDLPFC was located solely in the CON (in both the within-subjects and meta-analytic conjunctions), the rVLPFC region also included parcels from the FPN.

Together, these findings confirm the role of both the right anterior DLPFC and rVLPFC for both motor and memory inhibition, consistent with prior evidence of a causal role of these regions in the stopping function of inhibitory control[27–30]. Moreover, they show that stopping recruits a larger network of regions, dominated by the CON, and to a lesser degree, FPN. These findings suggest that to achieve stopping, domain-general inhibitory control may reflect a special configuration of the CON that includes elements of the FPN and other networks. Notably, key regions of the FPN were absent from all analyses (no suprathreshold activations), including the large middle frontal region often taken as a hallmark of domain-general cognitive control[51,52].

**Action and thought stopping abilities are related.** We next confirmed that action-stopping efficiency was associated with successful thought suppression. To quantify action stopping efficiency, we computed stop-signal reaction times (SSRTs) using the consensus standard integration method[64]. We confirmed that the probability of responding to Stop trials ($M = 0.49$, SD $= 0.07$; ranging from 0.36 to 0.69) fell within the recommended range for reliable estimation of SSRTs[64], and that the probabilities of Go omissions ($M = 0.002$, SD $= 0.01$) and choice errors on Go trials ($M = 0.04$, SD $= 0.02$) were low. We next verified that the correct Go RT ($M = 600.91$ ms, SD $= 54.63$ ms) exceeded the failed Stop RT ($M = 556.92$ ms, SD $= 56.77$) in all but one participant (9 ms difference between the failed Stop RT and correct Go RT; including this participant makes little difference to any analysis, so they were not excluded). Given that the integration method requirements were met, the average SSRT, our measure of interest, was 348.34 ms (SD $= 51.25$ ms), with an average SSD of 230 ms (SD $= 35.68$ ms).

We next verified that the Think/No-Think task had induced forgetting of suppressed items. We compared final recall of No-Think items to that of Baseline items that had neither been suppressed nor retrieved (see the "Methods" section). Consistent with a previous analysis of these data[8] and with prior findings[1,62,65,70], suppressing retrieval impaired No-Think recall ($M = 72\%$, SD $= 9\%$) relative to Baseline recall ($M = 77\%$,

SD = 9%), yielding a suppression-induced forgetting (SIF) effect (Baseline − No-Think = 5%, SD = 9%, one-tailed $t_{23} = 2.55$, $p = 0.009$, $d = 0.521$). Thus, suppressing retrieval yielded the predicted inhibitory aftereffects on unwanted thoughts.

To test the relationship between thought suppression and action stopping, we calculated a SIF score for each participant by subtracting No-Think from Baseline recall performance (Baseline − No-think). This indexes the efficiency with which each participant could down-regulate later accessibility of suppressed items, an aftereffect of suppression believed to be sensitive to inhibitory control[62]. We then correlated the SSRT and SIF scores (excluding one bi-variate outlier; see the "Methods" section). Consistent with a shared inhibition process, better action stopping efficiency (faster SSRTs) was associated with greater SIF ($r_{ss} = -0.492$, $p = 0.014$, see Fig. 4a; A detailed report of behavioural results is available in the supplementary analysis notebook).

Although we quantified SSRT with the integration method, this method may, at times, overestimate SSRTs because it does not consider times when participants fail to trigger the stopping process, known as trigger failures[71]. Trigger failures may arise, for example, when a participant is inattentive and misses a stop signal. We recomputed SSRTs using a method that estimates trigger failure rate and that corrects SSRTs for these events[71,72]. This method yielded shorter SSRTs ($M = 278.84$ ms, SD = 41.13 ms) than the integration method ($M = 348.34$ ms), but the relation between stopping efficiency and SIF was qualitatively similar ($r = -0.383$, $p = 0.065$). This alternate SSRT measure also did not qualitatively alter brain–behaviour relationships reported in later analyses.

**Right DLPFC and VLPFC underlie successful stopping behaviour**. We next examined whether action stopping and thought suppression depend on activity in the putative domain-general regions identified in our conjunction analysis, consistent with behavioural relevance (Attribute 2). To ensure that our region was representative of domain-general stopping activations across published studies and was based on data independent of the current experiment, we focused on the meta-analytic conjunction region (n.b. results are similar if the within-subjects conjunction is used). We tested whether activation in the very same voxels would predict SIF and SSRT. This test used behavioural PLS analysis (see the "Methods" section), excluding one behavioural bi-variate outlier from this analysis (see the "Methods" section), although the results with the outlier included did not qualitatively differ.

The first latent variable (LV1) identified by PLS accounted for 78% of the covariance between stopping activations and behavioural measures of SSRT and SIF. The correlation profile of LV1 showed a negative correlation with SSRT scores ($r = -0.432$, [−0.724, −0.030] bootstrapped 95% CI) and a positive correlation with SIF scores ($r = 0.441$, [0.044, 0.729] bootstrapped 95% CI; Fig. 4b). According to this correlation pattern, for the brain voxels with significant positive salience, a higher BOLD signal for the Inhibit > Respond contrast predicted faster SSRTs (i.e., better action stopping speed) and larger amounts of SIF (i.e., better memory inhibition). Voxels associated with such significant positive salience arose across the entire set of domain-general conjunction regions except for the inferior parietal lobules (see Table 2 and Fig. 4c). No voxels were associated with a significant negative salience (i.e., the opposite pattern). These findings support the hypothesis that the stopping-evoked activity identified in our conjunction analyses plays behaviourally important roles both in stopping actions efficiently and in forgetting unwanted thoughts, a key attribute necessary to establish dynamic targeting.

**Stopping inhibits goal-relevant domain-specific target areas**. A key attribute of dynamic targeting is that the domain-specific target areas are inhibited in response to activity of the domain-general source of inhibitory control when the specific task goals require it (Attribute 3: suppression of function in target regions). For example, when stopping a motor action is the goal, inhibitory control processes supported by domain-general source regions in the prefrontal cortex downregulate activity in M1, cancelling motor actions[73–77]; when the goal is to stop memory retrieval, however, the same prefrontal process downregulates hippocampal activity, interrupting retrieval[1,2,4,7,9,37–39,65,78,79]. Previously, we reported both of the foregoing patterns in a separate analysis of the current data[8]. In the analyses below, we reconfirmed these findings using the left M1 and the right hippocampus ROIs which we defined specifically for the current analyses (see the "Methods" section).

Dynamic targeting predicts a crossover interaction such that action stopping suppresses M1 more than it does the hippocampus, whereas thought stopping should do the reverse. A repeated-measures analysis of variance (ANOVA) confirmed a significant interaction between modulatory target regions (M1 vs. hippocampus) and stopping modality (stopping actions vs. stopping thoughts) on the BOLD signal difference between the respective inhibition and non-inhibition conditions in each modality ($F_{1,23} = 45.99$, $p < 0.001$; Fig. 5a). The main effects were significant for both the modulatory target regions ($F_{1,23} = 10.01$, $p = 0.004$) and the stopping modality ($F_{1,23} = 9.28$, $p = 0.006$). Post-hoc pairwise comparisons showed that whereas stopping motor responses (Stop–Go) evoked greater downregulation of the M1 than the hippocampus ROI ($t_{23} = 6.26$, $p < 0.001$, $d = 1.279$), suppressing thoughts (No-Think–Think) evoked larger down-regulation of the hippocampus than the M1 ROI ($t_{23} = 3.53$, $p = 0.002$, $d = 0.720$). Thus, action stopping and thought suppression preferentially modulated the left M1 and right hippocampus, respectively. Critically, these modulations were not solely produced by up-regulation in the Go or Think conditions, as illustrated by negative BOLD response during Stop ($t_{23} = -5.08$, $p < 0.001$, $d = 1.037$) and No-Think ($t_{23} = -2.23$, $p = 0.018$, $d = 0.455$) conditions (see Fig. 5b). Thus, brain regions involved in representing the type of content requiring inhibition for each stopping task showed evidence of interrupted function during stopping, consistent with the requirements of dynamic targeting.

**Domain general stopping representations in prefrontal cortex**. It is possible that despite sharing activations in the rDLPFC and rVLPFC, the pattern of activation across voxels within these regions fundamentally differs for action and thought stopping, a possibility that cannot be excluded with univariate methods. However, dynamic targeting predicts similarities in the pattern of activation observed in prefrontal regions for the two stopping domains (Attribute 1: Domain-generality). Specifically, domain-general univariate activations in rDLPFC and rVLPFC should reflect three types of processes: (a) processes that implement the domain-general stopping mechanism (domain-general stopping features); (b) processes that accept domain-specific inputs needed to drive stopping (input features); and (c) processes that effectuate stopping through their interaction with domain-specific posterior cortical or subcortical regions (output features). Input, stopping, and output features are each necessary components of a stopping mechanism with the flexibility to be triggered by multiple modalities and act on diverse processing domains. However, aspects of stopping also may be unique to each domain, yielding a fourth type of process: domain-specific stopping features. Domain-specific stopping features differ from

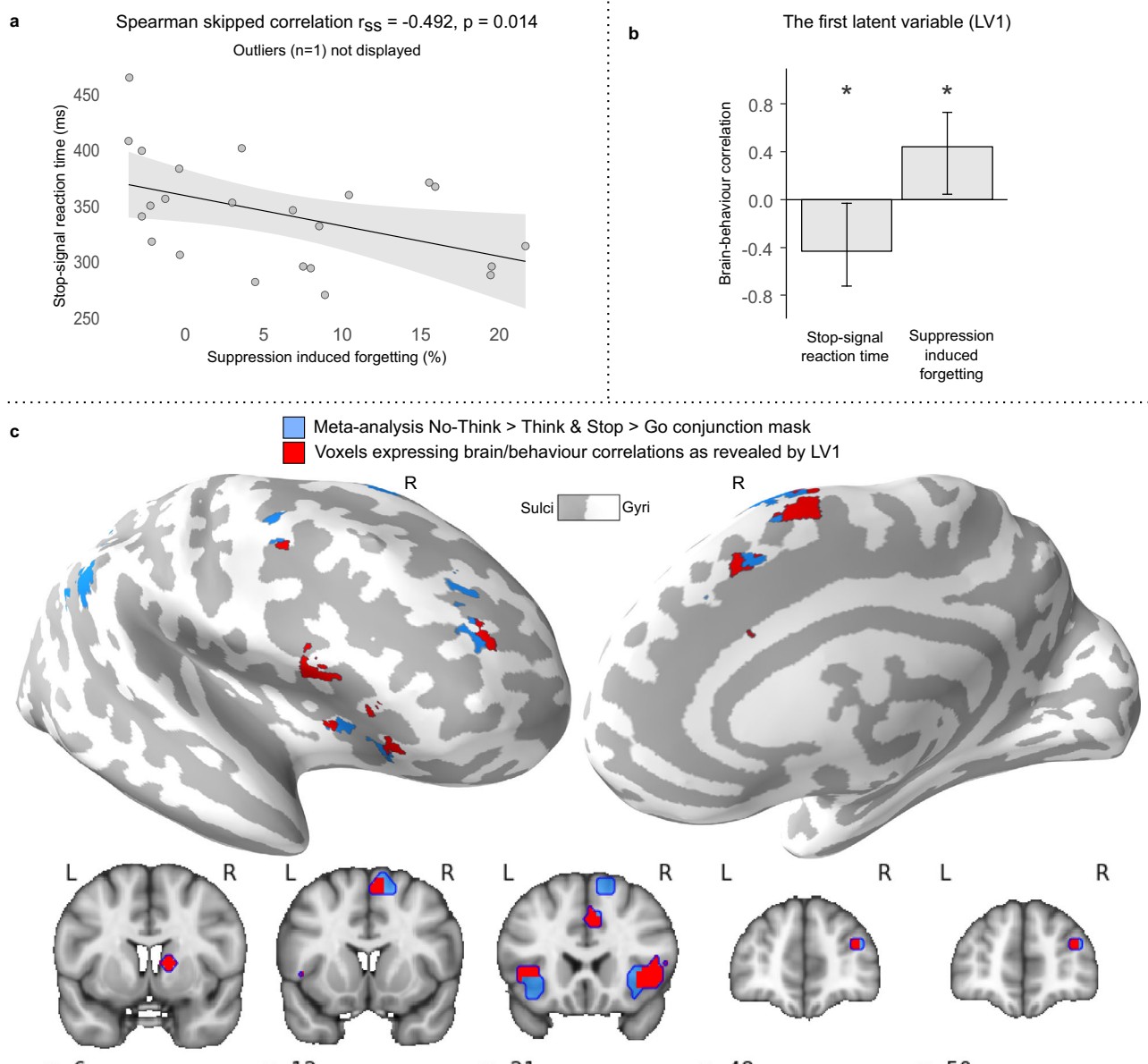

**Fig. 4 Domain-general behavioural and brain/behaviour relationships. a** and **b** Source data are provided as a Source Data file. **a** Better action stopping efficiency (shorter stop-signal reaction time) was associated with better inhibitory control over thoughts (percentage of items forgotten for No-Think relative to Baseline conditions at the final recall phase), i.e. suppression-induced forgetting; $r_{ss} = -0.492$, $p = 0.014$, $n = 24$. One bivariate outlier is not displayed on the scatterplot. Shading represents 95% CI. **b** and **c** A behavioural partial least squares (PLS) analysis was conducted to identify brain areas where individual variation in inhibition ability (SSRT and SIF) was related to increased inhibition-induced activity (main effect contrast of inhibition from the within-subject experiment, masked by the meta-analytic conjunction). **b** The first latent variable (LV1) identified voxels showing a significant pattern of brain/behaviour correlations to both SSRT and SIF (error bars indicate bootstrapped 95% CI, $n = 5000$, *$p < 0.05$). **c** The voxel salience map expressing LV1. Blue: meta-analytic conjunction mask. Red: voxels showing a significant pattern of brain/behaviour correlations as revealed by the LV1; thresholded at bootstrapped standard ratio 1.96, corresponding to $p < 0.05$, two-tailed. Results are displayed on an inflated MNI-152 surface (top panel), as well as on MNI-152 volume slices (bottom panel). R: right hemisphere; L: left hemisphere. The brain images were generated using FreeSurfer software (http://surfer.nmr.mgh.harvard.edu), and PySurfer (https://pysurfer.github.io) and Nilearn (https://nilearn.github.io) Python (Python Software Foundation, DE, USA) packages. Conjunction mask, PLS results and visualisation notebook are available on the GitHub repository[104] (https://github.com/dcdace/Domain-general/).

domain-specific output features in that the latter govern the interaction of the stopping mechanism with posterior-cortical or subcortical target regions, whereas the former reflect computations specific to a stopping domain that are local to the prefrontal source region. Domain-general stopping features should yield similarities between the multivariate patterns for action and thought stopping; in contrast, input, output, and domain-specific stopping features should yield differences in the multivariate

patterns for thought and action stopping, with the relative contributions of each being difficult to disentangle. Cross-modality decoding should not be possible in domain-specific target regions, reflecting their specialised involvement in action or memory stopping. Conversely, between-modality decoding, reflecting domain-specific features, must exist in the domain-specific target regions and to some extent in the domain-general source regions.

**Table 2 Control network regions showing a significant pattern of brain/behaviour correlations as revealed by the first latent variable of the PLS analysis.**

|  | Brain region | ~BA | MNI of the peak | | | Volume (mm$^3$) |
|---|---|---|---|---|---|---|
|  |  |  | x | y | z |  |
| Right | Inferior frontal gyrus (VLPFC) Insula | 44, 45 | 45 | 21 | 0 | 3375 |
| Right | Anterior cingulate cortex | 24, 32 | 6 | 30 | 34 | 1418 |
| Left | Inferior frontal gyrus Insula | 44, 45 | −33 | 21 | 4 | 1046 |
| Right/Left | Supplementary motor area | 6, 8 | 6 | 9 | 64 | 1013 |
| Right | Basal ganglia |  | 15 | 3 | 8 | 709 |
| Right | Middle frontal gyrus (DLPFC) | 10, 46 | 33 | 48 | 19 | 304 |
| Right | Precentral gyrus | 6 | 42 | 3 | 41 | 68 |

To identify the predicted cross-modality similarities, within each subject, we trained a classifier to distinguish Inhibit and Respond conditions in one modality and tested the ability to distinguish Inhibit and Respond conditions in the other modality. We performed the classification analysis on the rDLPFC, rVLPFC, right hippocampus, and left M1 ROIs (see the "Methods" section). The analysis revealed that a classifier trained on one modality could discriminate Inhibition from Respond conditions in the other modality significantly above chance (50%) for both rDLPFC ($M = 58\%$, SD $= 9\%$, one-tailed $t_{23} = 4.17$, $p_{adj} = 0.001$, $d = 0.852$) and rVLPFC ($M = 60\%$, SD $= 11\%$, one-tailed $t_{23} = 4.46$, $p_{adj} < 0.001$, $d = 0.911$). This cross-modality decoding suggests that a domain-general inhibitory control process contributes to activity in these regions (see Fig. 5c; cross-modality prediction accuracy is even stronger in early task blocks—see next section). To identify between-task differences (the domain-specific features), we trained a classifier to discriminate Stop from No-Think trials (see the "Methods" section). The classifier could indeed discriminate action and thought stopping in both rDLPFC ($M = 70\%$, SD $= 13\%$, one-tailed $t_{23} = 7.37$, $p_{adj} < 0.001$, $d = 1.504$) and rVLPFC ($M = 82\%$, SD $= 11\%$, $t_{23} = 13.89$, $p_{adj} < 0.001$, $d = 2.835$). The superior discrimination of action and thought stopping in the rVLPFC compared to rDLPFC was reliable; however, control analyses matching ROI size eliminated this advantage, suggesting that it was an artefact of the larger ROI used for rVLPFC (see Supplementary Fig. 4). It is unclear whether this domain-specific component in the LPFC reflects evidence for the input and output features required by the dynamic targeting hypothesis or instead domain-specific stopping processes, either of which may be exploited by a classifier to enhance between-modality prediction performance. The cross-modality prediction findings, however, clearly confirm predictions of the domain-generality attribute of dynamic targeting.

In contrast to the patterns observed in the prefrontal cortex, we observed no evidence of cross-modality decoding in the modality-specific regions targeted by inhibitory control. This pattern arose for both right hippocampus ($M = 49\%$, SD $= 10\%$, one-tailed $t_{23} = -0.37$, $p_{adj} = 1$, d $= 0.075$) and also left M1 ($M = 48\%$, SD $= 10$, one-tailed $t_{23} = -1.16$, $p_{adj} = 1$, $d = 0.236$), in which the cross-modality classifier accuracy did not significantly differ from chance performance (see Fig. 5c). An estimated one-sample $t$-test Bayes factor (one-tailed; medium prior Cauchy scale 0.707; null/ alternative) suggested that the data were substantially in favour of the null hypothesis for both the hippocampus ($B_{01} = 6.01$, posterior distribution: Median $= 0.109$, 95% CI $= [0.004, 0.385]$) and M1 ($B_{01} = 9.14$, posterior distribution: Median $= 0.071$, 95% CI $= [0.003, 0.294]$). Nevertheless, these putative target regions responded very differently to the two modalities of inhibitory control, as evidenced by presence of significant domain-specific information in each region (Attribute 3: suppression of function).

A classifier could reliably distinguish No-Think trials from Stop trials within both the right hippocampus ($M = 63\%$, SD $= 11\%$, $t_{23} = 5.89$, $p_{adj} < 0.001$, $d = 1.202$) and left M1 ($M = 65\%$, SD $= 12\%$, $t_{23} = 6.56$, $p_{adj} < 0.001$, $d = 1.338$; see Fig. 5c). Again, these differences may reflect input features (either from perception or top-down control), output features, or the impact of the domain-specific inhibition processes on the target regions. Notably, although comparisons of classification accuracies across ROIs should be interpreted with caution[80], the ability of the classifier to distinguish No-Think from Stop trials did not vary across our four ROIs (rDLPFC, rVLPFC, hippocampus, M1) when ROI size was controlled (see Supplementary Fig. 4). Thus, all ROIs supported comparable classification performance in domain-specific classification, making it unlikely that the null classification results in the between-domain classifier in the hippocampus and M1 simply reflect poor signal quality in those target regions.

Because we z-normalised activation within each of these regions within each task, the ability to distinguish No-Think from Stop trials was not based on differences in overall univariate signal, but instead on information contained in distinct patterns of activity in each task. These findings reinforce the assumption that the hippocampus and M1 are uniquely affected by thought and action stopping respectively, as expected for domain-specific targets of inhibitory control. Taken together, these contrasting findings from the PFC and domain-specific regions are compatible with the view that rDLPFC and rVLPFC jointly contribute to a domain-general stopping process that dynamically targets different regions, depending on the nature of the content to be suppressed.

**Action stopping representations predict adaptive forgetting.** Because dynamic targeting posits that LPFC contains domain-general stopping representations, training a classifier to distinguish stopping in one domain should predict stopping behaviour in other domains. For example, the ability of an action stopping classifier to distinguish when people are suppressing thoughts raises the intriguing possibility that it also may identify participants who successfully forget those thoughts (establishing further evidence of Attribute 2, behavioural relevance). To test this possibility, we capitalised on an adaptive forgetting phenomenon known as the conflict reduction benefit (for a review, see[6]). The conflict-reduction benefit refers to the declining need to expend inhibitory control resources that arises when people repeatedly suppress the same intrusive thoughts. This benefit arises because inhibitory control induces forgetting of inhibited items, which thereafter cause fewer control problems. For example, over repeated inhibition trials, activation in rDPLFC, rVLPFC, and anterior cingulate cortex decline, with larger declines in participants who forget more of the memories they suppressed[6,81,82]. If an action-stopping classifier detects the inhibition process, two findings related to conflict-reduction benefits should emerge.

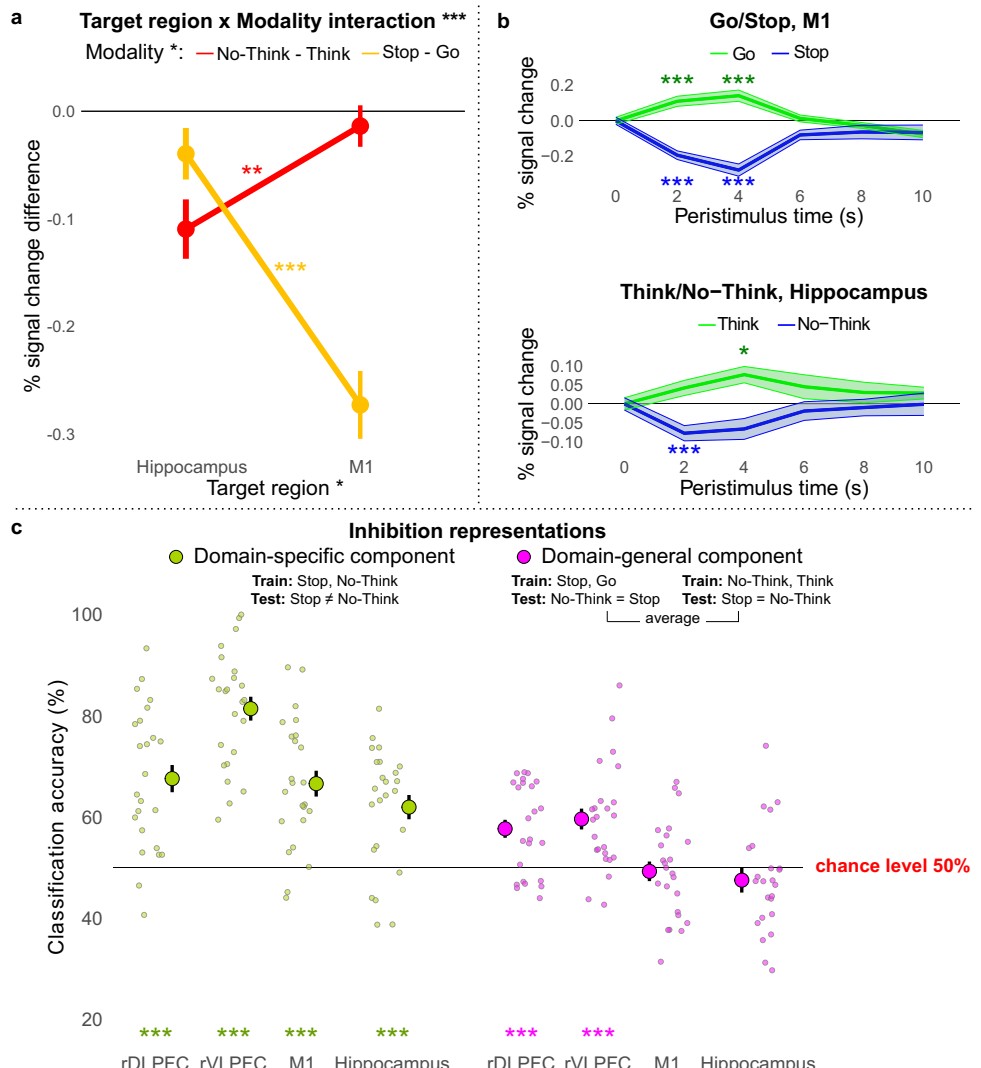

**Fig. 5 ROI analysis of domain-specific and domain-general modulation during thought and action suppression. a–c** ***$p < 0.001$; **$p < 0.01$; *$p < 0.05$. Data are presented as mean values. Error bars/shading represent within-subject standard error. Sample size $n = 24$. Source data are provided as a Source Data file. **a** Target areas M1 and hippocampus were modulated in a domain-specific manner. We calculated the BOLD signal in each target ROI for each condition by averaging across the time points from 2 to 6 s post-stimulus onset and subtracting out the onset value to account for pretrial variability. Then we subtracted the values of Go from Stop and Think from No-Think and entered them into a region by modality repeated-measures ANOVA. The ANOVA confirmed a significant interaction between modulatory target regions and stopping modality ($F_{1,23} = 45.99$, $p < 0.001$). Stopping actions (in yellow) evoked greater downregulation of M1 than of the hippocampus ($t_{23} = 6.26$, $p < 0.001$, $d = 1.279$) but suppressing thoughts (in red) evoked greater downregulation of the hippocampus than of M1 ($t_{23} = 3.53$, $p = 0.002$, $d = 0.720$). **b** The BOLD signal time-course in M1 (top panel) and hippocampus (bottom panel). During inhibition conditions (Stop and No-Think; in blue), the BOLD signal decreased below the baseline, whereas during respond conditions (Go and Think; in green) the BOLD signal increased above the baseline. Significance stars represent one-tailed one-sample $t$-test results, Bonferroni corrected for multiple comparisons. **c** Using MVPA, we tested whether action and thought inhibition share a common voxel activation pattern within the four ROIs. We performed two types of pattern classification to identify domain-general (cross-task classification; in violet) and domain-specific (between-task classification; in green) components within each ROI. Large circles represent group average classification accuracies, and small circles represent individual participant accuracies. The stars represent the significance of classification accuracy being above 50% chance level (Bonferroni corrected for the number of ROIs).

First, over Think/No-Think task blocks, the action-stopping classifier should discriminate thought suppression less well, with high classification in early blocks that drops as memories are inhibited. Second, this decline should be larger for people showing greater SIF.

We examined how accurately an action-stopping classifier distinguishes No-Think from Think conditions for the 8 fMRI runs (we note that there were three missing data points for the 8th run and one missing data point for the 7th run due to exclusion of some functional runs; see Methods). The rDLPFC showed a robust linear decline ($F_{7,157} = 9.61$, $p = 0.002$) in classification accuracy from the first ($M = 77\%$) to the eighth ($M = 40\%$) run (see Fig. 6a). This result is consistent with a conflict-reduction benefit and suggests that domain-general processes are especially important during early attempts at thought stopping. The rVLPFC exhibited a marginal linear decline ($F_{1,157} = 2.88$, $p = 0.092$) in classification accuracy from the first ($M = 67\%$) to the eighth ($M = 29\%$) run (see Fig. 6b). Critically, for both rDLPFC ($r_{ss} = -0.676$, $p < 0.001$; Fig. 6c) and rVLPFC ($r_{ss} = -0.570$, $p = 0.004$; Fig. 6d), participants showing

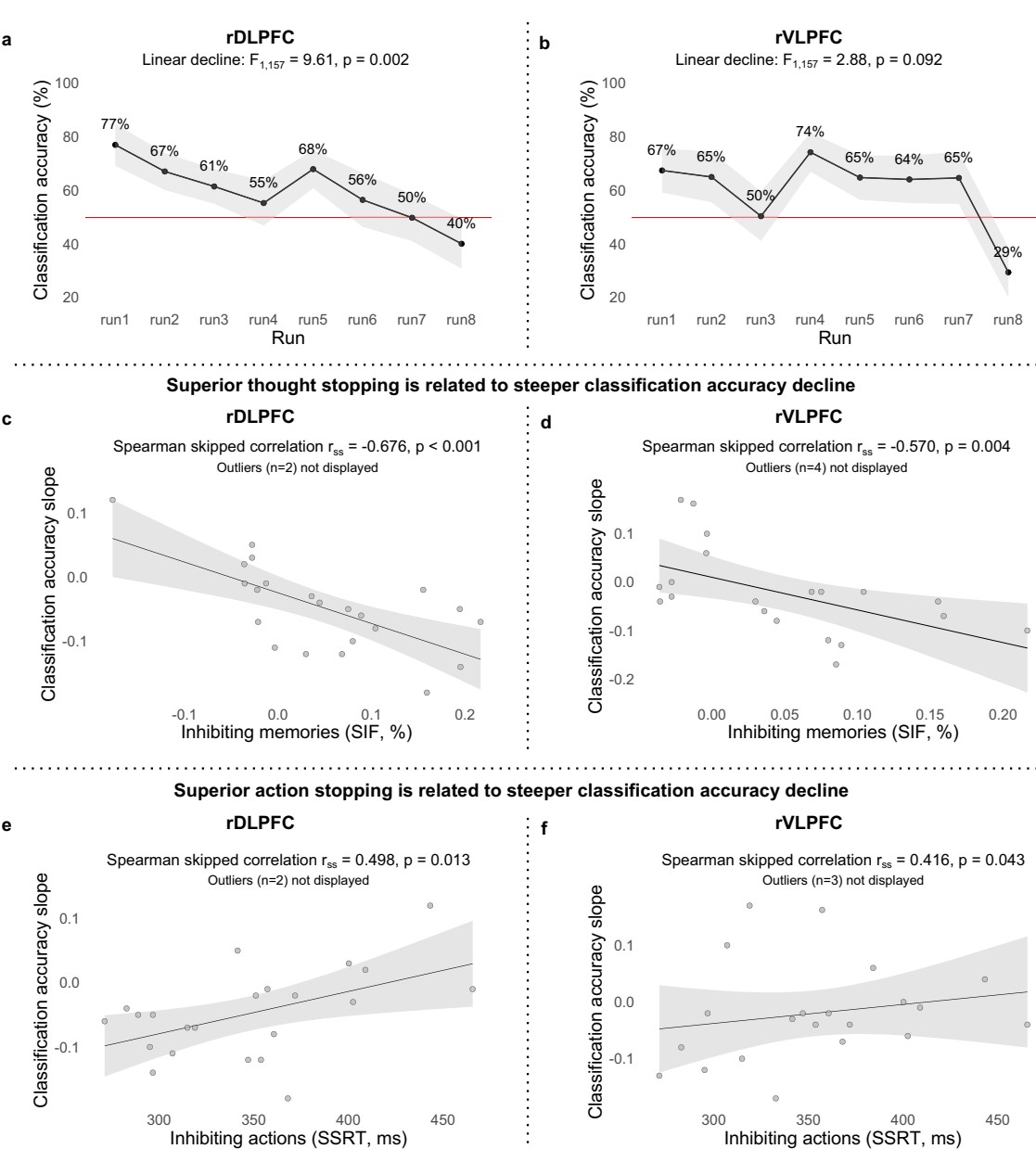

**Fig. 6 Conflict-reduction benefit. a–f** Sample size $n = 24$. Source data are provided as a Source Data file. **a**, **b** Accuracy of the action-stopping classifier (trained to distinguish Stop from Go conditions) to discriminate thought suppression (classifying No-Think as Stop) for each of the fMRI task runs in the rDLPFC and rVLPFC, respectively. Data are presented as mean values. Shading represents within-subject standard error. Linear decline was assessed by ANOVA linear contrast. **c**, **d** Correlation between the classification accuracy slope across the runs and suppression-induced-forgetting scores. Shading represents 95% CI. **e**, **f** Correlation between the classification accuracy slope across the runs and stop-signal reaction time. Shading represents 95% CI.

greater SIF exhibited a steeper classification accuracy decline. This suggests that adaptive forgetting had diminished demands on inhibitory control as blocks of thought suppression progressed. Notably, this cross-block decline should reduce the contribution of domain-general inhibition features to classifiers trained on all blocks together, deflating cross-domain prediction accuracy. If so, the cross-domain prediction accuracies reported for rDLPFC and rVLPFC in the preceding section underestimate the similarity of action and thought stopping. To further confirm that the conflict reduction benefit in the thought stopping task likely arises from a domain-general inhibition process, we related this decline to individual differences in motor inhibition speed.

Consistent with the involvement of inhibition, the decline in classifier performance was associated to SSRT for both rDLPFC ($r_{ss} = 0.498$, $p = 0.013$; Fig. 6e) and rVLPFC ($r_{ss} = 0.416$, $p = 0.043$; Fig. 6f). Together, these findings support the view that suppressing unwanted thoughts engages a domain-general inhibition process indexed by action stopping and suggests that both rDLPFC and rVLPFC support this process.

**Prefrontal areas shift coupling to inhibit target regions.** Although rDLPFC and rVLPFC contribute to action and thought stopping, it remained to be shown whether either or both regions causally modulate target regions during each task, one of the five

key attributes of dynamic targeting (Attribute 4: Causal modulation). On the one hand, rVLPFC alone might show dynamic targeting, exerting inhibitory modulation on the hippocampus or M1 in a task-dependent manner, as emphasised in research on motor response inhibition[10,11]; rDLPFC may only be involved to maintain the inhibition task set in working memory, possibly exerting a modulatory influence on rVLPFC to achieve this (rVLPFC alone model). On the other hand, rDLPFC alone might show dynamic inhibitory targeting, consistent with the emphasis on the rDLPFC as the primary source of inhibitory control in research on thought suppression[2,6]; rVLPFC may only be involved when attention is captured by salient stimuli[83,84], such as the stop signal or intrusions, possibly exerting a modulatory effect on rDLPFC to upregulate its activity (rDLPFC alone model). A third possibility is that rDLPFC and rVLPFC each contribute to top-down modulation in a content-specific manner, with only rDLPFC modulating the hippocampus during memory control, but only rVLPFC modulating M1 during action stopping. By this independent pathway hypothesis, both structures are pivotal to inhibitory control functions, but only with respect to their special domains, contrary to dynamic targeting. Finally, both rDLPFC and rVLPFC may be involved in dynamic targeting, modulating both hippocampus and M1 in a task-dependent manner; they may interact with one another to support stopping (Parallel modulation hypothesis).

To determine the way that rDLPFC and rVLPFC interact with each other and with the target regions (M1 and hippocampus), we analysed effective connectivity between regions using dynamic causal modelling (DCM, see Methods). DCM accommodates mono- and poly-synaptic mediation of the causal influence that prefrontal regions could exert on activity in the hippocampus and in M1[9]. DCM is ideally suited to test our hypotheses about which prefrontal regions drive inhibitory interactions, whether these vary by task context, and whether and how those prefrontal regions interact with one another to achieve inhibitory control during stopping.

Our model space included a null model with no modulatory connections and 72 distinct modulatory models (see Fig. 7a) differing according to whether the source-target modulation was bidirectional, top-down, or bottom-up, whether rDLPFC, rVLPFC or both were sources of modulation, whether rDLPFC and rVLPFC interacted during inhibition tasks, and whether the site on which top-down modulation acted was appropriate to the inhibition task or not. We first compared the null model and models in which the direction of source-target modulation was either bidirectional, top-down, or bottom-up (24 models in each of the three families). The findings from these connectivity analyses were unambiguous. Bayesian Model Selection (BMS) overwhelmingly favoured models with bidirectional connections between the sources (rDLPFC and rVLPFC) and targets (M1 and hippocampus) with an exceedance probability (EP) of 0.9999. In contrast, the null modulation, top-down, and bottom-up models had EP of 0/0.0001/0, respectively (see Fig. 7b). Exceedance probability refers to the extent to which a model is more likely in relation to other models considered. The bidirectional modulation confirms the existence of a top-down (our focus of interest) influence that prefrontal regions exert on activity in the hippocampus and in M1, alongside bottom-up modulation.

We next compared, within the 24 bidirectional models (models 1–24, see Fig. 7a), whether either rDLPFC or rVLPFC was the sole dominant top-down source of inhibitory control (rDLPFC only vs. rVLPFC only models) to models in which both regions comprised independent modulatory pathways (independent pathways model) or instead, contributed cooperatively to achieving top-down inhibitory control (parallel inhibition model). The BMS overwhelmingly favoured models in which

both rDLPFC and rVLPFC contributed to modulating both the hippocampus and M1 with an exceedance probability (EP) of 0.9999; in contrast, Independent Pathways, rDLPFC alone, and rVLPFC alone models had an EP of 0.0001/0/0, respectively (see Fig. 7c).

We next sought to distinguish subfamilies within this parallel model (models 9–12, and 21–24, see Fig. 7a) that varied according to whether and how rDLPFC and rVLPFC interacted during inhibition: no-interaction at all between rDLPFC and rVLPFC (none); unidirectional interaction from rVLPFC to rDLPFC (unidirectional rVLPFC); unidirectional interaction from rDLPFC to rVLPFC (unidirectional rDLPFC) and bidirectional interaction (rDLPFC and rVLPFC interact with each other). If rDLPFC and rVLPFC work as a functional unit to achieve inhibitory control, one would expect clear evidence that some form of interaction occurs. Consistent with this view, BMS strongly favoured models with bidirectional interactions between the rDLPFC and rVLPFC (EP = 0.91; EP for the none, unidirectional rDLPFC, and unidirectional rVLPFC being 0.01/0.07/0.02; see Fig. 7d).

Next, we tested whether inhibitory control is dynamically targeted to the appropriate target structure (e.g., hippocampus or M1), depending on which process needs to be stopped (memory retrieval or action production). According to our hypothesis, the rDLPFC and rVLPFC should down-regulate hippocampal activity during thought suppression, but should instead modulate M1, during action stopping (Attribute 5: Goal Dependence). To test this goal-dependence, we compared the two remaining models (12 and 24, see Fig. 7a) within our winning parallel/bidirectional subfamily. In the "preferred targets" model, rDLPFC and rVLPFC modulated the hippocampus during thought suppression, but M1 during action stopping; in the "non-preferred targets" model, these structures modulated content-inappropriate targets (e.g., M1 during thought suppression, but hippocampus during action stopping). BMS strongly favoured the model with preferred (EP = 0.95) over the non-preferred (EP = 0.05) target modulation (see Fig. 7e). Indeed, the overall winning model also was strongly favoured by BMS even when directly assessing all 73 models, side by side, without grouping them into model families and subfamilies (BMS = 0.92; see Fig. 7f).

The preferential modulations of hippocampus vs. M1, depending on whether thoughts vs. actions are to be suppressed, confirm our key hypothesis that top-down modulation by rDLPFC and rVLPFC is dynamically targeted depending on participants' task goals. However, a winning model with goal-dependent top-down connectivity to the hippocampus and M1 might be identified for any brain region robustly activated by both action and retrieval stopping, and not just the rDLPFC and rVLPFC. To test this possibility, we modified our DCM analysis by replacing the rDLPFC and rVLPFC nodes with two other regions from our meta-analytic conjunction analysis as sources of control. To choose regions, we performed our domain-general classification analysis on all ten meta-analytic conjunction regions (see Table 1). Apart from rDLPFC and rVLPFC, only the right and left inferior parietal lobule (IPL) exhibited significant domain-general components (see Supplementary Fig. 5). Using the right and left IPL as sources of control, DCM did not reveal a model with clear evidence for top-down modulation of hippocampus and M1 (see Supplementary Fig. 6). Thus, to be activated by both stopping tasks and to show cross-task decoding is not sufficient to infer goal-dependent inhibitory modulation of connectivity. Instead, our results suggest that rDLPFC and rVLPFC may be particularly important origins of this targeted signal. Together, the results of the DCM analysis suggest that, when stopping a prepotent response, rDLPFC and rVLPFC, interact with each other and are both selectively coupled with M1

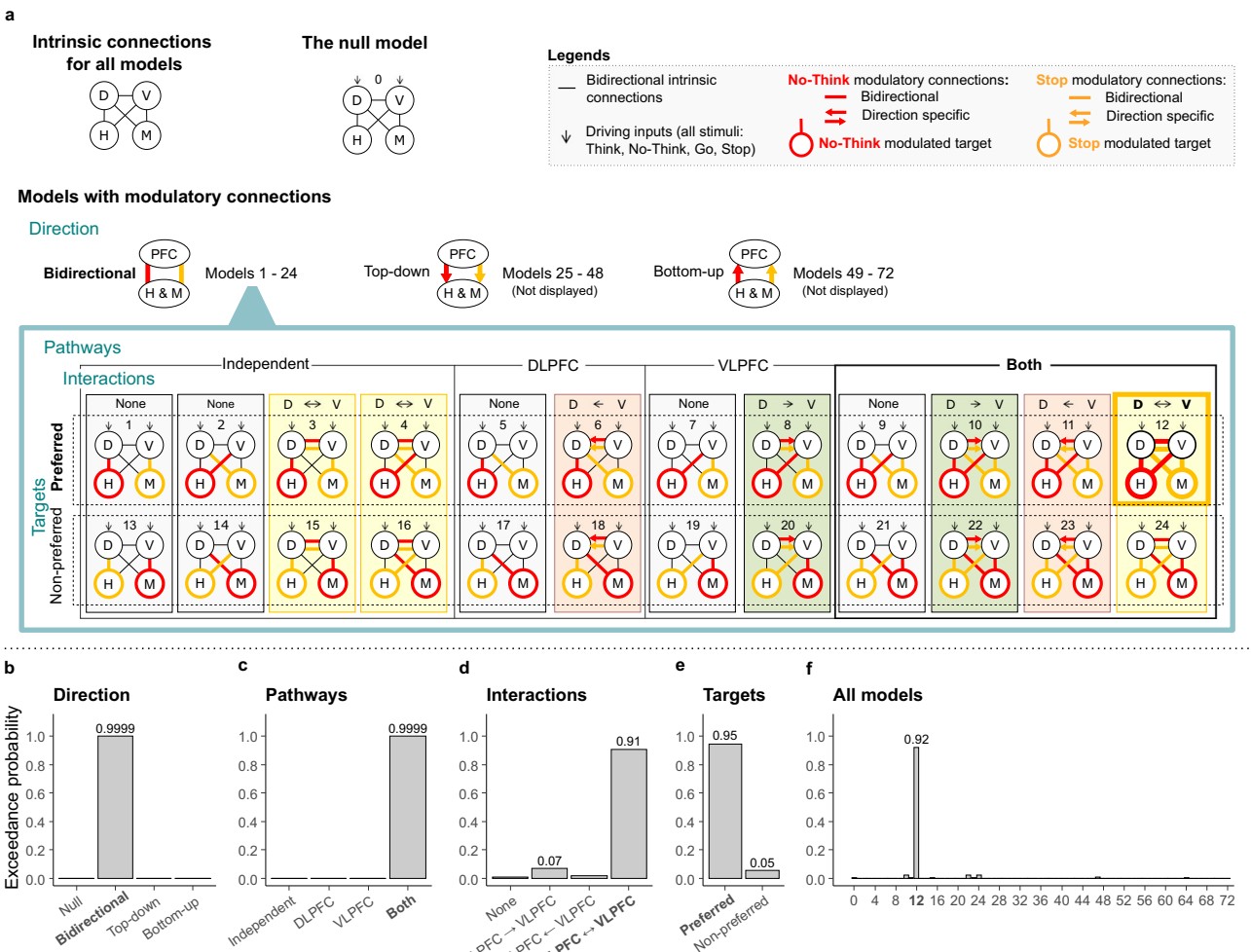

**Fig. 7 DCM model space and results. a** DCM analysis determined the most likely inhibition-related interactions between domain-general inhibitory control source areas (D: rDLPFC, V: rVLPFC) and domain-specific target areas (H: right hippocampus, M: left M1). We compared 73 alternative models grouped into four family types. Direction: three families according to whether the source-target modulation is bidirectional, top-down, or bottom-up (we display only the 24 models within the bidirectional family as the further grouping was identical within each of the three families). Pathways: four families differing according to how Stop and No-Think modulate the pathways: independent modulation of target regions by rDLPFC and rVLPFC; rDLPFC only modulation; rVLPFC only modulation; or modulation by both rDLPFC and rVLPFC. Interactions: four families differing according to how Stop and No-Think modulate interactions between the rDLPFC and rVLPFC regions: no interactions; rVLPFC modulates rDLPFC; rDLPFC modulates rVLPFC; or bidirectional interaction between rDLPFC and rVLPFC. Targets: two families differing according to whether Stop and No-Think modulate the prefrontal connectivity with the preferred targets (M1 when stopping actions and hippocampus when stopping thoughts) or with the non-preferred targets (hippocampus when stopping actions and M1 when stopping thoughts). BMS (reporting exceedance probability to which a model is more likely to other models considered) overwhelmingly favoured models with **b** bidirectional source-target modulation; **c** both rDLPFC and rVLPFC modulating both the hippocampus and M1; **d** bidirectional interactions between the rDLPFC and rVLPFC; **e** the preferred target modulation. **f** The overall winning model also was strongly favoured by BMS even when directly assessing all 73 models, side by side, without grouping them into model families. **b**–**f** Source data are provided as a Source Data file.

when stopping actions and selectively coupled with the hippocampus when stopping thoughts: in other words, both regions manifest dynamic targeting.

## Discussion
The current study identified two regions within the right prefrontal cortex that show dynamic targeting when stopping unwanted motor actions and thoughts. The rDLPFC and rVLPFC exhibited all five attributes needed to infer dynamic targeting. Both are engaged by diverse domains of inhibitory control, a finding supported not only by a within-subject conjunction analysis, but also via a meta-analytical conjunction; both show evidence of cross-task decoding. Both regions are relevant to

individual variation in inhibitory efficiency in both action stopping and thought suppression. Indeed, the multivariate activation pattern for action stopping resembled that for thought suppression sufficiently that it could be used as a proxy to predict how successfully people had suppressed their thoughts. Both regions are engaged alongside significant down-regulations in domain-specific target regions that we predicted a priori likely would require top-down inhibition; and both prefrontal regions show top-down effective connectivity with M1 and hippocampus during action stopping and thought suppression, supporting a causal role in their down-regulation. Critically, effective connectivity from both rDLPFC and rVLPFC to these two target regions shifted dynamically according to whether participants were stopping actions or thoughts, as expected of a domain-general

mechanism that is flexibly targeted to suppress specialised content in multiple domains.

Based on these and related findings, we propose that anterior rDLPFC and rVLPFC constitute key hubs for a domain-general inhibitory control mechanism that can be dynamically targeted to stop processing of diverse content represented throughout the brain. This proposal complements recent work positing a broad prefrontal inhibition mechanism that can interrupt both cognition and action[32,33,85]. We focused here on the stopping of simple manual actions and verbal thoughts. Given this approach, this study does not address the breadth of thought content that can be targeted by this mechanism. However, when considered alongside the growing literature on retrieval suppression, the breadth of content is considerable. For example, the anterior rDLPFC and rVLPFC regions identified in the meta-analytic conjunction have been observed during the suppression of a range of stimuli, including words[1,37,65], visual objects[40,41], neutral and aversive scenes[4,38,39,79] and person-specific fears about the future[7]. In addition, during retrieval suppression, these frontal regions exert top-down inhibitory modulation not only of the hippocampus[9,65], but also of other domain-specific content regions, including areas involved in representing visual objects[40,41], places[38,39], and also emotional content in the amygdala[4,39]. Content-specific modulations are triggered especially when these types of content intrude into awareness in response to a cue and need to be purged[39], indicating that inhibition can be dynamically targeted to diverse cortical sites to meet control demands. The current findings broaden the scope of this mechanism further by showing that it is not limited to stopping retrieval processes, but also extends to stopping the preparation and execution of motor responses, consistent with a broad mechanism involved in self-control over action and thought.

The proposed role of the rDLPFC and rVLPFC as hubs of domain-general inhibitory control during stopping does not imply that these regions are exclusively dedicated to stopping. Indeed, it seems likely that these regions contribute to many cognitive functions. Rather, the current evidence suggests that when stopping an action or thought is required, these regions are recruited to cancel processing in target areas involved in representing to-be-suppressed content, thereby achieving the desired stopping outcome. Methodologically diverse evidence indicates that this contribution is causally necessary to successful inhibitory control and is not a mere epiphenomenon of doing difficult tasks. First, the current effective connectivity analyses indicate a robust top-down modulation of target regions by these putative prefrontal sources. This finding comports well with lesion and brain stimulation work in both humans and animals, indicating that disrupting the function of rDLPFC and rVLPFC severely disrupts the capacity to stop, consistent with causal necessity[27–30]. Second, although action and thought stopping are both difficult tasks, the network dynamically reconfigured its connectivity to target regions to suppress their function in a manner compatible with task goals. These features have the hallmarks of a control process configured to implement a particular regulatory function, rather than a generic response to task difficulty. RDLPFC and rVLPFC are likely to work in concert with a broader network to achieve these goals, as our conjunction analyses suggest. The current work does not address the functional role of domain-general regions outside of the prefrontal cortex, the contributions of which should be examined in future work.

We considered the possibility that only one of these two frontal regions is central to implementing top-down inhibitory control during stopping, with the other providing upstream inputs essential to initiate successful control. Our effective connectivity analysis probed alternative hypotheses about the way rDLPFC and rVLPFC interact during stopping. RDLPFC might implement the true inhibitory signal, receiving salience detection input from rVLPFC that up-regulates rDLPFC function, consistent with a possible role of the VLPFC in the ventral attention network[83,84]. Alternatively, rVLPFC may implement inhibition, with rDLPFC preserving task set by sending driving inputs to the rVLPFC. Our findings indicate that both structures contributed in parallel to top-down inhibitory control and interacted bidirectionally during both action and thought stopping. Little evidence suggested a strong asymmetry in how rDLPFC and rVLPFC interacted, as should arise if one region simply served a role in salience detection or task-set maintenance. It remains possible, however, that rDLPFC and rVLPFC serve distinct functions that are not readily separable given the current manipulations and the level of temporal resolution available in fMRI data. Nevertheless, these findings suggest that rDLPFC and rVLPFC, at a minimum, act together to implement top-down inhibitory control during stopping. Although it might seem surprising that two spatially segregated prefrontal regions would act in concert to achieve this function, it seems less unusual considering their potential role in the cingulo-opercular network (CON). Most of the regions identified in our inhibition conjunction analysis participate in this network, suggesting that it may play an important role in achieving stopping. Given the strong integrated activity of this network, elements of which are distributed throughout the brain[51,54], this suggests future work should examine how rDLPFC and rVLPFC work together with other elements of this network to achieve successful stopping.

The current proposal contrasts with models that emphasise the primacy of either rVLPFC or rDLPFC in inhibitory control, and which have not addressed dynamic targeting to diverse content. Research on motor inhibition has emphasised the rVLPFC as the source of top-down inhibitory control[10,11], although without evidence to exclude the role of rDLPFC. Indeed, studies cited as favouring the selective role of rVLPFC often support contributions of the anterior rDLPFC structure identified here. For example, whereas intracranial stimulation in primates establishes the causal necessity of the rVLPFC in motor stopping, so too does stimulation of the dorsal bank of the principal sulcus, the putative monkey homologue of the rDLPFC in humans[30]; and whereas intracranial recordings in humans show stopping-related activity in rVLPFC, they also reveal it in anterior rDLPFC and often prior to rVLPFC[86]. Research on thought suppression has emphasised the rDLPFC as the source of top-down inhibitory control[1,2,9]; but most studies supporting the role of rDLPFC in thought suppression also reveal activations in the rVLPFC[22]. Indeed, as our within-subjects and meta-analytic conjunctions unambiguously confirm, both regions are recruited during both stopping tasks. The current study goes further than establishing conjoint activation: Pattern classification and connectivity analyses show the involvement of both regions in the dynamics of control, without selectivity. These findings validate the importance of both regions, establish the domain-generality of their influence, and demonstrate the dynamic inhibitory targeting capacity necessary to infer a flexible control mechanism.

Although rDLPFC and rVLPFC exhibit core properties needed to achieve domain-general stopping, there is also evidence of domain-specificity within these regions. A classifier trained to distinguish stopping actions from stopping thoughts performed well. Although seemingly inconsistent with domain generality, such effects can be understood as consequences of the dynamic targeting functionality afforded by these regions. For a region to serve as a flexible hub of inhibitory control, it must be able to receive inputs from diverse cortical sites, representing information needed to drive the stopping (e.g., the perception of the coloured word for the No-Think task, and of the tone for the stop-signal task). Moreover, to flexibly target inhibitory control,

source regions must interact with diverse cortical and subcortical targets (e.g., hippocampus vs. M1), the processing which must be stopped. These input and output processes may manifest in unique multivariate patterns over common voxels within each region. We cannot rule out, however, the possibility that domain-specific inhibition processes are also manifested by unique patterns in these regions. Future work should examine the additional contributions of input, output, and domain-specific stopping features to the activations found in rDLPFC and rVLPFC.

The present findings highlight a potentially important difference between the brain networks involved in stopping and other forms of cognitive control that do not require the full cessation of a motor or cognitive process. Maintaining rules in working memory, implementing task sets, performing multi-tasking, and manipulating information actively are all clear cases of cognitive control that can require interference resolution, but do not necessarily entail active stopping. The above functions engage the widely discussed frontoparietal network (FPN), often assigned a central role in implementing cognitive control more broadly[50–53]. One might assume that because stopping is a form of cognitive control that the FPN would be central to it as well. Nevertheless, the FPN, though involved in our tasks, appeared less prominent than the CON, which accounted for most of the distinct cortical parcels participating in our domain-general stopping regions. We found little evidence for the involvement of major areas of the FPN, including much of the middle frontal gyrus bilaterally in our multimodal inhibition regions. As our meta-analysis and within-subjects comparisons confirm, inhibitory control during stopping is strongly right lateralised, which also is not a feature emphasised in research on the FPN. Our findings raise the possibility that stopping actions and thoughts may rely on a distinct network, with different functional characteristics to the FPN. Whether other functions thought to require inhibitory control (e.g., selection between competing responses, as in Stroop or Flanker interference) also preferentially engage the stopping networks identified here is not addressed in the current work, although some empirical precedents suggest that stopping and selection may engage partially distinct mechanisms[26,37,87].

Dynamic inhibitory targeting provides a neurocognitive framework that can account for both associations and dissociations in the abilities to suppress unwanted thoughts and actions. On the one hand, deficits in both action and thought stopping should arise with dysfunction in the rDLPFC or rVLPFC, given the common reliance of these abilities on those regions. Such associations occur frequently. In the general population, people scoring highly on self-report measures of impulsivity or compulsivity also report greater difficulty with intrusive thoughts[88,89]. Clinically, persistent intrusive thoughts and action stopping deficits co-occur in numerous disorders: Obsessive thoughts and compulsive actions in obsessive-compulsive disorder[90,91]; intrusive memories and impaired response inhibition in PTSD[92–96]; persistent worry and impulsivity in anxiety disorders[97] and intrusive thoughts and compulsivity in addiction[98–100]. These co-morbid deficits may reflect dysfunction in the rDLPFC, the rVLPFC or in other shared components of their control pathways. On the other hand, dissociations should arise when dysfunction selectively disrupts a domain-specific pathway linking rLPFC to target sites involved in generating actions and thoughts, including dysfunction to local inhibition at the target site itself. For example, individual variation in local GABAergic inhibition within the hippocampus or M1 predict inhibitory control over memories and actions, respectively, independently of prefrontal function[8,49]. Thus, selective difficulties in action stopping or thought inhibition may arise, given focal deficits in either motor cortical or hippocampal GABA[8]. The separate contributions of domain-general and domain-specific factors to inhibitory control

implied by dynamic targeting constrains the utility of motor inhibition as a metric of inhibitory control over thought and may explain the surprisingly small SSRT deficits in major depression and anxiety, relative to attention deficit hyperactivity disorder or obsessive-compulsive disorder[19].

The current study did not seek to characterise the polysynaptic pathways through which the rDLPFC and rVLPFC suppress activity in either M1 or the hippocampus[5,9]. Rather, we focused on the existence of a central, domain-general inhibitory control function capable of flexibly shifting its top-down influence to stop actions and thoughts. By juxtaposing two well characterised model systems for stopping actions and thoughts, each with distinct neural targets of inhibition, we were able to show that the same set of prefrontal regions is involved in stopping processing in different cortical target areas, in a rapid, flexible manner. In doing so, we established evidence for dynamic inhibitory targeting as a key mechanism of domain-general inhibitory control during stopping in the human brain. More broadly, this work suggests that the human capacity for self-control in the face of life's challenges may emerge from a common wellspring of control over our actions and thoughts.

## Methods
We used a dataset from a published study[8]. However, here the data were independently re-analysed with a different focus.

**Participants**. Thirty right-handed native English speakers participated. Participants gave written informed consent and received money for participating. Five participants did not reach the 40% learning criterion on the Think/No-Think task, and one fell asleep during fMRI acquisition. The final sample comprised 24 participants (7 males, 17 females), 19–36 years old ($M = 24.67$ years, SD = 4.31). Participants had normal or corrected-to-normal vision and no reported history of neurological, medical, or memory disorders, and they were asked not to consume psychostimulants, drugs, or alcohol before the experiment. The Cambridge Psychology Research Ethics Committee approved the project.

**Experimental paradigm**. Participants performed adapted versions of the Stop-signal[20] and Think/No-Think[62] tasks. Both tasks require participants to stop unwanted processes, but in the motor and memory domains, respectively.

The Stop-signal task assesses the ability to stop unwanted actions. Participants first learn stimulus–response associations and then perform speeded motor responses to the presented (Go) stimuli. Occasionally, shortly after the Go stimulus, a stop signal occurs, and participants must withhold their response. We measured the stop-signal reaction time (SSRT), an estimate of how long it takes the participant to stop.

The Think/No-Think task assesses the ability to stop unwanted memory retrievals. Participants first form associations between unrelated cue-target word pairs. Then participants receive two-thirds of the cues as reminders (one at a time) and are asked to either think (Think items) or to not-think (No-Think items) of the associated target memory, with each Think and No-Think repeated numerous times throughout the task. Finally, participants attempt to recall all initially learned associations. Typically, recall performance suffers for No-Think items compared to Baseline items that were neither retrieved nor suppressed during the think/no-think phase. This phenomenon, known as suppression-induced forgetting (SIF), indirectly measures the ability to stop unwanted memory retrievals by quantifying inhibitory aftereffects of this process[2,101].

**Stimuli and apparatus**. We presented stimuli and recorded responses with Presentation software (Neurobehavioral Systems, Albany, CA, USA). For the Stop-signal task, four visually discriminable red, green, blue, and yellow coloured circles of 2.5 cm in diameter, presented on a grey background, constituted the Go stimuli (Fig. 2a). Participants responded by pressing one of the two buttons (left or right) with a dominant (right) hand on a button box. An auditory 1000 Hz "beep" tone presented at a comfortable volume for 100 ms signalled participants to stop their responses. A fixation cross appeared in 50-point black Arial Rounded font on a grey background prior to the onset of the Go stimulus.

For the Think/No-Think task, we constructed 78 weakly relatable English word pairs (cue-target words, e.g., Part-Bowl) as stimuli and an additional 68 semantically related cue words for 68 of the target words (e.g., a cue word 'Cornflake' for the target word 'Bowl'). We used 60 of the target words and their related and weak cues in the critical task, with the other items used as fillers. We divided the critical items into three lists composed of 20 targets and their corresponding weak cue words (the related word cues were set aside to be used as independent test cues on the final test; see procedure). We counterbalanced these lists across the within-subjects

experimental conditions (Think, No-Think, and Baseline) so that across all participants, every pair participated equally often in each condition. We used the filler words both as practice items and also to minimise primacy and recency effects in the study list[102]. Words appeared in a 32-point Arial font in capital letters on a grey background (Fig. 2b). During the initial encoding and final recall phases, we presented all cues and targets in black. For the Think/No-Think phase, we presented the Think cues in green and the No-Think cues in red, each preceded by a fixation cross in 50-point black Arial Rounded font on a grey background.

**Procedure**. The procedure consisted of seven steps: (1) stimulus-response learning for the Stop-signal task: (2) Stop-signal task practice; (3) encoding phase of the Think/No-Think task; (4) Think/No-Think practice; (5) practice of interleaved Stop-signal and Think/No-Think tasks; (6) experimental phase during fMRI acquisition; (7) recall phase of the Think/No-Think task. We elaborate these steps below (see also Fig. 2c).

*Step 1—Stop-signal task stimulus-response learning*. Participants first formed stimulus-response associations for the Stop-signal task. As Go stimuli, we presented circles in four different colours (red, green, blue, and yellow) and participants had to respond by pressing one of the two buttons depending on the circle's colour. Thus, each response button had two colours randomly assigned to it and participants associated each colour to its particular response.

Participants learned the colour-button mappings in two sets of two colours, with the first colour in a set associated with one button, and the second with the other button. After practising the responses to these colours in random order 10 times each, the same training was done on the second set. Subsequently, participants practised the colour-button mappings of all four colours in random order until they responded correctly to each colour on 10 consecutive trials. During the practice, we instructed participants to respond as quickly and accurately as possible and provided feedback for incorrect or slow (>1000 ms) responses.

*Step 2—Stop-signal task practice*. Once participants learned the stimulus–response associations, we introduced the Stop-signal task. We instructed participants to keep responding to each coloured circle as quickly and accurately as possible but indicated that on some trials, after the circle appeared, a beep would sound and that they should not press any button on these trials. We also told participants to avoid slowing down and waiting for the beep, requesting instead that they treat failures to stop as normal and always keep responding quickly and accurately. Thus, on Go trials, participants responded as quickly as possible, whereas, on Stop trials, a tone succeeded the cue onset, signalling participants to suppress their response. To facilitate performance, participants received on-screen feedback for incorrect and too slow (>700 ms) responses to Go trials, and for pressing a button on Stop trials.

Figure 2a presents the trial timings. Go trials started with a fixation cross, presented for 250 ms, followed by a coloured circle until response or for up to 2500 ms. After the response and a jittered inter-trial interval ($M = 750$ ms, $SD = 158.7$ ms), a new trial commenced. Stop trials proceeded identically except that a tone sounded shortly after the circle appeared. This stop signal delay varied dynamically in 50 ms steps (starting with 250 or 300 ms) according to a staircase tracking algorithm to achieve approximately a 50% success-to-stop rate for each participant. Note that the longer the stop signal delay is, the harder it is to not press the button. The dynamic tracking algorithm reduces participants' ability to anticipate stop signal delay timing and provides a method for calculating the SSRT. In this practice step, participants performed 96 trials, of which 68 (71%) were Go trials and 28 (29%) were Stop trials.

*Step 3—Think/No-Think task encoding phase*. Once participants had learned the Stop-signal task, we introduced the Think/No-Think task. In the encoding phase, participants formed associations between 60 critical weakly related word pairs (e.g., Part-Bowl) and between 18 filler pairs. First, participants studied each cue-target word pair for 3.4 s with an inter-stimulus interval of 600 ms. Next, from each studied pair, participants saw the cue word only and recalled aloud the corresponding target. We presented each cue for up to 6 s or until a response was given. 600 ms after cue offset, regardless of whether the participant recalled the item, the correct target appeared for 1 s. We repeated this procedure until participants recalled at least 40% of the critical pairs (all but 5 participants succeeded within the maximum of three repetitions). Finally, to assess which word-pairs participants learned, each cue word appeared again for 3.3 s with an inter-stimulus interval of 1.1 s and participants recalled aloud the corresponding target. We provided no feedback on this test.

*Step 4—Think/No-Think practice*. After participants encoded the word pairs, the Think/No-Think practice phase commenced. On each trial, a cue word appeared on the screen in either green or red. We instructed participants to recall and think of the target words for cues presented in green (Think condition) but to suppress the recall and avoid thinking of the target words for those cues presented in red (No-Think condition). Participants performed the direct suppression variant of the Think/No-Think task[37,103] in which, after reading and comprehending the cue, they suppressed all thoughts of the associated memory without engaging in any

distracting activity or thoughts. We asked participants to "push the memory out of mind" whenever it intruded.

Trial timings appear in Fig. 2b. A trial consisted of presenting a cue in the centre of the screen for 3 s, followed by an inter-stimulus interval ($≥0.5$ s, $M = 2.3$ s, $SD = 1.7$ s) during which we displayed a fixation cross. We jittered the inter-stimulus interval ($≥0.5$ s, $M = 2.3$ s, $SD = 1.7$ s) to optimise the event-related design (as determined by optseq2: http://surfer.nmr.mgh.harvard.edu/optseq).

In this practice phase, we used 12 filler items, six of which were allocated to the Think condition and six to the No-Think condition. We presented each item three times in random order (36 trials in total). In the middle of the practice, we administered a diagnostic questionnaire to ensure participants had understood and followed the instructions.

*Step 5—Interleaved Stop-signal and Think/No-Think practice*. Before moving into the MRI scanner, participants performed an extended practice phase interleaving the Stop-signal and Think/No-Think tasks. For the Think/No-Think task, we again used 12 filler items. Other than that, and the fact that the practice took place outside the MRI scanner, this phase was identical to a single fMRI acquisition session described into more detail next.

*Step 6—Experimental phase and fMRI acquisition*. In the main experimental phase, participants underwent 8 fMRI scanning runs in a single session. Before the scanning began, participants saw the correct button-colour mappings and all 78-word pairs briefly presented on the screen to remind them of the task and items. After the brief refresher, the fMRI acquisition started. During each fMRI run, participants performed 8 blocks of the Think/No-Think task interleaved with 8 blocks of the Stop-signal task. All blocks lasted 30 s. To minimise carry-over effects, we interspersed 4 s rest periods (blank screen with a grey background) between blocks. Each block began with items that we did not score (the filler items) to reduce task-set switching effects between blocks. Within each block, we pseudo-randomly ordered all trials, and the trial timings for both tasks were identical to those used in their respective practice phases (steps 2 and 4; Fig. 2a and b).

Four of the Stop-signal task blocks contained Go trials only. We did not use these blocks in this report. Each of the other four Stop-signal blocks contained 12 trials, yielding 384 trials in total (8 runs * 4 blocks per run * 12 trials per block). On average, across participants, Stop trials constituted 32% ($SD = 2$%) of the trials. As in the practice phase, a staircase tracking algorithm varied the delay between cue onset and stop-signal tone according to each participant's performance, keeping the stopping success at ~50%.

Each of the Think/No-Think blocks contained 6 trials, starting with a filler item as a Think trial followed by 5 Think or No-Think items in a pseudo-random order. Within each fMRI run, participants saw all 20 critical Think and 20 critical No-Think items once. Thus, across the 8 runs, participants recalled or suppressed each memory item 8 times. The proportion of the Think trials (58%) exceeded the proportion of the No-Think trials (42%) to better resemble the higher frequency of Go trials than Stop trials during the Stop-signal task. We accomplished this by assigning Think trials to the filler items, without changing the frequency of Think trials on critical experimental items. After the fourth (middle) run, to allow participants to rest, we acquired their anatomical scan and administered the diagnostic questionnaire to ensure that participants closely followed the instructions of the Think/No-Think task.

*Step 7—Think/No-Think recall phase*. In the final step (inside the scanner but without any scan acquisition), we measured the aftereffects of memory retrieval and suppression via a cued-recall task on all word pairs (encoded in step 3). This included 20 Baseline items that were neither retrieved nor suppressed during the Think/No-Think phase and that this provided a baseline estimate of the memorability of the pairs.

To reinstate the context of the initial encoding phase, we first tested participants on 10 filler cue words, 6 of which they had not seen since the encoding phase (step 3) and 4 of which they saw during the interleaved Stop-signal and Think/No-Think practice phase (step 5). We warned participants that the cues in this phase could be ones they had not seen for a long time and encouraged them to think back to the encoding phases to retrieve targets.

Following context reinstatement, participants performed the same-probe and independent-probe memory tests. In the same-probe test, we probed memory with the original cues (e.g., the weakly related cue word 'Part' for the target word 'Bowl'). We included the independent-probe test to test whether forgetting generalised to novel cues[62], using the related cues we had designed for each target. For example, we cued with the semantic associate of the memory and its first letter (e.g., 'Cornflake—B' for the target 'Bowl'). Across participants, we counterbalanced the order in which the tests appeared. In both tests, cues appeared for a maximum of 3.3 s or until participants gave a response, with an inter-stimulus interval of 1.1 s. We coded response as correct if participants correctly recalled the target while the cue was onscreen.

Finally, we debriefed participants and administered a post-experimental questionnaire to capture participants' experiences and the strategies they used in the Think/No-Think and Stop-signal tasks.

**Brain image acquisition**. We collected MRI data using a 3-T Siemens Tim Trio MRI scanner (Siemens, Erlangen, Germany) fitted with a 32-channel head coil.

Participants underwent eight functional runs of the blood-oxygenation-level-dependent (BOLD) signal acquisitions. We acquired functional brain volumes using a gradient-echo, T2*-weighted echoplanar pulse sequence (TR = 2000 ms, TE = 30 ms, flip angle = 90°, 32 axial slices, descending slice acquisition, voxel resolution = 3 mm³, 0.75 mm interslice gap). We discarded the first four volumes of each session to allow for magnetic field stabilisation. Due to technical problems encountered during task performance, we discarded from the analysis one functional run from two participants each, and two functional runs from another participant. After the fourth functional run, we acquired an anatomical reference for each participant, a high-resolution whole-brain 3D T1-weighted magnetisation-prepared rapid gradient echo (MP-RAGE) image (TR = 2250 ms, TE = 2.99 ms, flip angle = 9°, field of view = 256 × 240 × 192 mm, voxel resolution = 1 mm³). Following the acquisition of the anatomical scan, participants underwent the remaining four functional runs.

**Behavioural performance analysis**. For statistical analyses of the behavioural data, we used R (v4, 2020-04-24) in Jupyter Notebook (Anaconda, Inc., Austin, TX). The data and detailed analysis notebook are freely available at GitHub repository[104] (https://github.com/dcdace/Domain-general/). For all statistical comparisons, we adopted $p < 0.05$ as the significance threshold.

For correlation analyses, we followed recommendations by Pernet et al.[105] and used one of three correlation methods depending on whether the data were normally distributed or contained outliers. If there were no outliers and data were normally distributed, we performed Pearson correlation and reported it as '$r$'. If there were univariate outliers (but no bivariate) or data were not normally distributed, we performed robust 20% Bend correlation and reported it as '$r_{pp}$'. If there were bivariate outliers, we performed robust Spearman skipped correlation using the minimum covariance determinant (MCD) estimator and reported it as '$r_{ss}$'. For univariate and bivariate outlier detection, we used boxplot and bagplot methods, respectively.

For the analysis of Stop-signal task data, we followed the guidelines by Verbruggen et al.[64] and calculated SSRT using the integration method with the replacement of Go omissions. Specifically, we included all Stop trials and all Go trials (correct and incorrect), replacing missed Go responses with the maximum Go RT. To identify the nth fastest Go RT, we multiplied the number of total Go trials by the probability of responding to stop signal (unsuccessful stopping). The difference between the nth fastest Go RT and the mean SSD provided our estimate of SSRT.

In addition to SSRT, we calculated the probability of Go omissions, probability of choice errors on Go trials, probability of responding to Stop trials, mean SSD of all Stop trials, mean correct Go RT, and mean failed Stop RT. We also compared RTs of all Go trials against RTs of failed Stop trials to test the assumption of an independent race between a go and a stop runner. Besides, we assessed the change of Go RTs across the eight experimental blocks. Prior work suggests that the experiment-wide integration method can result in underestimation bias of SSRT if participants slow their RT gradually across experimental runs. In that case, a blocked integration method would provide a better measure of SSRT[106]. In our data, however, on average within the group, we observed a negligible decrease in RT across runs ($B = -2.555$, $p = 0.250$), suggesting that the experiment-wide integration method was more appropriate.

For the Think/No-Think task data, we focused on the critical measure: SIF. We used the final recall scores (from step 7) of No-Think and Baseline items conditionalized on correct initial training performance (at step 3), as in prior work[1]. Thus, in the final recall scores, we did not include items that were not correctly recalled ($M = 29\%$, $SD = 17$) during the criterion test of the encoding phase, as the unlearned items can be neither suppressed nor retrieved during the Think/No-Think phase (step 6). As in our previous work[8], we averaged the scores across the same-probe and independent-probe tests and the difference between the Baseline and No-think item recall scores constituted our measure of SIF. To assess the group effect of SIF, we tested the data for normality ($W = 0.95$, $p = 0.264$) and performed a one-sample, one-sided $t$-test to determine if SIF is greater than zero. Finally, to assess whether inhibition ability generalises across motor and memory domains, we performed a correlation between the SSRT and SIF scores.

To identify univariate and bi-variate outliers in the SSRT and SIF scores, we used box plot method, which relies on the interquartile range. Univariate outliers were not present for any of the two measures. One bi-variate outlier was removed from the correlation analysis and the behavioural partial least squares analysis (described below). Nevertheless, outlier removal did not qualitatively alter the results.

**Brain imaging data analysis**

*Pre-processing*. We pre-processed and analysed the brain imaging data using Statistical Parametric Mapping v12 release 7487 (SPM12; Wellcome Trust Centre for Neuroimaging, London) in MATLAB vR2012a (The MathWorks, MA, USA). To approximate the orientation of the standard Montreal Neurological Institute (MNI) coordinate space, we reoriented all acquired MRI images to the anterior-posterior commissure line and set the origins to the anterior commissure. Next, we applied our pre-processing procedure to correct for head movement between the scans (images realigned to the mean functional image) and to adjust for temporal differences between slice acquisitions (slice-time correction relative to the middle axial

slice). The procedure then co-registered each participant's anatomical image to the mean functional image and segmented it into grey matter, white matter, and cerebrospinal fluid. We then submitted the segmented images for each participant to the DARTEL procedure[107] to create a group-specific anatomical template that optimises inter-participant alignment. The DARTEL procedure alternates between computing a group template and warping an individual's tissue probability maps into alignment with this template and ultimately creates an individual flow field of each participant. Subsequently, the procedure transformed the group template into MNI-152 space. Finally, we applied the MNI transformation and smoothing with an 8 mm full-width-at-half-maximum (FWHM) Gaussian kernel to the functional images for the whole-brain voxel-wise analysis.

*Univariate whole-brain analysis*. To identify brain areas engaged in both inhibiting actions and inhibiting memories, we performed a whole-brain voxel-wise univariate analysis. We high-pass filtered the time series of each voxel in the normalised and smoothed images with a cut-off frequency of 1/128 Hz, to remove low-frequency trends, and modelled for temporal autocorrelation across scans with the first-order autoregressive (AR(1)) process. We then submitted the pre-processed data of each participant to the first-level, subject-specific, general linear model (GLM) modelling a single design matrix for all functional runs.

We modelled the Stop-signal task and Think/No-Think task conditions as boxcar functions, convolved with a haemodynamic response function (HRF). In the model, we used group average response latencies for each trial type as the trial durations for the Stop-signal task condition, but we used 3 s epochs for the Think/No-Think task condition. As in the behavioural analysis, we conditionalized the Think and No-Think conditions on initial encoding performance. The main conditions of interest for our analysis included: correct Stop, correct Go (from the mixed Stop-signal and Go trial blocks only), conditionalized No-Think and conditionalized Think. Unlearned No-Think and Think items, filler items, incorrect Stop, incorrect Go and Go trials from the Go-only blocks we modelled as separate regressors of no interest. We also included the six realignment parameters for each run as additional regressors of no interest, to account for head motion artefacts, and a constant regressor for each run. We obtained the first-level contrast estimates for Stop, Go, No-Think, and Think conditions, and the main effect of Inhibit [Stop, No-Think] > Respond [Go, Think].

At the second-level random-effect group analysis we entered the first-level contrast estimates of Stop, Go, No-Think, and Think conditions into a repeated-measures analysis of variance (ANOVA), which used pooled error and correction for non-sphericity, with participants as between-subject factor. We then performed a conjunction analysis of Stop > Go and No-Think > Think contrasts, using the minimum statistics analysis method implemented in SPM12, and testing the conjunction null hypothesis[108,109]. The results of the conjunction analysis represent voxels that were significant for each individual contrast thresholded at $p < 0.05$ false discovery rate (FDR) corrected for whole-brain multiple comparisons.

*Behavioural partial least squares (PLS) analysis*. We hypothesised that domain-general inhibitory control brain activity would be related to domain-general inhibitory behaviour. To test our hypothesis, we performed behavioural PLS analysis[110,111] following a previously employed strategy[39]. We restricted our analysis to an independent domain-general inhibitory control mask derived from a meta-analytic conjunction analysis of 40 Stop-signal and 16 Think/No-Think fMRI studies (described below). Within this mask, we identified voxels where the BOLD signal from the main effect of Inhibit > Respond contrast depicted the largest joint covariance with the SSRT and SIF scores.

Specifically, Inhibit > Respond contrast values from each voxel of an MNI-normalised brain volume were aligned and stacked across participants into a brain activation matrix X, and SSRT and SIF scores were entered into a matrix Y. In both matrices, rows represented participants. We then individually mean-centred the X and Y matrices and normalised each row in the matrix X (representing each participant's voxel activations) so that the row sum of squares equalled to one. Setting an equal variance of voxel activities across subjects ensured that the observed differences between participants were not due to overall differences in activation. Hereafter, a correlation of X and Y matrices produced a matrix R encoding the relationship between voxel activity and behavioural scores across participants. We then applied a singular-value decomposition to the correlation matrix R to identify LVs that maximise the covariance between voxel activation (X) and behavioural measurements (Y). Each LV contains a singular value, singular image, and correlation profile. The singular value represents the amount of the covariance explained by the LV. The singular image identifies a collection of voxels that, as a group, are most related to the effects expressed in the LV. The numerical weights within the singular image are called brain saliences and represent the strength of the relationship between the BOLD signal and the behavioural scores. A correlation profile represents how the behaviour correlates with the pattern of brain activity identified in the singular saliences image. A dot product of subject's raw image volume and the singular saliences image produces a brain score for each subject. Brain scores indicate how strongly individual subjects express the pattern on the LV.

To assess the statistical significance of each LV and the robustness of voxel saliences, we used 5000 permutation tests and 5000 bootstrapped resamples, respectively. By dividing each voxel's initial salience by the standard error of its bootstrapped distribution, we obtained a bootstrapped standard ratio, equivalent to a $z$-score, to assess the significance of a given voxel. We thresholded the acquired

scores at 1.96, corresponding to $p < 0.05$, two-tailed. The multivariate PLS analysis method does not require correction for multiple comparisons as it quantifies the relationship between the BOLD signal and behavioural scores in a single analytic step[110].

*Dynamic causal modelling (DCM) analysis.* We conducted a DCM analysis[66] to determine the most likely inhibition-related interactions between domain-general inhibitory control areas in the right prefrontal cortex and domain-specific target areas. For the domain-specific target areas, we selected the left primary motor cortex (M1) and right hippocampus, based on our previous findings showing that stopping actions and stopping memories preferentially downregulates M1 and hippocampus, respectively[8].

DCM enables one to investigate hypothesised interactions among pre-defined brain regions by estimating the effective connectivity according to (1) the activity of other regions via intrinsic connections; (2) modulatory influences on connections arising through experimental manipulations; and (3) experimentally defined driving inputs to one or more of the regions[66]. The intrinsic, modulatory, and driving inputs one specifies constitute the model structure assumed to model the hypothesised neuronal network underlying the cognitive function of interest.

With DCM, a set of models can be defined that embody alternate hypotheses about the average connectivity and conditional moderation of connectivity. These models are inverted to the data and then compared in terms of the relative model evidence using Bayesian model selection (BMS). The differential model evidence from BMS indicates the probability that a given model is more likely to have generated the data than the other models and allows to infer both the presence and direction of modulatory connections. This can be estimated for individual models, or families of models that share critical features.

For the DCM analysis, we defined four regions of interest (ROIs): the right dorsolateral prefrontal cortex (rDLPFC), the right ventrolateral prefrontal cortex (rVLPFC), the right hippocampus, and the left M1. We obtained the rDLPFC and rVLPFC ROIs, centred at MNI coordinates 35, 45, 24 and 44, 21, −1, respectively, from an independent meta-analytic conjunction analysis (described below). We defined the M1 ROI (centred at MNI coordinates −33, −22, 46) from a group analysis ($N = 30$) of an independent M1 localiser study on different participants (Button Press > View contrast). We mapped the rDLPFC, rVLPFC, and M1 ROIs from the MNI space to participants' native space. We manually traced the hippocampal ROIs in native space for each participant, using ITK-SNAP v3.8.0 (www.itksnap.org[112]) and following established anatomical guidelines[113,114]. Within each subject-specific ROI, we identified all significant voxels (thresholded at $p < 0.05$, uncorrected for multiple comparisons) for that participant based on the main effect of interest, which included Stop, Go, No-Think, and Think conditions. This selection was performed on the first-level GLMs with concatenated functional runs, described in the next paragraph. Only the identified significant voxels were included in the final ROIs for the DCM analysis.

We performed the DCM analysis on participants' native-space, unsmoothed brain images, to maximise the anatomical specificity of the hand-traced hippocampal ROI. We estimated a first-level GLM for each participant in their native space. The GLM model was closely similar to the first-level model defined for the univariate whole-brain analysis (see above). But in this new model, we concatenated all functional runs into a single run to form a single time series per participant. Because we concatenated the runs, we did not model conditions that started <24 s before the end of each run (apart from the very last run), and we did not use the SPM high-pass filtering and temporal autocorrelation options, but as additional regressors of no interest we included sines and cosines of up to three cycles per run to capture low-frequency drifts, and regressors modelling each run.

From each of the four ROIs, we extracted the first eigenvariate of the BOLD signal time-course, adjusted for effects of interest. Based on these data, we estimated and compared a null model with no modulatory connections and 72 models with modulatory connections (73 models in total) to test alternative hypotheses about how suppressing actions and memories modulate connectivity between the four ROIs (see Fig. 7a). All 72 models with modulatory connections were variants of the same basic model with intrinsic bidirectional connections between all regions except no intrinsic connections between M1 and hippocampus, and with driving inputs from the Stop-signal (Stop and Go trials) and Think/No-Think (No-Think and Think trials) tasks into both rDLPFC and rVLPFC regions. Across models, we varied the modulatory influences on the intrinsic connections arising through Stop or No-Think trials.

We grouped the 72 models into three families differing according to whether the source–target modulation was bidirectional, top-down, or bottom-up. Within each family, we defined four subfamilies that differed according to how Stop and No-Think trials modulate the prefrontal control and inhibitory target pathways: independent modulation of target regions by rDLPFC and rVLPFC (testing the idea that two parallel inhibition pathways might exist); rDLPFC only modulation (testing the idea that only rDLPFC supports inhibition); rVLPFC only modulation (testing the idea that only rVLPFC supports inhibition); or modulation of both rDLPFC and rVLPFC (testing the idea that both contribute to inhibition). Within the four subfamilies, we defined further four subfamilies according to how Stop and No-Think trials modulate interactions between the rDLPFC and rVLPFC regions: no interactions; rVLPFC modulates rDLPFC; rDLPFC modulates rVLPFC; or bidirectional interaction between rDLPFC and rVLPFC.

Furthermore, within each subfamily, we defined two additional subfamilies according to whether Stop and No-Think trials modulate the prefrontal connectivity with the preferred targets (M1 when stopping actions and hippocampus when stopping memories) or with the non-preferred targets (hippocampus when stopping actions and M1 when stopping memories), testing the idea that inhibitory modulation must affect a task appropriate structure to model the data well.

We compared the model evidence for the 73 models (the null model and 72 models with modulatory connections) and the groups and subgroups of families across the 24 subjects using random-effects BMS[115,116]. BMS reports the exceedance probability, which is a probability that a given model, or family of models, is more likely than any other model or family tested, given the group data.

*Multi-voxel pattern analysis.* We performed multi-voxel pattern analysis (MVPA) to test whether action and memory inhibition share a common voxel activation pattern within an ROI. We used linear discriminant analysis (LDA) and within-subject classification to classify voxel activity patterns within the same four ROIs that we used for the DCM analysis (rDLPFC, rVLPFC, right hippocampus, and left M1). Same as for the DCM analysis, within each subject-specific ROI, we identified all significant voxels (thresholded at $p < 0.05$, uncorrected for multiple comparisons) for that participant based on the main effect of interest, which included Stop, Go, No-Think, and Think conditions. This selection was performed on the first-level GLMs described in the next paragraph. Only the identified significant voxels were included in the final ROIs for the MVPA analysis. Note that the selected voxels within the basis ROIs might not be exactly the same as used in the DCM analysis as for the DCM analysis we used the first-level model with concatenated functional runs. Nevertheless, the selection was based on the same principles and using the same basis ROIs and does not affect the interpretation of the used regions.

We performed a run-wise classification. For designs like ours (non-blocked event-related, within-subject classification), compared to trial-wise classification, run-wise classification is proven to be a better approach. It provides a clearer class identity and temporal independence and improves the signal-to-noise[117–121]. For each participant on their native-space unsmoothed brain images, we estimated a first-level GLM which was identical to the first-level model defined for the univariate whole-brain analysis (see above). The estimated beta weights of the voxels in each ROI were extracted and pre-whitened to construct noise normalised activity patterns for each event of interest (No-Think, Think, Stop, Go) within each of the eight functional fMRI runs.

To increase the reliability of pattern classification accuracy, we used a random subset approach[122]. Specifically, for each ROI separately, we created up to 2000 unique subsets of randomly drawn 90% of ROI voxels (for smaller ROIs, there were <2000 possible combinations). We then applied the LDA on each subset of voxels and averaged the subset results to obtain the final classification accuracy for each ROI. We performed two types of pattern classification to identify domain-general and domain-specific components within each ROI. Note that with the within-subject classification, the trained classifiers cannot be generalised to other subjects.

For the domain-general component, we performed a within-subject cross-task classification. We trained the LDA classifier to distinguish Inhibit from Respond conditions in one modality (e.g., No-Think from Think) and tested whether the trained classifier could distinguish Inhibit from Respond in the other modality (e.g., Stop from Go). Both training and testing data consisted of two (conditions) by eight (runs) activation estimates for a set of voxels (e.g., $13 \times 16$ matrix for a set of 13 voxels). For training and testing sets separately, for each voxel, we z-scored the activity patterns across the 16 activation estimates setting the mean activity within each voxel to zero. This way, each voxel represented only the relative contribution of Inhibit vs Respond conditions within the Think/No-Think and Stop-signal tasks. For each ROI subset, we performed the LDA twice. The first classifier trained to discriminate Think from No-Think and returned the accuracy of distinguishing Stop from Go; the second classifier trained to discriminate Stop from Go and returned the accuracy of distinguishing Think from No-Think. The final score was the average classification accuracy of all subsets and the two classification variants (up to $2000 \times 2$) per ROI and subject.

For the domain-specific component, we trained and tested the within-subject LDA classifier to distinguish No-Think from Stop conditions. The input data consisted of two (conditions) by eight (runs) activation estimates for a set of voxels. We z-scored the activity patterns across voxels for each event of interest. Thus, the mean ROI activity for each event was zero, and each voxel represented only its relative contribution to the given event. That way, we accounted for the univariate intensity differences between No-Think and Stop conditions. For each ROI subset, we performed leave-one-run out cross-validated LDA and averaged the classification accuracies across the eight cross-validation folds. The final score was the average classification accuracy of all subsets and cross-validation folds (up to $2000 \times 8$) per ROI and subject.

At the group level, for each ROI, we tested the data for normality (for all domains and ROIs $p > 0.05$, confirming normality assumption) and performed one-tailed t-tests to assess the statistical significance of classification accuracy being above the 50% chance level (each classifier was distinguishing between two conditions). All tests were Bonferroni corrected for the number of ROIs, and adjusted p-values ($p_{adj}$) reported.

*A meta-analytic conjunction analysis of Stop-signal and Think/No-Think studies.* To acquire an independent mask of brain areas involved in domain-general inhibitory control, we updated a previously published meta-analysis of Stop-signal and Think/No-Think fMRI studies[22]. The study selection process and included studies are reported in detail in ref. [22]. From the original meta-analysis, we excluded the current dataset[8] and included a different within-subjects (but with each task performed on different days) Stop-signal and Think/No-Think study from our lab[123]. Consequently, our analysis included 40 Stop-signal and 16 Think/No-Think studies. We focused the meta-analysis on the conjunction of Stop > Go and No-Think > Think contrasts which we conducted using Activation Likelihood Estimation (ALE) with GingerALE v3.0.2 (http://www.brainmap.org/ale/[124–127]). We used the same settings as reported before[22]. Specifically, we used a less conservative mask size, a non-additive ALE method, no additional FWHM, and cluster analysis peaks at all extrema. In addition, we set the coordinate space to MNI152.

First, we conducted separate meta-analyses of Stop > Go, No-Think > Think, and their pooled data using cluster-level FWE corrected inference ($p < 0.05$, cluster-forming threshold uncorrected $p < 0.001$, threshold permutations = 1000). We then submitted the obtained thresholded ALE maps from the three individual meta-analyses to a meta-analytic contrast analysis[128], which produced the conjunction of the Stop > Go & No-Think > Think contrasts. We thresholded the conjunction results at voxel-wise uncorrected $p < 0.001$, with the $p$-value permutations of 10,000 iterations, and the minimum cluster volume of 200 mm$^3$.

**Reporting summary**. Further information on research design is available in the Nature Research Reporting Summary linked to this article.

## Data availability

We used a dataset from a previously published study[8]. The behavioural and group-level imaging data have been deposited on the GitHub repository (https://github.com/dcdace/Domain-general/) and are also available on Zenodo (https://doi.org/10.5281/zenodo.5732892)[104]. Raw brain imaging data may be made available via data request at MRC Cognition & Brain Sciences Unit, University of Cambridge (info@mrc-cbu.cam.ac.uks). The raw imaging data cannot be made publicly available due to Research Ethics Board restrictions for the current project. These data can only be shared with researchers working on similar ethically-approved projects and requires managed access. Source data are provided with this paper.

## Code availability

The analysis code and notebooks have been deposited on the GitHub repository (https://github.com/dcdace/Domain-general/) and are also available on Zenodo (https://doi.org/10.5281/zenodo.5732892)[104].

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

## Acknowledgements

This work was supported by the UK Medical Research Council grant MC-A060-5PR00 to M.C.A. and SUAG051/101400. J.B.R. is also supported by the Wellcome Trust (103838) and the NIHR Cambridge Biomedical Research Centre (BRC-1215-20014). The views expressed are those of the authors and not necessarily those of the NIHR or the Department of Health and Social Care.

## Author contributions

C.S.F. and M.C.A. designed and conducted the experiment. D.A. and T.W.S. analysed the data. D.A., C.S.F., T.W.S, J.B.R and M.C.A. contributed to the analysis approach and data interpretation. D.A. and M.C.A. wrote the paper, and C.S.F., T.W.S. and J.B.R. provided critical feedback and approved the final version of the paper.

## Competing interests

The authors declare no competing interests.
