## [Peer Review File · Nature Communications]

Dynamic targeting enables domain-general inhibitory control over action and thought by the prefrontal cortexReviewers' comments:

Reviewer #1 (Remarks to the Author):

This is a reanalysis of a previously described, human fMRI study (Schmitz et al., 2017). The central focus of the current paper is to test whether there is a common prefrontal mechanism(s) that dynamically inhibits different cognitive processes / neural structures. This is tested using a variety of different fMRI analyses applied to two different tasks: a memory stopping task (Think / No Think) and a motor stopping task (Go / No Go). The manuscript lays out several criteria that should be met in order to support the general prediction. The key point is that No Think and No Go trials should both engage the same prefrontal mechanism in a domain general way whereas that target regions (the hippocampus and M1, respectively) should show down regulation in a domain specific way.

The motivating idea for the paper is very interesting and of potentially broad relevance. Several of the analyses are compelling, but are either a conceptual replication or modest advance from prior work. Other findings have substantial logical weaknesses. Or, more specifically, somewhat ambiguous results appear to be interpreted more strongly than is warranted. Ultimately, the main conclusions the reader draws from the paper are (1) that a couple of regions of right prefrontal cortex show some overlap (but not total overlap) for No Think and No Go trials, and (2) the specific prefrontal regions implicated (right VLPFC, right DLPFC) while anatomically distinct, do not seem to show any functional difference. The latter point (the lack of difference between the prefrontal regions) is also a bit unsatisfying because it is hard to fully embrace the null (that these regions are functionally identical).

Comments

1. For the cross-classification (from memory to motor inhibition), the ideal result would be an interaction between the frontal vs. target regions and within domain vs. cross-domain classification. In other words, the relative 'cost' to cross-domain classification should be larger for the target regions than the prefrontal regions. But this is not tested and does not look like it would be statistically supported. Instead, it appears that there is really just a main effect where the prefrontal regions AND the target regions show a clear cost with cross-classification, but because the target regions start out with lower overall classification, they drop to chance whereas the prefrontal regions do not.

2. Relatedly: cross classification works, which is interesting, but there is certainly a cost—even for prefrontal regions. It's hard to know, a priori, how much of a cost is reasonable for a domain general mechanism. Successful transfer is taken as evidence of domain generality, but the cost also can be taken as evidence of domain specificity. The authors acknowledge this briefly, but it feels a bit like a glass half empty vs. half full scenario.

3. Inferentially, it is also a bit strange that the successful cross-classification is taken as evidence for a domain general mechanism, but the successful classification of Stop vs. No Think trials (i.e., evidence that inhibition trials differed, in some way, across tasks) is NOT taken as evidence against a domain general mechanism. This is difficult to reconcile.

4. Continuing on the same theme: the logic is even harder to follow because for M1 and hippocampus, the successful discrimination between Stop vs. No Think trials IS taken as critical evidence for domain specific mechanisms. So, why is successful discrimination of Stop vs. No Think trials in prefrontal regions NOT taken as evidence against a domain general mechanism while it IS taken as evidence against a domain general mechanism for M1 and hippocampus? I thought I might be mis-reading the arguments, but I read this section of the text several times and could not understand the logic. As with several of the other results, this appears to be an example where there is a confirmation bias in how the results are interpreted.

5. There is no positive control—i.e., some other condition, ideally a cognitively taxing condition—that

is contrasted against the motor and memory inhibition effects. This would be useful in several ways. For example, if there was some other cognitively demanding task (maybe mental arithmetic), would right VLPFC and right DLPFC also show connectivity with a brain region involved with this task? The issue is that inhibition (for both motor and memory) is “harder” than Think/Go trials, so we simply don’t know whether or to what degree the prefrontal effects are related to inhibition, per se, as opposed to a much more generic effect of difficulty or effort.

6. Line 498: the connectivity models overwhelmingly favored a bidirectional relationship, but it is the top-down component of this relationship that the authors focus on. This seems like a selective emphasis. The bidirectional model means that there is no evidence that the top-down influence of prefrontal regions on target regions is any greater than the top-down influence of target regions on prefrontal cortex. I could easily imagine this very result being interpreted as evidence AGAINST a top-down relationship between prefrontal and target regions; instead, it is taken as clear evidence for a top-down influence of prefrontal on target regions. Again, the focus goes on the aspect of the results that is theory consistent.

7. The conclusion that right VLPFC and right DLPFC work together—and, effectively, are indistinguishable—is a bit unsatisfying. It does not seem likely that these regions are redundant. Obviously, the current study did not tease apart the contributions of these regions, but perhaps this just reflects a weakness of the current methods—e.g., DCM is perhaps not as sensitive as true causal methods (stimulation). I am not arguing that there is any evidence here for a difference between these regions, but there is only so much that can be inferred from a null result like this. To be fair, the functional equivalence of these regions is not the main take home from the paper, but given that the goal is to reconcile thought suppression and motor stopping literatures—and these literatures have tended to highlight different regions (right DLPFC and right VLPFC, respectively)—it seems a bit too tidy to conclude that, in fact, both regions contribute to both kinds of inhibition in exactly the same way.

8. Line 412 and 413, there are two p values that are listed as “p = 1”—this seems to be some kind of error (the p values should not be exactly 1).

Reviewer #2 (Remarks to the Author):

In their manuscript “Dynamic targeting enables domain-general inhibitory control over action and thought by the prefrontal cortex”, Apsvalka and colleagues present data from an fMRI investigation in N=24 that aimed to test whether domain-general regions in (pre)frontal cortex were conjointly involved in the inhibition of action (operationalized in the stop-signal task) and associative memory (operationalized in the Think/NoThink paradigm). The authors report that indeed, two regions in the prefrontal cortex – specifically, in the right ventro- and dorso-lateral prefrontal cortices – are involved in inhibiting both representations. Moreover, the authors find that these regions work together in specifically targeting the downstream neural targets of the purported inhibitory process in either domain (M1 for action, hippocampus for associative memory). The empirical demonstration is exhaustive, and the question is explored in several complementary ways that result in a coherent picture.

Overall, this is a phenomenal paper. The research presented is very impressive and the authors went through great lengths to truly explore the underlying question from multiple complementary angles. Indeed, I would describe this work as the empirical pinnacle-to-date of a research idea that has received a lot of interest in the recent past (domain-generality of inhibitory control beyond motoric representations), with many high-quality papers already published by this group and others. The careful theoretical justification of the criteria used to evaluate domain-generality, alongside the affirmative empirical demonstration thereof in the data itself make for a highly compelling package. The manuscript is also extraordinarily well-written. I have very little to offer in the way of criticism, and indeed nothing that would be considered major. A few suggestion, largely regarding the theoretical

framing, are listed in the following.

1. In regards to the motivation of the hypothesis of a possibly domain-general inhibitory control system, I have always liked that fact that Logan and colleagues had already suggested this when they described the original horse race model of the stop-signal paradigm (indeed, one of the earliest papers was titled “On the ability to inhibit thought and action”). I personally often like to acknowledge this amazing foresight by Logan and colleagues, but I leave it up to the authors of the current manuscript if they want to incorporate this into the framing and motivation of the study.
2. In a similar vein – and with awareness that it may seem gauche to bring up our own research – I think that two of our papers were, if not the first, then at least two of the more influential neuroscientific papers to empirically demonstrate (Wessel et al., Nature Communications 2016) as well as theoretically and neuroanatomically embed (Wessel & Aron, Neuron 2017) the proposal that the neural system underlying action stopping may also underlie memory suppression. Despite my obvious bias, I think it would be fair to ask the authors to perhaps cite and discuss those papers. They may also consider our recent work expanding this picture to attentional representations (Soh & Wessel, Cerebral Cortex, in press, PMID: 33140100), though I would not deem this essential, as it is not strictly about memory.
3. Finally, since the vIPFC has been found to be a key cog in the neural network identified here, Corbetta and Shulman’s circuit breaker proposal may warrant some discussion in this paper as well.
4. I very much liked the inclusion of the BEESTS model, especially since I noticed that the traditional SSRT estimate was quite unusually large. I would like some more detail on this analysis, specifically with regards to the quality of the model fit (BEESTS needs a substantial amount of trials per subject to work properly). Moreover, I’d be curious if the trigger failure parameter correlated with any of the variables of interest.

Other than that, I have no substantial points of concern, and I congratulate the authors on this beautiful piece of work.

Signed,
Jan Wessel

Reviewer #3 (Remarks to the Author):

Manuscript in-review

“Dynamic targeting enables domain-general inhibitory control over action and thought by the prefrontal cortex”. Apšvalka et al., 2020. Nature Communications. NCOMMS-20-44422.

Summary & merits

In the present study by Apšvalka et al., neural mechanisms of inhibitory control are assessed via an impressively multi-pronged study. This included univariate, multivariate, meta-analytic, and causal models. The authors identified two prefrontal source regions in the right hemisphere (the vIPFC and dIPFC) that dynamically modulated activity in context-specific target regions (hippocampus and M1; specific to thought and action suppression, respectively) in support of inhibitory control. To this end, the authors provided a distinct and testable set of criteria that must be met (the ‘five attributes of dynamic targeting’) to successfully identify a domain-general inhibitory system. This is specifically useful for future work on inhibitory control, and also generally provides an excellent template for future work seeking to characterize complex neural processes. Further, the explication of domain-general sources and domain-specific targets (both being outlined as essential to the dynamic targeting process) is clarifying and useful for theory building on flexible neural processes. As pointed out in the Discussion, inhibitory control is central to multiple mental and neurological disorders, positioning associated neural processes as vital to clinical application. Moreover, the reported differences between inhibitory control processes and general cognitive control processes deepens our understanding of the important topic of cognitive control, as well as opening up the possibility of characterizing other specific instantiations of cognitive control with similar analytic rigor. For example,

given that a subset of cingulo-opercular network (CON) regions (and to a lesser extent herein, frontoparietal network or FPN regions), which are implicated as a whole in general cognitive control, appear to be specifically important for inhibitory control, one may ask if the other CON (and/or FPN) regions similarly have a specialized control function. Major and minor considerations are addressed below.

Major considerations

1. The expansion of this study into the causal modeling domain using DCM is helpful for more explicitly testing the proposed domain-general inhibitory control hypothesis. The analysis appears to be well done for a DCM analysis. However, there are known limitations of DCM, especially with the particular use of that framework here. Most importantly, there's the clear possibility of causal confounds, given the hundreds of brain regions not included in the model. For instance, it is possible that some other regions actually implement inhibitory control, simply updating the PFC regions about the control being implemented (i.e., acting as a confounder to PFC and domain-specific regions). This should at least be mentioned as a limitation in the Discussion. Ideally this would be addressed by including all other brain regions in the DCM model (perhaps the "winning" model #12 only) somehow, perhaps using regression-based DCM (see Frässle, Stefan, et al. "Regression DCM for fMRI." *Neuroimage* 155 (2017): 406-421.). Another limitation that should be discussed and/or addressed is the choice of inputs into the DCM models: Clearly the actual inputs of the task activity into the brain are visual and auditory regions, not PFC regions. It is unclear whether the results are biased based on the choice of PFC regions as inputs, especially for the inference that the connections include directional links from the PFC inputs to the domain-specific target regions.

2. The focus on right dlPFC and vlPFC as the source ROIs is not fully clear. While it is clear why left M1 and right hippocampus were chosen as target ROIs (a priori based on prior findings from the same group), it appears that there were more candidates for source ROIs. Tables 1-2 and Figures 3-4 (and supplementary results), as well as associated text in Results and Methods, list other candidate regions (e.g., parietal regions) based on the conjunction analyses, but ultimately right prefrontal regions were chosen for subsequent analyses. Given the focus of prior literature on inhibitory control enacted by the PFC this is a reasonable choice, but this might represent a bias in the literature toward PFC regions when other regions may be just as (or more) important for implementing inhibitory control. It would be important to see (perhaps as a follow-up analysis) what the main results would look like if all candidate source regions are included, even to potentially act as controls (i.e., if the other candidate sources did not show effects that the PFC sources did). If there is a strong reason for excluding them (perhaps the dimensionality of including so many regions in this study is unreasonably large/difficult to interpret), then this should be justified/explained in the Methods and pointed out as a limitation in the Discussion.

3. The supplement needs a bit of work. Firstly, the methods that went into supplementary results are not explained fully; with a noticeable gap in how the conjunction maps and Glasser et al (2016) MMP atlas/Ji et al (2019) CAB-NP partition were overlaid. Secondly, the Figures require summary text with more specificity. Thirdly, the conflict reduction results (S4-S5) might be more suitably placed in the main Results section (it's not immediately clear why it's in the supplement).

4. It appears that the Results sub-sections generally follow the order of attributes listed in Figure 1 (which provides an incredibly useful and clear-cut framework for the study). However, I was left wondering why the Results sub-section titles didn't match those five attributes? There are clearly multiple analyses for some of the attributes/criteria, but it might offer full clarity to have those as umbrella titles (for example: 'Domain generality of inhibitory sources', 'Behavioral relevance of inhibitory sources', and so on) and the specific analyses as secondary sub-sections/titles (if allowed by the formatting specifications of the journal). This would map the results directly onto the framework (in order) provided by Fig. 1.

5. Another consideration for the Discussion section (which would go a long way toward building an intuition about inhibitory control mechanisms) is to expand upon other domains outside of thought and action. A meta-analysis was included in defining the source ROIs - why was this restricted to studies

that similarly probed think/no-think and go/stop? It might be helpful to either explicitly mention this as a limitation, or to include another meta-analysis (or an extension to the current meta-analysis) that includes studies of inhibitory control in other domains. More concretely: thought/memory and action/motor domains may specifically differ (in terms of identified source ROIs, as well as how their downstream impact on targets is characterized) from other well-studied domains such as visual/auditory representation, emotional, etc. The Stroop and Flanker tasks also come to mind as prominent assessments of inhibition.

Minor considerations

1. The 2 measures of brain score and voxel salience could be clarified further. It would be helpful to provide more detail on the salience measure specifically, as it appears to contain some circularity in its estimation (which may be just confusion over phrasing): in the Results, salience is described as: "...correlation between brain scores and behavioral measurements identify the direction and the strength of the relationship captured by a LV..."; and in the Methods as "...weighted pattern across brain voxels representing the strength of the relationship between the BOLD signal and the behavioral scores." In the Results definition, 'brain score' presumably refers to per-subject variance explained (by the LV) mentioned in the Methods, which implies that correlating those brain scores back with the same behavioral measurements they were based upon (i.e., how the LV is generated in this application of PLS) is how salience is estimated. If this is a mis-reading, then please clarify. Further, the terms brain score/voxel salience are used in the Results text and variance explained/brain salience in the Methods text; using consistent terms across manuscript sections (and related figures; such as the y-axis in Fig. 4b) might help clarify these complex measures.

2. On page 16, MVPA similarities are described as an indicator of dynamic targeting. It is unclear to me if similarities refers to similarity in decoding prediction accuracy, or similarities in the patterns themselves? Relatedly, it appears that dlPFC and vlPFC exhibit high domain-specific prediction accuracies (higher than domain-general component; Fig. 5c). How can we interpret this given the proposal that those source regions function in a primarily domain-general fashion? (potentially a point for the Discussion).

3. Why were the Results on pg. 33 (related to Fig. 4) masked to the meta-analytic conjunction ROIs (i.e., Table 1b)? What is the rationale for using those regions as opposed to the ROIs in Table 1a (ROIs from the current empirical dataset) or 1c (empirical plus the meta-analysis ROIs)?

4. Were the subject-specific ROIs (and voxel thresholding) used in the DCM analyses also used in other analyses (involving those same ROIs; for example those in Fig. 5)? If not, text on how one can compare (and interpret) the findings is warranted (e.g., synthesizing results between Fig. 5 and Fig. 6). Note that this may generally be warranted (i.e., details on how, for each type of analysis including univariate, multivariate, and connectivity, ROI definitions are different/the same and what that means for interpreting results) given the text in the Methods that describes alterations to the GLM in the DCM vs. the univariate analyses (i.e., that the data was processed differently).

5. The outlining of domain-general sources and domain-specific targets in Figure 1 is quite clear. However, the text (e.g., pages 2-3 of the Introduction) describing the 5 attributes does not explicitly delineate this. It would be beneficial for reader comprehension to explain this in-text, perhaps as an initial sentence or two in that section, to give context to the attributes that are about to be outlined.

6. The language used on pages 13-14, lines 348-350, could be more precise in terms of the order of processes being proposed (i.e., one piece of evidence used to build up to the causal inference). For example, it might be more precise to say something akin to: "...downregulated hippocampal activity inhibits memory retrieval." (i.e., indicating that cognition/behavior is at the end of the causal process). This is a subtle difference, but important given the typically higher standards/scrutiny for causal inference. To fully explicate the process would be something akin to: "...inhibitory control mechanisms enacted by the source ROIs downregulates activity in the hippocampus which inhibits memory retrieval."

7. The assertion on page 5 (lines 155-157) is not immediately obvious to me: "If the capacities to stop actions and thoughts are related, more efficient action stopping, as measured by stop-signal reaction time, should correlate with greater suppression-induced forgetting." While this is certainly one option (that is supported by the results), it could also be the case that downstream, at the level of domain-specific processing, different target regions exhibit different levels of 'efficiency'; and moreover, this could vary by subject. That is to say, even with comparable exhibition of domain-general inhibitory control by the rIPFC regions herein, why would one predict that the downstream processes are strongly linked to each other, given that they (the targets) presumably have their own local processing and mechanisms to enact the task at hand?
8. The interaction effect of the rm-ANOVA is reported (on pg. 14), what about the main effects? T-test results are reported, which likely more directly test the effects of interest; some explanation (in the Methods) of why t-stats are reported instead of main effects (f) would be useful.
9. In the Methods (pg. 30), it's reported that select functional runs of select participants were excluded due to technical issues. How did this impact the conflict reduction results in the Supplement? Was there a different amount of data (i.e., more or less subjects) used for different runs here?
10. In the Methods (pg. 33), it's mentioned that six nuisance parameters were used in the task GLM; were these all related to head motion? Did you account for physiological noise (such as white matter and ventricle timeseries)?
11. Consider swapping the presentation order of Fig. 3 and Fig. 4 given that Fig. 4a results are reported in-text first.
12. The word 'transcending' is used to describe activation patterns at least twice; consider substituting it with a more quantitatively precise term.
13. What do the numbers in Tables 1 and 2 (left most column) indicate? Is this a ranking?

Reviewer #4 (Remarks to the Author):

Thank you for the opportunity to review this manuscript. I found the topic very interesting and the overall ideas to be presented very clearly. I think the paper has the potential to make important contributions to the field.

I especially appreciated the author's willingness to lay out multiple steps for testing their hypotheses, and willingness to proactively address potential issues with data/additional tests. The analyses multi-method approach offers convergence across multiple analyses, which I see as a major strength. Below I offer some suggestions that I hope will be helpful for improving the manuscript, particularly with respect to the classification method – which is, in my view, the biggest weakness of the paper at present but can also be straightforwardly addressed.

Can the authors provide more details on the classifier? For example, how is chance 50% when the trials Think/No Think and No/NoGo were not balanced for data collection trials? The methods indicate that a random subset was drawn, but it's not clear how class imbalance was dealt with. Typically in a classification task – or when using machine learning principles - the class distribution in the training data can be changed (upsampled/downsampled, etc.), but only in the training set. It is typically not appropriate to change the distribution of the test set.

Critically, the authors say leave one out cross-validation. This is possibly great, but does it refer to leave one instance out or leave one person out? Leave one instance out offers the possibility of overfitting because of person-dependent instances in the training set. 8-folds are mentioned but it's unclear how this maps on to leave one out, or perhaps I missing what is "left out." This is important to

address given the well-known issues with overfitting in such models.

I also encourage the authors to consider the practical significance of their classifier accuracy a bit more. For example, a kappa of .07 (7% increase in classification accuracy; 57% accuracy) above chance would not be considered strong performance in other fields. In short, it is meaningful and is different from chance, but it's not doing that well prediction-wise in some cases.

Can the authors provide a bit more review of thought suppression and related regions? In general, the paper is clearly motivated. At the same time, it would be helpful to more explicitly contrast (up front) how previous studies have found specific inhibitory mechanisms (rDLPFC vs rVLPFC) and how this proposal incorporates and extends those findings in a domain general framework.

This is not a critique of the current paper, but I am curious if the authors' view this as domain general process is because people are inherently meta-aware/conscious of their stopping in both cases? So, is the ability to initiate stopping (thought/motor) general necessarily involve that awareness, or would stopping behaviors with a lack of awareness have the same patterns? Again, this is not a concern for the contribution of the current paper, but perhaps could be addressed in the discussion section.

How might the dynamic targeting work for suppression in people with hippocampal damage? This is not a critique, but I'm wondering if the authors may offer some caveats to the domain general process proposed here. Are there exceptions to the dynamic targeting mechanisms?

The figures were excellent; thanks to the authors for their thoughtfulness in graphical representation.

**Response to reviews:
NCOMMS-20-44422**

Reviewer Comment 3.1.

Summary & merits

In the present study by Apšvalka et al., neural mechanisms of inhibitory control are assessed via an impressively multi-pronged study. This included univariate, multivariate, meta-analytic, and causal models. The authors identified two prefrontal source regions in the right hemisphere (the vIPFC and dIPFC) that dynamically modulated activity in context-specific target regions (hippocampus and M1; specific to thought and action suppression, respectively) in support of inhibitory control. To this end, the authors provided a distinct and testable set of criteria that must be met (the ‘five attributes of dynamic targeting’) to successfully identify a domain-general inhibitory system. This is specifically useful for future work on inhibitory control, and also generally provides an excellent template for future work seeking to characterize complex neural processes. Further, the explication of domain-general sources and domain-specific targets (both being outlined as essential to the dynamic targeting process) is clarifying and useful for theory building on flexible neural processes.

Author Reply 3.1.

We are grateful for the reviewer’s positive remarks about the work. We agree that the criteria we lay out for inhibitory control ought to be of general use beyond our particular study, and we hope that this can provide a useful template for other efforts to characterize complex cognitive processes. We believe that the dynamic targeting concept will focus research on inhibitory control not merely on the prefrontal cortex, but on necessary interactions with targeted regions, and on the issue of how domain generality is achieved, which has been neglected.

Reviewer Comment 3.2.

As pointed out in the Discussion, inhibitory control is central to multiple mental and neurological disorders, positioning associated neural processes as vital to clinical application.

Author Reply 3.2.

We agree. Indeed, apart from the basic theoretical issues addressed here, we see this work very much as building on one of the core mechanisms included in the NIMH’s RDOC framework, a framework specifically designed to link basic neuroscience to psychiatric conditions. Perseverative cognition is pervasive and inhibitory control is critical to this symptom. In neurological disorders such as Parkinson’s disease and Parkinson-plus disorders, methods derived from cognitive neuroscience to characterise inhibitory control in terms of connectivity of the prefrontal cortex have already led to experimental medicines studies (e.g., from our own work PMIDs 27343257, 27282105, 26837463, 26757216, 24655598) and clinical trials (eg. ISRCTN99462035). So, we are very optimistic about the clinical utility of this work.

Reviewer Comment 3.3.

Moreover, the reported differences between inhibitory control processes and general cognitive control processes deepens our understanding of the important topic of cognitive control, as well as opening up the possibility of characterizing other specific instantiations of cognitive control with similar analytic rigor. For example, given that a subset of cingulo-opercular network (CON) regions (and to a lesser extent herein, frontoparietal network or FPN regions), which are implicated as a whole in general cognitive control, appear to be specifically important for inhibitory control, one may ask if the other CON (and/or FPN) regions similarly have a specialized control function. Major and minor considerations are addressed below.

Author Reply 3.3

We thank the reviewer for these insightful remarks. We agree that the current work contributes to a deeper understanding of the distinct contributions of the elements of large-scale networks like the CON and FPN. One of the more exciting aspects of this project, in general, has been the opportunity to link research on inhibitory control to work on large scale networks, and also to other forms of cognitive control. It is intriguing that, at least in this study, inhibitory control seems to represent a distinct configuration from what is typically observed in other forms of cognitive control. This certainly cries out for further work.

Reviewer Comment 3.4

Major considerations

1. The expansion of this study into the causal modeling domain using DCM is helpful for more explicitly testing the proposed domain-general inhibitory control hypothesis. The analysis appears to be well done for a DCM analysis. However, there are known limitations of DCM, especially with the particular use of that framework here. Most importantly, there's the clear possibility of causal confounds, given the hundreds of brain regions not included in the model. For instance, it is possible that some other regions actually implement inhibitory control, simply updating the PFC regions about the control being implemented (i.e., acting as a confounder to PFC and domain-specific regions). This should at least be mentioned as a limitation in the Discussion. Ideally this would be addressed by including all other brain regions in the DCM model (perhaps the "winning" model #12 only) somehow, perhaps using regression-based DCM (see Frässle, Stefan, et al. "Regression DCM for fMRI." *Neuroimage* 155 (2017): 406-421.).

Author Reply 3.4

The reviewer's comment raises two distinct issues:

1. *Causal necessity of the right DLPFC /VLPFC.* Can we conclude, based on DCM, that these regions are causally involved in stopping memories and thoughts? Could some other region be the true source of inhibition, with right DLPFC and VLPFC only receiving updating information from the actual inhibition source?
2. *Complexity of inhibitory control.* Many regions involved in stopping were not entered into the model; should these be modelled to fully understand the mechanisms involved?

The issue of “confounders” is a manifestation of the “third variable” problem in correlational research and is not unique to DCM modeling. In principle, where X and Y are related, X might cause Y or Y might cause X, or a third variable Z could cause both X and Y. So, could a region Z simultaneously activate right DLPFC and inhibit the hippocampus (or M1), leaving the DLPFC epiphenomenal?

There are several reasons why this is unlikely to explain our results.

1. *Causal Status of Lateral PFC regions is likely on a priori grounds.* Right DLPFC and VLPFC are unlikely to be confounders. For example, there is causal evidence for the role of right VLPFC and right DLPFC in both motor and memory stopping: (a) Lesion volume to the right VLPFC and DLPFC significantly correlate with stop-signal reaction time (Aron et al., 2003); (b) in monkeys, intracranial electrical stimulation of right DLPFC and VLPFC homologues during a “Go trial” induces a monkey to stop its action (Sasaki et al., 1989; for analogous results in humans, see, Wessel et al. 2013); and TMS to the right VLPFC disrupts stop signal reaction time (e.g., Chambers et al., 2006). We also have found that with retrieval stopping, right lateral PFC lesions, but not left lateral prefrontal regions abolish people’s ability to suppress memory retrieval, as reflected in a lack of suppression-induced forgetting (Anderson et al., 2019). I have pasted the figure from this study below, which included 50 frontal patients (25 with left lesions, 25 with right).

Of course, these observations do not imply that OTHER regions don’t also play a causal role. But they offer reassurance that our DCM findings are not misleading about the causal status of LPFC regions. Indeed, in this paper, we do not seek to establish the causal necessity of these regions in general, which is *already done*; but rather to (a) establish that the very same regions that are causally involved in action stopping are causally involved in thought stopping, and that dynamic targeting of domain-specific regions occurs.

2. DCM is a way of testing *a priori* hypotheses about the relationship between regions, that are chosen (ideally) based on both prior functional and anatomical work. Thus, traditional DCM is not an exploratory method into which all possible brain regions are blindly placed, but only those of direct theoretical interest. As such, it would be inappropriate to include all regions, or even all regions arising from our meta-analysis.

Instead, we focused on right DLPFC and VLPFC because of the already persuasive evidence that these regions are sources of control. The DCM then tests alternate hypotheses

about the potential causal interactions between regions (here, DLPFC, VLPFC, M1, Hippocampus), in the two tasks. Our analysis indicates a homologous causal role of the PFC regions in stopping actions and in stopping thoughts, and that task-specific stopping is achieved by dynamic targeting of specific regions.

We agree of course that other brain regions are involved in inhibitory control, as our meta-analysis clearly indicates. The functions of these other regions and their causality is not shown. There is strong evidence however, that right DLPFC and VLPFC are causally necessary, and that is the main focus of our tests and claims.

A more complex connectivity analysis, for example with all other brain regions, is becoming feasible with methods such as regression-DCM and sparse rDCM. This would be an advantage if (i) seeking a connectomics characterisation of a disorder, based on effective connectivity rather than the functional connectivity that is the basis of current fMRI connectomics; or (ii) if one lacks clear *a priori* knowledge or assumptions about a network's structure. However, the focus of the present paper is on domain-generality, and dynamic targeting, and the role of different prefrontal cortical structures in achieving that. For this purpose, the DCM model space is sufficient. So, we hope that the reviewer will understand that we have not undertaken the analysis suggested for this particular paper.

Author Action Taken 3.4

We agree with the reviewer that the issue of causality is worth discussing. Readers will appreciate and benefit from an explicit discussion of this point. To address this, we added a new discussion paragraph that (among other things) explicitly states our claim that we believe these regions to be causally necessary for inhibitory control based on methodologically diverse sources of evidence. We do not explicitly discuss the issue of confounders as this seemed a bit discursive when the multiple forms of evidence are considered at once. Readers can then decide whether they agree with our assessment. This paragraph also provides an opportunity to address the issue of task difficulty raised by reviewer 1 and to frankly state that our discussion is limited to the PFC. We thank the reviewer for this suggestion, which allows us to address a potential reader concern, and improve the paper.

The new paragraph is the third one in the Discussion section and is repeated below for the reviewer's convenience.

"The proposed role of the rDLPFC and rVLPFC as hubs of domain-general inhibitory control during stopping does not imply that these regions are exclusively dedicated to stopping. Indeed, it seems likely that these regions contribute to many cognitive functions. Rather, the current evidence suggests that when stopping an action or thought is required, these regions are recruited to cancel processing in target areas involved in representing to-be-suppressed content, thereby achieving the desired stopping outcome. Methodologically diverse evidence indicates that this contribution is causally necessary to successful inhibitory control and is not a mere epiphenomenon of doing difficult tasks. First, the current effective connectivity analyses indicate a robust top-down modulation of target regions by these putative prefrontal sources. This finding comports well with lesion and brain stimulation work in both humans and animals indicating that disrupting the function of rDLPFC and rVLPFC severely disrupts the capacity to stop, consistent with causal necessity (Aron et al., 2003; Chambers et al., 2006; Sasaki et al., 1989; Wessel et al., 2013). Second, although action and thought stopping are both difficult tasks, the network dynamically reconfigured its connectivity to target regions to suppress their function in a manner compatible with task goals. These features have the

hallmarks of a control process configured to implement a particular regulatory function, rather than a generic response to task difficulty. RDLPFC and rVLPFC are likely to work in concert with a broader network to achieve these goals, as our conjunction analyses suggest. The current work does not address the functional role of domain-general regions outside of the prefrontal cortex, the contributions of which should be examined in future work.”

Reviewer Comment 3.5

Another limitation that should be discussed and/or addressed is the choice of inputs into the DCM models: Clearly the actual inputs of the task activity into the brain are visual and auditory regions, not PFC regions. It is unclear whether the results are biased based on the choice of PFC regions as inputs, especially for the inference that the connections include directional links from the PFC inputs to the domain-specific target regions.

Author Reply 3.5 A tenet of model based analyses is that the models should be sufficient to test the hypotheses at hand, and not more complex than necessary. The connections within a DCM model are not claimed to be mono-synaptic, direct, or singular, but rather indicative of the net causal interaction between source and target region. This aspect of the modelling is made clear in the revised manuscript, as in the original methodological papers introducing DCM.

A corollary is that the models do not need to include regions up-stream or down-stream of that part of the whole brain network that is relevant to the hypotheses. Regarding sensory input, one could if it were relevant to one’s hypotheses, include visual cortex, or lateral geniculate, or even retinal signals as specified regions. But this is simply not necessary here to specify each upstream sensory region, and to do so would potentially undermine the analysis by adding unnecessary complexity. Recall that all of the function and activity of these up-stream regions is already included in the model – but included in the form of the C matrix (driving inputs).

The reviewer asks whether the results are “biased based on the choice of PFC regions as inputs”. The results are of course constrained by the range of possibilities encompassed by the models, but they are not statistically biased given that the regions of LPFC are not chosen based on their selective response to driving inputs.

Author Action Taken 3.5 In revising the manuscript, we have included the caveats that the activity and connectivities of the sensory input regions are modelled collectively in terms of the driving inputs; and the connections within the DMCs are not restricted to mono-synaptic or singular direct paths.

Reviewer Comment 3.6

2. The focus on right dlPFC and vlPFC as the source ROIs is not fully clear. While it is clear why left M1 and right hippocampus were chosen as target ROIs (a priori based on prior findings from the same group), it appears that there were more candidates for source ROIs. Tables 1-2 and Figures 3-4 (and supplementary results), as well as associated text in Results and Methods, list other candidate regions (e.g., parietal regions) based on the conjunction analyses, but ultimately right prefrontal regions were chosen for subsequent analyses. Given the focus of prior literature on inhibitory control enacted by the PFC this is a reasonable choice, but this might represent a bias in the literature toward PFC regions when other regions may be just as (or more) important for implementing inhibitory control.

It would be important to see (perhaps as a follow-up analysis) what the main results would look like if all candidate source regions are included, even to potentially act as controls (i.e., if the other candidate sources did not show effects that the PFC sources did). If there is a strong reason for excluding them (perhaps the dimensionality of including so many regions in this study is unreasonably large/difficult to interpret), then this should be justified/explained in the Methods and pointed out as a limitation in the Discussion.

Author Reply 3.6

We do not claim exclusivity of DLPFC and VLPFC as sources of control, nor does the demonstration of the principle of dynamic targeting require or assume such exclusivity.

If one entered other nodes of the network into the model, and they behaved similarly to the RDLPFC and RVLVLPFC in exerting a top-down modulation, it would not alter the main conclusion: that there exist domain-general regions that are causally involved in top-down inhibitory control over domain-specific target regions, dynamically targeting them depending on the task goal.

Our focus on the right DLPFC and VLPFC reflects the abundant evidence pointing to the necessary causal role of these structures in action stopping. As noted in Author Reply 3.4, multiple lines of evidence from rodents, monkeys and humans using electrophysiology, electrical and magnetic stimulation, cooling and muscimol inactivation, lesions, and imaging converge on the causal role of these structures, and this is widely agreed in the field. Moreover, details of the anatomical pathways by which the lateral prefrontal cortex enacts inhibitory control, via striatum and STN are increasingly well established. Converging with action stopping work, as noted in 3.4, we now have causal evidence for the role of right lateral prefrontal cortex in retrieval suppression. So, the case in favour of these regions being causal sources is robust and motivates our focus. Having such an a priori theoretically motivated focus is a strong scientific approach that complements empirically driven explorations of connectivity and network discovery.

As noted earlier, we do not doubt the activity of other regions during inhibitory control, or a causal contribution of some of them to inhibitory control. Indeed, lesions to some of these regions may well impair inhibitory control as well, indicating an important causal role, although there is little evidence for this at present. We do not exclude this possibility in any of our claims—indeed, in our discussions of large-scale networks, our intention was to explicitly invite attention to this possibility. However, we are confident that the idea that ONLY these non-frontal areas are important is incorrect, as clearly indicated by the aforementioned causal evidence.

Author Action Taken 3.6.

The reviewer's questions about our ROI choices suggests that we did not do a good enough job at highlighting our specific reasons for the focus on right DLPFC and VLPFC and not the other regions in our meta-analysis. If the reviewer had this reaction, chances are that other readers might as well, suggesting it would be wise to improve this aspect of our paper. To address this concern, we have gone through the paper and have sought to highlight that we adopt this focus because of prior causal evidence, but to also explicitly acknowledge the potential role of other elements of the network in achieving inhibition. Improvements can be found in the following places:

1. Paragraph 2 of Intro now explicitly calls attention to causal evidence for PFC contribution:

“Disrupting rVLPFC impairs motor inhibition, whether via lesions (Aron et al., 2003), transcranial magnetic stimulation (Chambers et al., 2006), intracranial simulation in humans (Wessel et al., 2013) or monkeys (Sasaki et al., 1989), **establishing its causal role in stopping.**”

2. Paragraph 3, introduction, first sentence now overtly links our focus on PFC to prior evidence of causality alluded to in prior paragraph:

“If these rLPFC regions play a **causal role** in how domain-general inhibitory control achieves stopping, the question arises as to how inhibition is directed at actions or thoughts.”

3. Paragraph 3, Results Section on the initial conjunction analysis (page 10) now explicitly states that the broader network is not the focus of the current investigation:

“The domain-general stopping activations included areas outside of the prefrontal cortex (see **Error! Reference source not found.**a and **Error! Reference source not found.**). **Although not the focus on the current investigation**, we characterised these activations in relation to large-scale brain networks, using a publicly available Cole-Anticevic brain-wide network partition (CAB-NP) (Ji et al., 2019).”

4. Final paragraph in the section on the initial conjunction analysis (page 11) now explicitly links our focus on PFC to prior causal evidence:

“**Together, these findings confirm the role of both the right anterior DLPFC and rVLPFC for both motor and memory inhibition, consistent with prior evidence of a causal role of these regions in the stopping function of inhibitory control** (Aron et al., 2003; Chambers et al., 2006; Sasaki et al., 1989; Wessel et al., 2013).”

We agree with the reviewer that not examining the other elements of the network is a limitation of our a priori focus on the PFC, even if we feel that that focus is well justified. We therefore explicitly state this limitation of the current work at the end of paragraph 3 in the discussion.

“**RDLPFC and rVLPFC are likely to work in concert with a broader network to achieve these goals, as our conjunction analyses suggest. The current work does not address the functional role of domain-general regions outside of the prefrontal cortex, the contributions of which should be examined in future work.**”

Reviewer Comment 3.7.

3. The supplement needs a bit of work. Firstly, the methods that went into supplementary results are not explained fully; with a noticeable gap in how the conjunction maps and Glasser et al (2016) MMP atlas/Ji et al (2019) CAB-NP partition were overlaid. Secondly, the Figures require summary text with more specificity. Thirdly, the conflict reduction results (S4-S5) might be more suitably placed in the main Results section (it’s not immediately clear why it’s in the supplement).

Author Reply 3.7

We thank the reviewer for these helpful suggestions for improvements.

Author Action Taken 3.7

We did the following to address these points.

1. Expanded the Methods description of how the conjunction maps were overlaid onto the Glasser CAB-NP partitions.

At the end of the first paragraph of the Supplementary document we added these three sentences:

“Specifically, we mapped our volumetric activation clusters onto group average of 1200 subjects’ surface template provided by the Connectome Workbench. We used ‘-volume-to-surface-mapping’ command with trilinear volume interpolation. We then overlaid the resulting shape files of our activation results onto the CAB-NP and obtained the corresponding parcel and network locations.”

2. Added greater specificity to the Figure summary, as requested.

We shortened the second paragraph of the Supplementary document and incorporated parts of its original text into Table and Figure captions.

3. Moved Figures S4 and S5 from the supplements to the Main Results (now Figure 6), as Requested. We had been torn about this initially but put these in the supplements mainly because we already had so many figures. Your suggestion gives us the excuse to do what we wanted to do anyway! Thanks.

Reviewer Comment 3.8.

4. It appears that the Results sub-sections generally follow the order of attributes listed in Figure 1 (which provides an incredibly useful and clear-cut framework for the study). However, I was left wondering why the Results sub-section titles didn’t match those five attributes? There are clearly multiple analyses for some of the attributes/criteria, but it might offer full clarity to have those as umbrella titles (for example: ‘Domain generality of inhibitory sources’, ‘Behavioral relevance of inhibitory sources’, and so on) and the specific analyses as secondary sub-sections/titles (if allowed by the formatting specifications of the journal). This would map the results directly onto the framework (in order) provided by Fig. 1.

Author Reply 3.8

We agree that this would be an elegant way of organising the results, and we indeed considered it. However, not every section corresponds to an attribute while some attributes are supported by evidence in multiple sections that differ substantially in methods (e.g., behavioural relevance, which is supported by both PLS and MVPA). Although we could use a mixture of headings pointing to properties, and others summarising the claim of the section, this is likely to be more confusing for readers. Nevertheless, in revising the manuscript, we have adopted a solution that achieves the spirit of the reviewer’s suggestion without requiring complete reorganisation of the writing (see below)

Author Action Taken 3.8

The current section headers have the nice quality that they succinctly encapsulate the main finding of the section, allowing readers to scan the headings and get the gist of the argument. For these reasons, we did not change the headers.

Instead, to more strongly link the attributes to the focus of each section, we now include an explicit callout early on in each section, stating which attribute the section focuses on. In most cases this can be found in the first sentence or two, with an attribute number and title. There are also several cases where such callouts can be found further into the section, because additional attributes may arise within a section (e.g., in the section on DCM, both attributes 4 and 5 occur).

Here are some examples of the approach, each of which is the first sentence of their respective sections:

“A key attribute of dynamic targeting is that the domain-specific target areas are inhibited in response to activity of the domain-general source of inhibitory control, when the specific task goals require it (Attribute 3: suppression of function in target regions).”

“Although rDLPFC and rVLPFC contribute to action and thought stopping, it remains to be shown whether either or both regions causally modulate target regions during each task, one of the five key attributes of dynamic targeting (Attribute 4: Causal modulation).”

We believe that these changes achieve the broad goal of the reviewer without compromising our current organisation.

Reviewer Comment 3.10

5. Another consideration for the Discussion section (which would go a long way toward building an intuition about inhibitory control mechanisms) is to expand upon other domains outside of thought and action. A meta-analysis was included in defining the source ROIs - why was this restricted to studies that similarly probed think/no-think and go/stop? It might be helpful to either explicitly mention this as a limitation, or to include another meta-analysis (or an extension to the current meta-analysis) that includes studies of inhibitory control in other domains. More concretely: thought/memory and action/motor domains may specifically differ (in terms of identified source ROIs, as well as how their downstream impact on targets is characterized) from other well-studied domains such as visual/auditory representation, emotional, etc. The Stroop and Flanker tasks also come to mind as prominent assessments of inhibition.

Author Reply 3.10

We agree with the thrust of the reviewer’s suggestion—that it would be good to include domains other than action and thought. Moreover, other tasks, such as the Stroop and Flanker tasks are associated with inhibitory control and might, at first blush, be considered reasonable to include.

We didn’t do so for several important theoretical and practical reasons.

1. *Selection vs Stopping (hippocampal dissociations)*. First, our main concern is the role of inhibitory control in stopping, not in selection amongst competing responses. Why differentiate these? We have, over several papers now, argued that cancellation processes operate in a qualitatively different manner to selection processes, even though some authors might lump them under the single construct of inhibitory control. In our paper on the Amnesic Shadow (Hulbert, Henson, & Anderson, 2016, **Nature Communications**), we found that stopping retrieval of a particular association (e.g. Ordeal Roach) disrupted retention of entirely unrelated pictures that occurred before or after the suppression trial (separated by as much as 10 seconds, with filler activity in between). We predicted that this would occur because retrieval stopping was hypothesized to operate via a *global systemic suppression* of hippocampal activity in an untargeted fashion, which ought to disrupt encoding and consolidation processes for any memory currently reliant on the hippocampus. However, when instead of stopping retrieval, we asked people to recall an alternate association to the cue (inhibition by selection), the amnesic shadow does not occur, nor is hippocampal activity down-regulated. An analogous distinction exists in the motor domain between global and selective stopping.

2. *Selection vs Stopping (prefrontal dissociations)*. On top of this, we have anatomically dissociated retrieval stopping from retrieval selection: whereas stopping engages right DLPFC and VLPFC, thought substitution (recalling an alternate association to a cue to avoid retrieving the original) engages the left VLPFC; moreover, whereas direct suppression down-regulates hippocampal activity, thought substitution does not. Thus, even though both approaches (retrieval stopping and thought substitution) impair retention, they do so in different ways (Benoit & Anderson, 2012, Neuron).
3. *Possible Generality of this Distinction*. There is reason to believe that these lessons regarding the distinction between selection and stopping generalize beyond the memory/thought domain, to other tasks. For example, there is evidence from both Voxel-based lesion-symptom mapping (Cipolotti et al., 2016) and fMRI (Huang et al., 2020) that Stroop relies more on left PFC, and Stroop arguably involves selection to a greater degree than stopping (selecting between competing colour responses). Very broad meta-analyses of inhibitory control comparing interference resolution and action cancellation generally support this possibility (Zheng et al. 2017). Together with points 1 and 2 above, we would have significant reservations about including interference resolution paradigms together with stopping, as we believe they are not tapping the same thing.
4. *Systemic Suppression vs Biased Competition?* We have argued in several sources that whereas biased competition is an elegant model of inhibition in selection tasks, that it may not actually apply to tasks involving the full countermanding of a process in which a competitor is not involved (i.e, stopping), which may reflect a true case of inhibitory control (see, e.g., Hulbert, Henson, & Anderson, 2016, but also Anderson & Weaver, 2009). We believe that the task distinction of Stopping vs Selection roughly corresponds to the mechanistic distinction of Systemic Suppression vs Biased Competition. So, theoretically, we would not wish to mix these. This could possibly encompass the flanker task as well, which seems more likely to fit biased competition, at first blush.
5. *Stopping Tasks Mainly Concern Action and Memory*. Given the foregoing considerations, a focus on STOPPING leads to the inevitable restriction to thoughts and actions. This is because, as far as we are aware, few tasks have yet been devised to focus on stopping perception—or if they do, they would not be plentiful enough to include in a meta-analysis. In research on emotion regulation, there is some work related to affective stopping (see e.g., Depue et al., 2016, Cerebral Cortex), but most of this sizeable literature focuses on Reappraisal (which is likely related to Thought Substitution, a form of selection—see Engen & Anderson, 2018) or Expressive Suppression (stopping overt expression of emotions, rather than the feeling).
6. *Task Length and Matching*. Given that our currently scanned task which includes the TNT task and Stop Signal task lasts a full hour, it would be impractical to include any additional tasks, as it would be hugely fatiguing to participants. Given that our measurement is restricted to these tasks, it is more elegant to have our meta-analysis matched in its content to what participants did. Moreover, it is elegant that these two tasks are both conceptually about the same thing—stopping—only differing in the target content. This focus serves our current purposes well. However, we are certainly interested in conducting new work focusing on such dissociations between inhibitory control tasks.

7. *Self-Control Focus*. Finally, and not trivially, our research is concerned with how people control their actions and thoughts, and whether a common mechanism of self-control for these types of content exists. We have focused on this because these two elements are more directly relevant to mental health, a major focus of our group. Stopping a thought and stopping an action are transparently relevant to controlling intrusive thoughts and controlling impulses, whereas naming an ink colour is less so.

Given these reasons, we chose to focus exclusively on stopping. We have been careful to restrict our claims to stopping. Indeed, in the final discussion, we point to how stopping may be qualitatively different from “interference resolution” (which may be more squarely FPN). Notably, this distinction is not yet proven (a topic for another paper), but we are sufficiently attuned to it now that we would not wish to extend to other tasks like Stroop or Flanker. We think that the lumping of such tasks under inhibitory control may be more confusing than helpful.

Author Action Taken 3.10.

We have made two types of changes to respond to the reviewer’s suggestions.

1. We have undertaken revisions to the manuscript to make it even more clear that our focus is on stopping specifically.

Although our abstract and introduction were very clear and specific that we were focused on the role of inhibitory control in stopping, when we carefully examined the whole manuscript, we discovered that the precision of our language drifted to be more general than this. To rectify this, we hunted down all such instances and edited the manuscript to place all mentions of inhibitory control in alignment with our desired focus. This involved 30 distinct edits that we will not bore you with here (though we have highlighted them in text, if you care to look). Changes usually took the form of either swapping “inhibitory control” or “inhibition” with “stopping”, OR qualifying a phrase (e.g. “inhibitory control during stopping” instead of “inhibitory control”) to make it more specific. We did not change all instances of “inhibitory control” or “inhibition” however, as some were contextually appropriate.

We are very pleased with these changes as this has helped to clarify our precise focus for readers, in a consistent way.

2. We have included a specific statement in the Discussion section that makes it clear that we do not yet know whether the current conclusions extend to inhibition during selection, and that some evidence actually suggests a dissociation. This therefore plainly states a clear boundary on what the current data apply to.

We already had a paragraph that addressed the distinction between stopping and interference resolution, which we have revised. See the whole paragraph below, with the relevant addition highlighted at the very end.

“The present findings highlight a potentially important difference between the brain networks involved in **stopping** and other forms of cognitive control that do not require the full cessation of a motor or cognitive process. Maintaining rules in working memory, implementing task sets, performing multi-tasking, and manipulating information actively are all clear cases of cognitive control that can require interference resolution, but do not necessarily entail active stopping. The above functions engage the widely discussed fronto-parietal network (FPN), often

assigned a central role in implementing cognitive control more broadly (Cole and Schneider, 2007; Cole et al., 2013; Duncan, 2010; Fox et al., 2005). One might assume that because **stopping** is a form of cognitive control that the FPN would be central to it as well. Nevertheless, the FPN, though involved in our tasks, appeared less prominent than the CON, which accounted for the majority of distinct cortical parcels participating in our domain-general **stopping** regions. We found little evidence for involvement of major areas of the FPN, including much of the middle frontal gyrus bilaterally in our multimodal inhibition regions. As our meta-analysis and within-subjects comparisons confirm, inhibitory control **during stopping** is strongly right lateralised, which also is not a feature emphasised in research on the FPN. Our findings raise the possibility that stopping actions and thoughts may rely on a distinct network, with different functional characteristics to the FPN. **Whether other functions thought to require inhibitory control (e.g., selection between competing responses, as in Stroop or Flanker interference) also preferentially engage the stopping networks identified here is not addressed in the current work, although some empirical precedents suggest that stopping and selection may engage partially distinct mechanisms (Benoit and Anderson, 2012; Hulbert et al., 2016; Zhang et al., 2017)."**

Reviewer Comment 3.11

Minor considerations

1. The 2 measures of brain score and voxel salience could be clarified further. It would be helpful to provide more detail on the salience measure specifically, as it appears to contain some circularity in its estimation (which may be just confusion over phrasing): in the Results, salience is described as: "...correlation between brain scores and behavioral measurements identify the direction and the strength of the relationship captured by a LV..."; and in the Methods as "...weighted pattern across brain voxels representing the strength of the relationship between the BOLD signal and the behavioral scores." In the Results definition, 'brain score' presumably refers to per-subject variance explained (by the LV) mentioned in the Methods, which implies that correlating those brain scores back with the same behavioral measurements they were based upon (i.e., how the LV is generated in this application of PLS) is how salience is estimated. If this is a mis-reading, then please clarify. Further, the terms brain score/voxel salience are used in the Results text and variance explained/brain salience in the Methods text; using consistent terms across manuscript sections (and related figures; such as the y-axis in Fig. 4b) might help clarify these complex measures.

Author Reply 3.11

We thank the reviewer for highlighting these inconsistencies in our writing, creating confusion. Indeed, we see that we made it confusing even for ourselves. Now, revising the PLS section, we realise that we did not look at individual subject's brain scores as it was not needed for what we wanted to present. Instead, we present only the overall correlation profile of the first latent variable. The label in the Figure 4b was not accurate. The plot does not show brain scores (which are individual subjects' values) but the overall correlation profile.

Author Action Taken 3.11

- We clarified the Methods of the PLS describing in better detail what measures we get with the PLS analysis.

“Each LV contains a singular value, singular image, and correlation profile. The singular value represents the amount of the covariance explained by the LV. The singular image identifies a collection of voxels that, as a group, are most related to the effects expressed in the LV. The numerical weights within the singular image are called brain saliences and represent the strength of the relationship between the BOLD signal and the behavioural scores. A correlation profile represents how the behaviour correlates with the pattern of brain activity identified in the singular saliences image. A dot product of subject’s raw image volume and the singular saliences image produces a brain score for each subject. Brain scores indicate how strongly individual subjects express the pattern on the LV.”

- We revised the corresponding Results section describing more accurately what found and what we report.

“The first latent variable (LV1) identified by PLS accounted for 78% of the covariance between stopping activations and behavioural measures of SSRT and SIF. The correlation profile of LV1 showed a negative correlation with SSRT scores ($r = -0.432$, $[-0.724, -0.030]$ bootstrapped 95% CI) and a positive correlation with SIF scores ($r = 0.441$, $[0.044, 0.729]$ bootstrapped 95% CI; **Error! Reference source not found.b**). According to this correlation pattern, for the brain voxels with significant positive salience, a higher BOLD signal for the Inhibit > Respond contrast predicted faster SSRTs (i.e., better action stopping speed) and larger amounts of SIF (i.e., better memory inhibition). Voxels associated with such significant positive salience arose across the entire set of domain-general conjunction regions except for the inferior parietal lobules (see **Error! Reference source not found.** and **Error! Reference source not found.c**). No voxels were associated with a significant negative salience (i.e., the opposite pattern).”

We changed the y-axis label on Figure 4b which now states “**Brain-behaviour correlation**”.

Reviewer Comment 3.12

On page 16, MVPA similarities are described as an indicator of dynamic targeting. It is unclear to me if similarities refers to similarity in decoding prediction accuracy, or similarities in the patterns themselves?

Author Replay 3.12

We thank the reviewer for this comment. Indeed, we could have made it clearer that we mean similarities in the patterns themselves.

Author Action Taken 3.12

We revised the second sentence of the corresponding section, which now reads as follows:

“However, dynamic targeting predicts similar pattern of activation in the prefrontal regions for the two stopping domains.”

Reviewer Comment 3.13

Relatedly, it appears that dlPFC and vlPFC exhibit high domain-specific prediction accuracies (higher than domain-general component; Fig. 5c). How can we interpret this given the proposal that those source regions function in a primarily domain-general fashion? (potentially a point for the Discussion).

Author Reply 3.13.

We thank the reviewer for raising this question. How can a domain-general region nevertheless contain information that allows one to distinguish Stopping Actions from Stopping Thoughts?

We realise that we did not adequately articulate our assumptions about what is happening in these putative domain-general regions when presenting the decoding data. What follows here is an explicit description of our assumptions, which is now represented in the text.

The dynamic targeting hypothesis predicts 3 types of process will be reflected in multivariate activity in DLPFC and VLPFC voxels:

1. processes that implement the domain-general stopping mechanism (*domain-general stopping features*)
2. processes that accept domain-specific inputs needed to drive stopping (*input features*),
3. processes that effectuate stopping through their interaction with domain-specific representation systems in posterior cortical or subcortical regions (*output features*)

All of these processes are required theoretically. For example, the TNT task involves visually presented reminder words, and the stop cue is the colour red; but in the stop signal task, coloured circles are the cues and the stop signal is auditory. Clearly, for a domain-general mechanism to work, it must be able to receive inputs from the distinct cortical regions that process these types of content. Similarly, because we explicitly manipulate the domain of stopping (thought vs motor), there is a built-in difference in the outputs of LPFC regions across tasks. *In the LPFC, these input and output features may activate the same collection of voxels (“input and output layers”, for example), but may do so according to different multivariate patterns, yielding domain-specific information that is present in domain-general univariate activity.*

In addition, a fourth type of process should be considered: Domain-specific stopping features. Domain-specific stopping processes are not specifically predicted by the dynamic targeting hypothesis to be part of a hub of domain-general control. However, their presence is compatible with domain-general processes in the PFC. However, domain-specific stopping processes should be a feature of target regions: When signals from the PFC reach the hippocampus (or M1), they should engage local inhibitory processes that only are triggered during the relevant type of stopping (e.g., thought stopping for Hippocampus).

These distinct types of features matter because classifiers don't care which features distinguish conditions of a given task. If input or output features provide additional strong bases for predicting accurately, **within-domain** classification (e.g., training to distinguish Inhibit from Respond in one domain and testing in the same domain) can be far higher than cross-task classification (e.g., training to distinguish Inhibit from Respond in one domain and testing in the other) for reasons entirely compatible with the hypothesis.

These distinct features accommodate the findings in the following way.

1. Cross-domain transfer in prediction would arise from domain-general features alone.
2. Within-domain prediction in PFC would reflect all 4 kinds of features—domain-general, input, output and domain-specific, not necessarily equally weighted. The classifier could easily anchor on one of these, for example.
3. Decoding No-Think from Stop trials in PFC would limit the involvement of domain-general features, and so would represent input, output and domain-specific features.
4. Decoding No-Think and Stop trials in Target regions would similarly limit the involvement of domain-general features (which we think aren't present anyway), and would represent input, output, and domain-specific stopping features.

We hope that this clarifies our assumptions. Next, we describe how we modified the manuscript to allow readers to benefit from this comment.

Author Action Taken 3.13

We made two major changes to the manuscript that answer the reviewer's question.

1. In the Results section reporting our decoding analyses, we have substantially revised the first paragraph to explicitly discuss our assumptions about the type of features. We thoroughly revised paragraph 1 of the decoding section (the section entitled "Action and thought stopping share common..." on page 17), pointing out the potential role of the 4 types of processes in domain-general univariate activity. We explicitly define domain-general stopping features, input features, output features, and domain-specific stopping features. We also revised paragraph 2 highlighting that it is difficult to know the relative contributions of input features output features, or domain-specific stopping. We made further revisions to paragraph 3 to integrate these ideas with findings from the target regions affected by inhibitory control.

The new material addressing these points can be found in the highlighted sections below from Paragraphs 1 and 2.

"It is possible that despite sharing activations in the rDLPFC and rVLPFC, the pattern of activation across voxels within these regions fundamentally differs for action and thought stopping, a possibility that cannot be excluded with univariate methods. However, dynamic targeting predicts similarities in the pattern of activation observed in prefrontal regions for the two stopping domains (Attribute 1: Domain-generality). Specifically, domain-general univariate activations in rDLPFC and rVLPFC should reflect three types of processes: (a) processes that implement the domain-general stopping mechanism (domain-general stopping features); (b) processes that accept domain-specific inputs needed to drive stopping (input features); and (c) processes that effectuate stopping through their interaction with domain-specific posterior cortical or subcortical regions (output features). Input, stopping, and output features are each necessary components of a stopping mechanism with the flexibility to be triggered by multiple modalities and act on diverse processing domains. However, aspects of stopping also may be unique to each domain, yielding a fourth type of process: domain-specific stopping features. Domain-general stopping features should yield similarities between the multivariate patterns for action and thought stopping; in contrast, input, output, and domain-specific stopping features should yield differences in the multivariate patterns for thought and action stopping, with the relative contributions of each being difficult to disentangle. Cross-modality decoding should not be possible in domain-specific target regions, reflecting their

specialised involvement in action or memory stopping. Conversely, between-modality decoding, reflecting domain-specific features, must exist in the domain-specific target regions and also to some extent in the domain-general source regions.

To identify the predicted cross-modality similarities, we trained a classifier to distinguish Inhibit and Respond conditions in one modality and tested the ability to distinguish Inhibit and Respond conditions in the other modality. We performed the classification analysis on the rDLPFC, rVLPFC, right hippocampus, and left M1 ROIs (see Methods). The analysis revealed that a classifier trained on one modality could discriminate Inhibition from Respond conditions in the other modality significantly above chance (50%) for both rDLPFC ($M = 58\%$, $SD = 9\%$, one-tailed $t_{23} = 4.17$, $p_{\text{adj}} = 0.001$, $d = 0.852$) and rVLPFC ($M = 60\%$, $SD = 11\%$, one-tailed $t_{23} = 4.46$, $p_{\text{adj}} < 0.001$, $d = 0.911$). This cross-modality decoding suggests that a domain-general inhibitory control process contributes to activity in these regions (see **Error! Reference source not found.c**; cross-modality prediction accuracy is even stronger in early task blocks—see next section). To identify between-task differences (the domain-specific features), we trained a classifier to discriminate Stop from No-Think trials (see Methods). The classifier could indeed discriminate action and thought stopping in both rDLPFC ($M = 70\%$, $SD = 13\%$, one-tailed $t_{23} = 7.37$, $p_{\text{adj}} < 0.001$, $d = 1.504$) and rVLPFC ($M = 82\%$, $SD = 11\%$, $t_{23} = 13.89$, $p_{\text{adj}} < 0.001$, $d = 2.835$). It is unclear whether this domain-specific component in the LPFC reflects evidence for the input and output features required by the dynamic targeting hypothesis or instead domain-specific stopping processes, either of which may be exploited by a classifier to enhance between-modality prediction performance. The cross-modality prediction findings, however, clearly confirm predictions of the domain-generality attribute of dynamic targeting.”

2. We added a new discussion paragraph that takes up the issue of domain-specific information in our domain-general region. In it, we present our view about why domain-specific information is found (i.e., it has to do with input and output features), but also raise the possibility that domain-specific inhibitory processes may also contribute. We are grateful to the reviewer for suggesting that we add this final discussion paragraph, as many readers undoubtedly would have had the same thought.

The new paragraph is the 6th one in the Discussion and is reprinted below for convenience.

“Although rDLPFC and rVLPFC exhibit core properties needed to achieve domain-general stopping, there is also evidence of domain-specificity within these regions. A classifier trained to distinguish stopping actions from stopping thoughts performed well. Although seemingly inconsistent with domain generality, such effects can be understood as consequences of the dynamic targeting functionality afforded by these regions. For a region to serve as a flexible hub of inhibitory control, it must be able to receive inputs from diverse cortical sites, representing information needed to drive the stopping (e.g., the perception of the coloured word for the No-Think task, and of the tone for the stop-signal task). Moreover, to flexibly target inhibitory control, source regions must interact with diverse cortical and subcortical targets (e.g., hippocampus vs M1), the processing in which must be stopped. These input and output processes may manifest in unique multivariate patterns over common voxels within each region. We cannot rule out, however, the possibility that domain-specific inhibition processes are also manifested by unique patterns in these regions. Future work should examine the additional contributions of input, output, and domain-specific stopping features to the activations found in rDLPFC and rVLPFC.”

Reviewer Comment 3.14

3. Why were the Results on pg. 33 (related to Fig. 4) masked to the meta-analytic conjunction ROIs (i.e., Table 1b)? What is the rationale for using those regions as opposed to the ROIs in Table 1a (ROIs from the current empirical dataset) or 1c (empirical plus the meta-analysis ROIs)?

Author Reply 3.14.

We apologise for not explaining our rationale more clearly in the paper. Basically, the idea was to use the meta-analytic conjunction as an *a priori* ROI. This ROI essentially reflects the best estimate of where a domain-general mechanism spanning thought and action stopping ought to be, based on all relevant preceding (and independent) evidence. The independence of this evidence from the current data set was intended to be a strength of our approach. Notably, however, due to the impressively good overlap between this meta-analytic conjunction and the within-subjects conjunction, it makes very little difference which ROI you use.

Author Action Taken 3.14.

We considered whether to also report all of the current analyses using the within-subjects conjunction, but elected not to do this, as our *a priori* approach is strong and clear. However, if the reviewer would like, we could add these analyses to a (very long) supplement. We have conducted them, and this change in ROI makes very little difference to any conclusion. We did, however, add text to clarify the rationale for using this meta-analytic ROI, so readers can understand why we took this approach. The relevant sentence was added as sentence 2 in the PLS section, and is printed below:

“To ensure that our region was representative of domain-general stopping activations across published studies and was based on data independent of the current experiment, we focused on the meta-analytic conjunction region (n.b. results are similar if the within-subjects conjunction is used).”

Reviewer Comment 3.15

4. Were the subject-specific ROIs (and voxel thresholding) used in the DCM analyses also used in other analyses (involving those same ROIs; for example those in Fig. 5)? If not, text on how one can compare (and interpret) the findings is warranted (e.g., synthesizing results between Fig. 5 and Fig. 6). Note that this may generally be warranted (i.e., details on how, for each type of analysis including univariate, multivariate, and connectivity, ROI definitions are different/the same and what that means for interpreting results) given the text in the Methods that describes alterations to the GLM in the DCM vs. the univariate analyses (i.e., that the data was processed differently).

Author Reply 3.15

We thank the reviewer for this important attention to detail! In the response to this comment, we realised that we had not updated some of our results to match our latest voxel selection approach for the DCM.

In an earlier version of the manuscript, we had used the top 60% of voxels for the DCM, MVPA and percent-signal-change (PSC) analyses. In that version the selected voxels were exactly the same for all three types of analyses. However, for the final submission we changed the voxel selection approach for the DCM to avoid the arbitrary choice of the top 60% (this approach was used in our previous paper, Schmitz et al, 2017) and adopted what we think is a more principled

approach. That is, we selected all voxels that are significant at $p < 0.05$. (Notably this new approach did not change our DCM results). Regrettably, we failed to update the voxel selection approach for our MVPA and PSC analyses. We have now done it and it did not qualitatively change any of the reported results.

For the MVPA and PSC analyses, same as for the DCM, within the same four basis ROIs (rDLPFC, rVLPFC, right hippocampus, and left M1) we selected all voxels significant at $p < 0.05$ based on the main-effect. However, for the DCM we did this selection and used the first-level models with concatenated runs, while for the MVPA and PSC we selected voxels and used models with unconcatenated (original) runs. As in all cases the voxel selection was based on the same principles and using the same basis ROIs, it does not affect the interpretation of the used regions which should be assumed the same across these analyses.

We have now clarified these details in our corresponding Methods and Results sections as detailed below.

Author Action Taken 3.15

- In the DCM Methods section, we now explicitly state the following:

“This selection was performed on the first-level GLMs with concatenated functional runs, described in the next paragraph.”

- In the MVPA Methods section, we included the following text:

“Same as for the DCM analysis, within each subject-specific ROI, we identified all significant voxels (thresholded at $p < 0.05$, uncorrected for multiple comparisons) for that participant based on the main effect of interest, which included Stop, Go, No-Think, and Think conditions. This selection was performed on the first-level GLMs described in the next paragraph. Only the identified significant voxels were included in the final ROIs for the MVPA analysis. Note that the selected voxels within the basis ROIs might not be exactly the same as used in the DCM analysis as for the DCM analysis we used a first-level model with concatenated functional runs. Nevertheless, the selection was based on the same principles and using the same basis ROIs and does not affect the interpretation of the used regions.”

- From the MVPA Results section we deleted the last part of this sentence: “*We performed the classification analysis on the rDLPFC, rVLPFC, right hippocampus, and left M1 ROIs which we defined for our DCM analyses* (see Methods)”, which now reads:

“We performed the classification analysis on the rDLPFC, rVLPFC, right hippocampus, and left M1 ROIs (see Methods).”

- In the Results section where we report target ROI percent-signal-change (‘Stopping actions and stopping thoughts downregulates domain-specific target areas’), we changed this sentence: “.. using the left M1 and the right hippocampus ROIs which we defined specifically for the current **DCM** (see Methods).” to this:

“.. using the left M1 and the right hippocampus ROIs which we defined specifically for the current **analyses** (see Methods).”

- We updated the PSC and MVPA results which did not qualitatively change from what was initially reported.

Reviewer Comment 3.16

5. The outlining of domain-general sources and domain-specific targets in Figure 1 is quite clear. However, the text (e.g., pages 2-3 of the Introduction) describing the 5 attributes does not explicitly delineate this. It would be beneficial for reader comprehension to explain this in-text, perhaps as an initial sentence or two in that section, to give context to the attributes that are about to be outlined.

Author Reply 3.16.

Thanks for this helpful suggestion!

Author Action Taken 3.16

We have made revisions to the introduction to address this idea. Specifically, we added the helpful context-setting sentence early in the paragraph covering the 5 attributes on page 3. This can be seen below, in sentence 2, a new addition implementing the reviewer's suggestion.

"If these rLPFC regions play a causal role in how domain-general inhibitory control achieves stopping, the question arises as to how inhibition is directed at actions or thoughts. In our dynamic targeting hypothesis, this function is achieved by domain-general sources of inhibitory control in the LPFC that interact with specialised domain-specific target regions, the activity of which may require stopping."

Reviewer Comment 3.17

6. The language used on pages 13-14, lines 348-350, could be more precise in terms of the order of processes being proposed (i.e., one piece of evidence used to build up to the causal inference). For example, it might be more precise to say something akin to: "...downregulated hippocampal activity inhibits memory retrieval." (i.e., indicating that cognition/behavior is at the end of the causal process). This is a subtle difference, but important given the typically higher standards/scrutiny for causal inference. To fully explicate the process would be something akin to: "...inhibitory control mechanisms enacted by the source ROIs downregulates activity in the hippocampus which inhibits memory retrieval."

Author Action Taken 3.17.

Thanks very much for this suggestion, which we implemented in the following revision of the sentence of concern:

"For example, when stopping a motor action is the goal, inhibitory control processes supported by domain-general source regions in the prefrontal cortex downregulate activity in M1, cancelling motor actions (Badry et al., 2009; Chowdhury et al., 2019; Mattia et al., 2012; Sumitash et al., 2019; Zandbelt and Vink, 2010); when the goal is to stop memory retrieval, however, the same prefrontal process downregulates hippocampal activity, interrupting retrieval (Anderson and Hanslmayr, 2014; Anderson et al., 2004, 2016; Benoit and Anderson, 2012; Benoit et al., 2015, 2016; Depue et al., 2007; Gagnepain et al., 2017; Hu et al., 2017; Levy and Anderson, 2012; Liu et al., 2016)."

Reviewer Comment 3.17

7. The assertion on page 5 (lines 155-157) is not immediately obvious to me: "If the capacities to stop actions and thoughts are related, more efficient action stopping, as measured by stop-signal reaction time, should correlate with greater suppression-induced forgetting." While this is certainly

one option (that is supported by the results), it could also be the case that downstream, at the level of domain-specific processing, different target regions exhibit different levels of ‘efficiency’; and moreover, this could vary by subject. That is to say, even with comparable exhibition of domain-general inhibitory control by the rIPFC regions herein, why would one predict that the downstream processes are strongly linked to each other, given that they (the targets) presumably have their own local processing and mechanisms to enact the task at hand?

Author Reply 3.17.

We agree with the reviewer completely on this point. In essence, the reviewer is saying (to put it simplistically) that the efficiency of inhibitory control is covered by two parameters:

S = the efficiency of the domain-general component.

T = the local processing characteristics of the target region.

S is shared across the motor and memory domains and is the likely source of any positive correlation between the two. But it is entirely possible for T to be quite different across domains, deflating any correlation between the two.

The reviewer’s comment echoes a major theme of one of our discussion paragraphs on this very point (the penultimate paragraph in the original submission). We point out that the dynamic targeting framework can accommodate both associations and dissociations of inhibitory control abilities depending on whether an inhibitory control deficit primarily originates from the domain-general or domain specific component. So this paragraph essentially makes the reviewer’s point.

Although we agree that the robustness of the correlation between motor and memory inhibition will be attenuated by variation in the domain-specific component, we believe that the shared variance due to prefrontal function will generally shine through, especially in healthy populations, which is why we wrote our claim as such.

Author Action Taken 3.17

It is possible that deficits in domain-specific components may be so strong that they overwhelm any positive correlation to be expected. In general, we would not expect this in healthy populations, especially when the sample size is large. For these reasons, we added a qualifier to our prediction constraining it to healthy populations. The text now reads:

“If the capacities to stop actions and thoughts are related, more efficient action stopping, as measured by stop-signal reaction time, should correlate with greater suppression-induced forgetting, **at least in healthy samples.**”

Reviewer Comment 3.18

8. The interaction effect of the rm-ANOVA is reported (on pg. 14), what about the main effects? T-test results are reported, which likely more directly test the effects of interest; some explanation (in the Methods) of why t-stats are reported instead of main effects (f) would be useful.

Author Action Taken 3.18

We thank the reviewer for this detail! For completeness, we now report the main effects as well with a following addition to the text:

“The main effects were significant for both the modulatory target regions ($F_{1,47} = 4.32$, $p = 0.043$) and the stopping modality ($F_{1,47} = 6.70$, $p = 0.013$). Post-hoc pairwise comparisons showed that ..”

Reviewer Comment 3.19

9. In the Methods (pg. 30), it's reported that select functional runs of select participants were excluded due to technical issues. How did this impact the conflict reduction results in the Supplement? Was there a different amount of data (i.e., more or less subjects) used for different runs here?

Author Action Taken 3.19

Yes indeed, we should have stated it which we now do with the following amendment:

“We examined how accurately an action stopping classifier distinguishes No-Think from Think conditions for the 8 fMRI runs (we note that there were three missing data points for the 8th run and one missing data point for the 7th run due to exclusion of some functional runs; see Methods).”

Reviewer Comment 3.20

10. In the Methods (pg. 33), it's mentioned that six nuisance parameters were used in the task GLM; were these all related to head motion? Did you account for physiological noise (such as white matter and ventricle timeseries)?

Author Reply 3.20

Yes, as we write in the corresponding sentence, indeed these were “six realignment parameters for each run as additional regressors of no interest, to account for head motion artefacts, and a constant regressor for each run.” We did not include any other nuisance parameters.

Reviewer Comment 3.21

11. Consider swapping the presentation order of Fig. 3 and Fig. 4 given that Fig. 4a results are reported in-text first.

Author Reply 3.21

We considered this, but it is a tricky situation, and we would prefer to not change this presentation order. Although we report the behavioural correlation before the univariate results, the presentation of the behavioural correlation fits with the PLS results. Presenting PLS results before the univariate results would not work at all. Presenting behavioural results alone also would not work as well as presenting them together with the PLS results. So, for the sake of the visualisation, we'd prefer to leave the order and content of our figures as they are.

Reviewer Comment 3.22

12. The word ‘transcending’ is used to describe activation patterns at least twice; consider substituting it with a more quantitatively precise term.

Author Reply 3.22.

Happy to do this. We replaced “transcending” to “generalize over”. For instance:

“We next tested whether activation patterns within these potential source regions generalised over the particular stopping domains”.

Reviewer Comment 3.23

13. What do the numbers in Tables 1 and 2 (left most column) indicate? Is this a ranking?

Author Reply 3.23

The clusters in the Table 1 are ordered based the cluster volume size. We added the numbers to align with the supplementary material tables where we need the cluster numbers as we display the corresponding numbers on the brain for visualisation. However, in the main text the numbers are not really needed and the alignment with the supplementary tables are not that crucial.

Author Action Taken 3.23

To avoid any confusion, we removed the cluster numbers from the Table 1 (there are no numbers on Table 2, though).

REVIEWER 1

Reviewer Comment 1.1

This is a reanalysis of a previously described, human fMRI study (Schmitz et al., 2017). The central focus of the current paper is to test whether there is a common prefrontal mechanism(s) that dynamically inhibits different cognitive processes / neural structures. This is tested using a variety of different fMRI analyses applied to two different tasks: a memory stopping task (Think / No Think) and a motor stopping task (Go / No Go). The manuscript lays out several criteria that should be met in order to support the general prediction. The key point is that No Think and No Go trials should both engage the same prefrontal mechanism in a domain general way whereas that target regions (the hippocampus and M1, respectively) should show down regulation in a domain specific way.

The motivating idea for the paper is very interesting and of potentially broad relevance.

Author Reply 1.1

We thank the reviewer for their kind remarks. We are pleased they recognise that this work is of broad theoretical and translational relevance.

Reviewer Comment 1.2

Several of the analyses are compelling, but are either a conceptual replication or modest advance from prior work. Other findings have substantial logical weaknesses. Or, more specifically, somewhat ambiguous results appear to be interpreted more strongly than is warranted. Ultimately, the main conclusions the reader draws from the paper are (1) that a couple of regions of right prefrontal cortex show some overlap (but not total overlap) for No Think and No Go trials, and (2) the specific prefrontal regions implicated (right VLPFC, right DLPFC) while anatomically distinct, do not seem to show any functional difference. The latter point (the lack of difference between the prefrontal regions) is also a bit unsatisfying because it is hard to fully embrace the null (that these regions are functionally identical).

Author Reply 1.2

Because several points raised here are repeated in the detailed comments, we will reply to each in the relevant sections below. Here we will highlight the importance of the contribution we are making.

Inhibitory control is a fundamental theoretical construct in cognitive neuroscience with considerable translational implications for mental health. Despite widespread discussion of this concept, a central pillar of this idea and how it is implemented in the brain has remained untested: that one or more core regions in the prefrontal cortex exist that dynamically alter their connectivity to diverse target areas to suppress them, depending on task goals.

Even given the volumes of research on inhibitory control, virtually no research has sought to pin down regions that have this theoretically central dynamic targeting capacity. This is a remarkable lacuna for such an important theoretical construct. Our work here lays out theoretical criteria for this construct, and then establishes two key regions that meet them. This work therefore makes a central theoretical contribution.

Reviewer Comment 1.3

Comments

1. For the cross-classification (from memory to motor inhibition), the ideal result would be an interaction between the frontal vs. target regions and within domain vs. cross-domain classification. In other words, the relative ‘cost’ to cross-domain classification should be larger for the target regions than the prefrontal regions. But this is not tested and does not look like it would be statistically supported. Instead, it appears that there is really just a main effect where the prefrontal regions AND the target regions show a clear cost with cross-classification, but because the target regions start out with lower overall classification, they drop to chance whereas the prefrontal regions do not.

Author Reply 1.3

We thank the reviewer for these comments. Before addressing this point, it is important to clarify what may be a misunderstanding about what we report in our classifier figure.

In our classifier figure, we juxtapose a “Domain-specific component” and the “domain-general component”. Our evidence for a domain-specific component is achieved by training a classifier to distinguish between Stop and No-Think conditions (i.e., and testing it on a “left out” run to distinguish Stop from No-Think; see Methods). This classifier identifies distinctive features of each Stopping task. In contrast, the evidence for a domain-general component is achieved with a different classifier—one trained to distinguish Inhibit from Non-Inhibit (e.g., No-Think from Think) in one domain and tested to distinguish Inhibit from Non-Inhibit in the other domain (e.g., Stop from Go). *Because the two classifiers are different, they cannot be compared in terms of “cost” transfer as the reviewer’s comment seeks to do.*

What the reviewer probably had in mind, instead of our Domain-specific component was to look at a within-domain test of the same classifier used to derive evidence for Domain Generality. Thus, instead of testing our Inhibit vs Non-Inhibit classifier in the non-trained stopping domain, test it in trained stopping domain. This would be the same classifier, and thus one could compare a “cost”

of testing on a different domain. One could then test whether this “cost” was greater in Source regions than in Target regions, the interaction the reviewer mentions.

Although we understand why it might be tempting to focus on the interaction described in the comment, there are problems with this approach as evidence for our hypothesis, that make the interaction undiagnostic.

1. *The Most Fundamental Prediction of the hypothesis concerns cross-domain prediction.*

Our key claims are that (a) Source regions should serve domain-general inhibitory control functions, which predicts cross-domain transfer; (b) Target domains represent domain-specific information affected by the source and should therefore not show cross-domain transfer. Without evidence for these two key elements, the hypothesis unambiguously fails. Our data support both of these key predictions.

In contrast, the interaction suggested by the reviewer is not fundamental to the hypothesis and may fail, even when the hypothesis is correct. In the next two sections, we first illustrate this, and then unpack why this happens.

2. *The Interaction of Region (Source vs Target) and Domain (Within vs Cross-Domain) is neither necessary nor sufficient evidence for the dynamic targeting hypothesis. One can get the interaction when the hypothesis is false; one can fail to get the interaction or even get the reverse interaction, when the hypothesis is true.*

Sufficiency. There are clear cases in which the proposed interaction should never be considered evidence for our claims. Consider the hypothetical case presented in the image to the left. The cost of cross-domain transfer is clearly greater in the Hippocampus than it is in the DLPFC. Yet, this would constitute troubling data for the hypothesis because (a) classification accuracy in the DLPFC for cross-domain transfer is at chance (50%), and (b) cross-domain transfer in the hippocampus is robustly above

chance (65%). Neither of these outcomes should happen according to the theory. Thus, obtaining the interaction is not sufficient, showing that it is ambiguous evidence for our hypothesis. In contrast, the theory requires that (a) cross-domain transfer in PFC, and (b) no transfer in target regions.

Necessity. There are cases in which the interaction is absent, but the data clearly favours the hypothesis. The hypothetical case presented in the image to the left is an example in which the interaction would not be significant (the cost is 15% in both cases). Yet, this constitutes excellent evidence for the hypothesis because (a) cross-domain transfer in the DLPFC is quite strong (70%), and (b) cross-domain transfer in the hippocampus is completely absent, as predicted (50%). Indeed, one can get the OPPOSITE interaction (larger cost in the DLPFC than in the hippocampus

the hypothesis. The hypothetical case presented in the image to the left is an example in which the interaction would not be significant (the cost is 15% in both cases). Yet, this constitutes excellent evidence for the hypothesis because (a) cross-domain transfer in the DLPFC is quite strong (70%), and (b) cross-domain transfer in the hippocampus is completely absent, as predicted (50%). Indeed, one can get the OPPOSITE interaction (larger cost in the DLPFC than in the hippocampus

(e.g., 95% → 75% in PFC; vs 60% → 50%, Hippocampus) and the data would still favour the hypothesis. Thus, obtaining the interaction is not necessary to support the theoretical hypothesis.

3. *Differential sensitivity to input and output features across ROIs makes the interaction difficult to interpret.*

Why are cross-domain costs undiagnostic? It boils down to “inputs and outputs”. A general multi-domain mechanism must both have the capacity to receive diverse inputs and to act on multiple domains, by definition. Thus, the theory predicts 3 types of process will be reflected in multivariate activity in DLPFC and VLPFC voxels:

- 1) processes that implement the domain-general stopping mechanism (*domain-general stopping features*),
- 2) processes that accept domain-specific inputs needed to drive stopping (*input features*),
- 3) processes that effectuate stopping through their interaction with domain-specific posterior cortical or subcortical regions (*output features*).

All of these processes are required theoretically. For example, the TNT task involves visually presented reminder words, and the stop cue is the colour red; but in the stop signal task, coloured circles are the cues and the stop signal is auditory. For a domain-general mechanism to work, it must be able to receive inputs from the distinct cortical regions that process these types of content. Similarly, because we explicitly manipulate the domain of stopping (thought vs motor), there is a built-in difference in the outputs of the region across tasks. In the PFC, these input and output features may activate the same collection of voxels (“input and output layers”, for example), but may do so according to different multivariate patterns, yielding domain-specific information that is present in domain-general activity.

In addition, a fourth type of process should be considered: *Domain-specific stopping features*. Domain-specific stopping processes are not specifically predicted by the dynamic targeting hypothesis to be part of a hub of domain-general control. However, domain-specific stopping processes should be a feature of target regions: When signals from the PFC reach the hippocampus (or M1), they should engage local inhibitory processes that

only are triggered during the relevant type of stopping (e.g., thought stopping for Hippocampus).

These distinct types of features matter because classifiers don't care which features distinguish conditions of a given task. If input or output features provide additional strong bases for predicting accurately, within-domain classification can be far higher than cross-task classification for reasons entirely compatible with the hypothesis. *Importantly, there is no good reason to assume that these input and output features influence all chosen ROIs equally.* If the DLPFC and hippocampus are differentially sensitive to such features, the within-to-cross domain prediction cost would be massively influenced in ways that have little to do with the importance of domain-general processes.

In a nutshell, whereas the cost to cross-domain prediction clearly must indicate that there is something task-specific influencing the within-domain classification, the task-specific advantage could easily be down to predicted input/output features that distinguish the conditions (e.g., Think and No-Think), the influence of which cannot be assumed to be constant across ROIs.

For this reason, a focus on cross-domain classification accuracy provides a stronger and more interpretable basis on which to evaluate the theory than does the interaction. Also, the between-domain classification (directly comparing No-Think and Stop) provides stronger and more direct evidence for domain-specific features than within-domain classification.

We hope the illustrations above clarify our approach and complement our discussion above. We thank the reviewer for their comments about the interaction between within-domain and cross-domain classification accuracy because it yielded very interesting discussions about the nature of the evidence essential to the theory.

Author Action Taken 1.3

We thoroughly revised paragraph 1 of the decoding section (the section entitled “Action and thought stopping share common...” on page 17), pointing out the potential role of the 4 types of processes in domain-general univariate activity. We explicitly define domain-general stopping features, input features, output features, and domain-specific stopping features. We also revised paragraph 2 to include the costs referred to by the reviewer, and to discuss what they might mean, highlighting that it is difficult to know the relative contributions of input features output features, or domain-specific stopping. We made further revisions to paragraph 3 to integrate these ideas with findings from the target regions affected by inhibitory control.

The new material addressing these points can be found in the highlighted sections below from Paragraphs 1 and 2.

“It is possible that despite sharing activations in the rDLPFC and rVLPFC, the pattern of activation across voxels within these regions fundamentally differs for action and thought stopping, a possibility that cannot be excluded with univariate methods. However, dynamic targeting predicts similarities in the pattern of activation observed in prefrontal regions for the two stopping domains (Attribute 1: Domain-generality). Specifically, domain-general univariate activations in rDLPFC and rVLPFC should reflect three types of processes: (a) processes that implement the domain-general stopping mechanism (domain-general stopping features); (b) processes that accept domain-specific inputs needed to drive stopping (input features); and (c) processes that effectuate stopping through their

interaction with domain-specific posterior cortical or subcortical regions (output features). Input, stopping, and output features are each necessary components of a stopping mechanism with the flexibility to be triggered by multiple modalities and act on diverse processing domains. However, aspects of stopping also may be unique to each domain, yielding a fourth type of process: domain-specific stopping features. Domain-general stopping features should yield similarities between the multivariate patterns for action and thought stopping; in contrast, input, output, and domain-specific stopping features should yield differences in the multivariate patterns for thought and action stopping, with the relative contributions of each being difficult to disentangle. Cross-modality decoding should not be possible in domain-specific target regions, reflecting their specialised involvement in action or memory stopping. Conversely, between-modality decoding, reflecting domain-specific features, must exist in the domain-specific target regions and also to some extent in the domain-general source regions.

To identify the predicted cross-modality similarities, we trained a classifier to distinguish Inhibit and Respond conditions in one modality and tested the ability to distinguish Inhibit and Respond conditions in the other modality. We performed the classification analysis on the rDLPFC, rVLPFC, right hippocampus, and left M1 ROIs (see Methods). The analysis revealed that a classifier trained on one modality could discriminate Inhibition from Respond conditions in the other modality significantly above chance (50%) for both rDLPFC (M = 58%, SD = 9%, one-tailed $t_{23} = 4.17$, $p_{\text{adj}} = 0.001$, $d = 0.852$) and rVLPFC (M = 60%, SD = 11%, one-tailed $t_{23} = 4.46$, $p_{\text{adj}} < 0.001$, $d = 0.911$). This cross-modality decoding suggests that a domain-general inhibitory control process contributes to activity in these regions (see **Error! Reference source not found.c**; cross-modality prediction accuracy is even stronger in early task blocks—see next section). To identify between-task differences (the domain-specific features), we trained a classifier to discriminate Stop from No-Think trials (see Methods). The classifier could indeed discriminate action and thought stopping in both rDLPFC (M = 70%, SD = 13%, one-tailed $t_{23} = 7.37$, $p_{\text{adj}} < 0.001$, $d = 1.504$) and rVLPFC (M = 82%, SD = 11%, $t_{23} = 13.89$, $p_{\text{adj}} < 0.001$, $d = 2.835$). It is unclear whether this domain-specific component in the LPFC reflects evidence for the input and output features required by the dynamic targeting hypothesis or instead domain-specific stopping processes, either of which may be exploited by a classifier to enhance between-modality prediction performance. The cross-modality prediction findings, however, clearly confirm predictions of the domain-generality attribute of dynamic targeting.”

In addition, we thought that the point about how to interpret the existence of domain-specific features was sufficiently important that it merited a discussion paragraph of its own, to call attention to it. This new paragraph is the 6 paragraph in the discussion and is reprinted below for convenience.

“Although rDLPFC and rVLPFC exhibit core properties needed to achieve domain-general stopping, there is also evidence of domain-specificity within these regions. A classifier trained to distinguish stopping actions from stopping thoughts performed well. Although seemingly inconsistent with domain generality, such effects can be understood as consequences of the dynamic targeting functionality afforded by these regions. For a region to serve as a flexible hub of inhibitory control, it must be able to receive inputs from diverse cortical sites, representing information needed to drive the stopping (e.g., the perception of the coloured word for the No-Think task, and of the tone for the stop-signal task). Moreover, to flexibly target inhibitory control, source regions must interact with diverse cortical and subcortical targets (e.g., hippocampus vs M1), the processing in which must be stopped. These input and output processes may manifest in unique multivariate patterns over common voxels within each region. We cannot rule out, however, the possibility that domain-specific inhibition processes are also manifested by unique patterns in these regions. Future work should

examine the additional contributions of input, output, and domain-specific stopping features to the activations found in rDLPFC and rVLPFC.”

We also updated Figure 5c to make it visually more obvious what type of classification each component contains.

Reviewer Comment 1.4

2. Relatedly: cross classification works, which is interesting, but there is certainly a cost—even for prefrontal regions. It’s hard to know, a priori, how much of a cost is reasonable for a domain general mechanism. Successful transfer is taken as evidence of domain generality, but the cost also can be taken as evidence of domain specificity. The authors acknowledge this briefly, but it feels a bit like a glass half empty vs. half full scenario.

Author Reply 1.4

If we understand correctly, the reviewer’s concern is that the domain-specific component is higher than the domain-general component even in the prefrontal source regions. As indicated in Author Reply 1.3., we agree that this observation might seem concerning to the domain-generality of the regions. Domain-specific features, however, do not render conclusions about domain-generality a matter of perspective as suggested by this comment. Several factors are worth considering.

1. *Significant cross-domain decoding occurs.* The key thing is that, despite wide variation in task structure, cross-domain decoding occurs, a simple fact that clearly supports domain generality. Critically, this cross-task decoding is related to behaviour, indicating that it makes a non-trivial contribution.
2. *A domain-specific classification advantage is consistent with the theory and does not weaken the evidence.* As noted in Author Reply 1.3, domain-specific features in the PFC are a necessary consequence of a domain-general mechanism. Domain-specific features must occur because distinct input and output channels from the PFC are needed to implement its diverse influence. If the between-domain classifier (which is trained to distinguish No-Think from Stop) leverages input or output features, the contribution of these domain-specific features could be quite large to prediction accuracy. This does not affect the importance of the shared (not input/output) computations that implement domain-general inhibitory control, revealed by the cross-domain transfer.
3. *The Overall Level of Cross-Domain transfer is deflated by the theoretically predicted Conflict Reduction Benefit (CRB).* The classification rate of 59% in cross-domain prediction is an underestimation of the relationship between inhibitory processes involved in stopping actions and thoughts. As highlighted in the section entitled “*Adaptive forgetting can be predicted from action stopping representations*”, the conflict reduction benefit is why overall cross-task classification accuracy is 59%. The conflict reduction benefit refers to the fact that, when a person suppresses the same thought multiple times, suppression becomes less necessary with each iteration, as the suppressed content becomes less accessible and control, less necessary. This effect has been observed repeatedly in various forms (see Anderson & Hulbert, 2021, Annual Review of Psychology).

The Conflict Reduction Benefit can be seen in the fact that cross-domain classification accuracy declines robustly across blocks of the TNT task, exactly as expected. In block 1, the ability to detect No-Think trials based on a Stop-signal classifier is 77% in the DLPFC.

This declines from 77% down to 40% by block 8, and the slope of this decline predicts later forgetting, as predicted.

The CRB indicates that during block 1, before the effects of memory inhibition have accumulated, people are robustly engaging the domain-general stopping process; but by block 8, this has become less necessary. Thus, the low overall classification arises from deflation in the accuracy, due to later blocks with less engagement of inhibition, which is theoretically predicted. Correspondingly, if you train a TNT classifier on all blocks and try to decode motor inhibition, accuracy will be lower than expected, here again, because of the contribution of the later blocks of TNT.

Author Action Taken 1.4

As noted in Author Reply 1.3, we now explicitly discuss what the presence of domain-specific features might mean. This discussion highlights that the domain-specific features could reflect input, output or domain-specific inhibition features, in unknown proportions, rendering it hard to interpret. This new material highlights the cost topic raised by the reviewer, enabling readers to consider the merits of the arguments. This is a valuable addition to the paper, and we thank the reviewer for calling our attention to it.

In addition to the text added in response to Reviewer point 1.3 (which partially addresses this comment), we also realised that it would be helpful to foreshadow for readers the findings in our later section on the Conflict Reduction Benefit which show that classification accuracy is as high as 77% in block 1. This foreshadowing can be seen below, which is part of the revised paragraph 2 of the decoding section.

“This cross-modality decoding suggests that a domain-general inhibitory control process contributes to activity in these regions (see **Error! Reference source not found.c**; cross-modality prediction accuracy is even stronger in early task blocks—see next section).“

In addition, in the section on the conflict reduction benefit, we now also explicitly discuss how the pattern observed (declining prediction over blocks) likely deflates overall classification accuracy, and that a kappa of 9% is an underestimate of the generalizable component of across these tasks. This discussion can be found in the final paragraph of the section on Adaptive Forgetting, marked in yellow below.

“We examined how accurately an action stopping classifier distinguishes No-Think from Think conditions for the 8 fMRI runs (we note that there were three missing data points for the 8th run and one missing data point for the 7th run due to exclusion of some functional runs; see Methods). The rDLPFC showed a robust linear decline ($F_{7,157} = 9.61$, $p = 0.002$) in classification accuracy from the first ($M = 77\%$) to the eighth ($M = 40\%$) run, consistent with a conflict-reduction benefit (see **Error! Reference source not found.a**). The rVLPFC exhibited a marginal linear decline ($F_{1,157} = 2.88$, $p = 0.092$) in classification accuracy from the first ($M = 67\%$) to the eighth ($M = 29\%$) run (see **Error! Reference source not found.b**). Thus, in both the rDLPFC and rVLPFC, early blocks showed markedly better prediction accuracy than later ones, suggesting that domain-general processes are especially important during early attempts at thought stopping. Critically, for both rDLPFC ($r_{ss} = -0.676$, $p < 0.001$; **Error! Reference source not found.c**) and rVLPFC ($r_{ss} = -0.570$, $p = 0.004$; **Error! Reference source not found.d**), participants showing greater SIF exhibited a steeper classification accuracy decline. **This suggests that adaptive forgetting had diminished demands on inhibitory control as blocks of thought suppression progressed. Notably, this cross-block decline should reduce the contribution of domain-**

general inhibition features to classifiers trained on all blocks together, deflating cross-domain prediction accuracy. If so, the cross-domain prediction accuracies reported for rDLPFC and rVLPFC in the preceding section underestimate the similarity of action and thought stopping. To further confirm that the conflict reduction benefit in the thought stopping task likely arises from a domain-general inhibition process, we related this decline to individual differences in motor inhibition speed. Consistent with the involvement of inhibition, the decline in classifier performance was associated to SSRT for both rDLPFC ($r_{ss} = 0.498$, $p = 0.013$; **Error! Reference source not found.e**) and rVLPFC ($r_{ss} = 0.416$, $p = 0.043$; **Error! Reference source not found.f**). Together, these findings support the view that suppressing unwanted thoughts engages a domain-general inhibition process indexed by action stopping and suggests that both rDLPFC and rVLPFC support this process.”

Reviewer Comment 1.5

3. Inferentially, it is also a bit strange that the successful cross-classification is taken as evidence for a domain general mechanism, but the successful classification of Stop vs. No Think trials (i.e., evidence that inhibition trials differed, in some way, across tasks) is NOT taken as evidence against a domain general mechanism. This is difficult to reconcile.

Author Reply 1.5

We apologise that the presentation of our analyses was unclear. Our interpretation of the foregoing analyses is internally consistent, and not difficult to reconcile. They centre around the distinction between domain-general stopping features, input features, output features, and domain-specific stopping features, as described in more depth in Author Reply 1.3.3 and now explicitly elaborated in first paragraph of the decoding section of the results.

These distinct features accommodate the findings in the following way.

1. Cross-domain transfer in prediction would arise from domain-general features alone.
2. Within-domain prediction in PFC would reflect all 4 kinds of features—domain-general, input, output and domain-specific, not necessarily equally weighted. The classifier could easily anchor on one of these, for example.
3. Decoding No-Think from Stop trials in PFC would limit the involvement of domain-general features, and so would represent input, output and domain-specific stopping features.
4. Decoding No-Think and Stop trials in Target regions would similarly limit the involvement of domain-general features (which we think aren't present anyway), and would represent input, output, and domain-specific stopping features.

As noted in our earlier responses, the ability to decode Stop trials from No-Think trials is fully expected and indeed necessary in the theoretical framework. The existence of such task-specific features is not evidence against a domain-general component that is (demonstrably) functionally important.

Author Action Taken 1.5

Reviewer Comment 1.5 is addressed by the changes described in Author Reply 1.3. By introducing clear language differentiating different types of features that classifiers may respond to, we

hopefully will have a clearer presentation that doesn't create the kind of confusion the reader experienced. We thank the reviewer for the chance to clarify this.

Reviewer Comment 1.6

4. Continuing on the same theme: the logic is even harder to follow because for M1 and hippocampus, the successful discrimination between Stop vs. No Think trials IS taken as critical evidence for domain specific mechanisms. So, why is successful discrimination of Stop vs. No Think trials in prefrontal regions NOT taken as evidence against a domain general mechanism while it IS taken as evidence against a domain general mechanism for M1 and hippocampus? I thought I might be mis-reading the arguments, but I read this section of the text several times and could not understand the logic. As with several of the other results, this appears to be an example where there is a confirmation bias in how the results are interpreted.

Author Reply 1.6

We apologise that our text was not sufficiently clear to convey our intended interpretation of these data. To be clear, the main evidence AGAINST domain generality in M1 and Hippocampus does not come from the ability to distinguish No-Think from Stop; it is from the inability of the classifier in one stopping domain (e.g., No-Think vs Think) to predict the corresponding distinction in the other stopping domain (Stop vs Go) in the target structures.

The ability to distinguish No-Think from Stop instead is NOT evidence against a domain-general component, but rather indicates the contribution of distinctive task-specific components (e.g., input features, output features, and domain-specific inhibition processes). One can definitely have both distinctive task-specific features and domain-general features within an ROI, and the presence of one does not imply the absence of the other. We see how this might have been confusing in our original manuscript.

As noted in our reply to reviewer comment 1.5, our interpretations of these data are fully internally consistent. Our theoretical framework (of necessity) must include a way in which PFC can receive inputs from multiple domains and extend its influence on diverse cortical areas. These input/output considerations are by definition domain-specific and cannot sensibly be eliminated. So, finding evidence for differences between Stop and No-Think in the PFC, for example, can readily be understood from these considerations.

In the case of Target structures, we would expect that an inhibitory process dynamically targeting the structure to be mediated by a particular mechanism that promotes GABAergic inhibition processes (as discussed in the paper); because dynamic targeting implies that the target structure is only affected by such inhibition during its relevant task (e.g., the hippocampus by retrieval stopping), retrieval-stopping in the hippocampus should be discriminable from action stopping.

Author Action Taken 1.6

Here too, Reviewer Comment 1.6 is addressed by the changes described in Author Reply 1.3. We hope that our explicit description of our assumptions clarifies things and enables readers to better understand our data and how to interpret it.

Reviewer Comment 1.7

5. There is no positive control—i.e., some other condition, ideally a cognitively taxing condition—that is contrasted against the motor and memory inhibition effects. This would be useful in several ways. For example, if there was some other cognitively demanding task (maybe mental arithmetic), would right VLPFC and right DLPFC also show connectivity with a brain region involved with this task? The issue is that inhibition (for both motor and memory) is “harder” than Think/Go trials, so we simply don’t know whether or to what degree the prefrontal effects are related to inhibition, per se, as opposed to a much more generic effect of difficulty or effort.

Author Reply 1.7

The reviewer is correct that Stop and No-Think trials are both more difficult than Go and Think trials. Univariate activity in lateral PFC may reflect effort, and this issue is often raised in cognitive control work. For example, our colleague John Duncan (together with Adrian Owen) posited a “multiple demand system” in the lateral prefrontal cortex which responds broadly to difficult cognitive tasks, irrespective of their nature (Duncan & Owen, 2000), and there is a surge of work on the Fronto-Parietal system’s role in cognitive control tasks of many different stripes (see Michael Cole’s work) So, we are fully attuned to the reviewer’s point.

Nonetheless, there are several reasons why an additional Positive Control condition is not necessary for the claims we are making here.

1. *The proposed positive control would make a different point than the one we are making.* A positive control is only necessary if we are seeking to claim that these two frontal regions (RDLPFC and RVL PFC) are intrinsically inhibitory control regions and do nothing else. We do not make such a bold claim. Rather, we claim that when one is engaged in inhibitory control (over action or thought), these regions are engaged. Thus, we are claiming “If Stopping (of retrieval or action) is to be achieved → VLPFC and DLPFC will be engaged”. We are not claiming “IF DLPFC or VLPFC is engaged, it implies response inhibition (the reverse inference).

The current data clearly support the novel claim we make, which is important on its own merits. Even if one found that arithmetic engaged these regions, it would not imply that the regions do not support domain-general inhibitory control.

2. *Our domain-general regions are mostly distinct from those associated with the multiple demand system.* Our univariate data suggests that effort is a less likely explanation of our data compared to inhibitory control. The intensively studied “multiple demand system” (John Duncan) and other analogous claims by other authors (e.g., Michael Cole) generally identify brain regions that are different than the ones we have identified here, though partially overlapping. As we took effort to show, our domain-general frontal activations mostly lie in the Cingulo-Opercular network, with relatively fewer in the Fronto-Parietal (Multiple Demand) system associated with task-independent effort. Indeed, John Duncan has read our paper and is persuaded that these regions are meaningfully different. Of course, work needs to be done to better understand the distinct contributions of these networks, but our data are not readily explained by empirically informed expectations about effort-driven activations.
3. *Our dynamic targeting criteria are sufficiently complex and specific, taken in total, to render a difficulty interpretation implausible.* For an alternative account of our data to

work, it would not merely need to explain shared univariate activity in DLPFC and VLPFC, but also every other one of the features of dynamic targeting we predicted in advance, including (a) domain-specific down-regulations in target regions, (b) top-down effective connectivity to the target regions, (c) dynamic shifting of targeting in a task-dependent way, and (d) behavioural relevance of both univariate activation and multivariate patterns in PFC. These well-specified properties of the inhibitory control construct contrast with an “effort” account, which is underspecified and unduly flexible, at best. Indeed, each of the two inhibitory tasks serves as an “effort control” for the other, enabling us to show that two highly effortful tasks differ dramatically in the interaction of these PFC regions with target regions and produce clear inhibitory outcomes. These findings support the view that this is a highly targeted process recruited to achieve a clear regulatory outcome. These well specified criteria and the data to support them render a positive task control unnecessary, especially when considering point 1.7.1, above.

Author Action Taken 1.7

We have now included a new Discussion paragraph in which we make it clear that we are not claiming that rDLPFC and rVLPFC are dedicated “inhibition-only” regions and clarifying the nature of the claim we are able to make. Specifically, we now explicitly state that it is possible that the brain regions identified may serve other functions apart from inhibitory control, and that one should not take presence of activation in these regions as implying that inhibition is present in a task. Most readers will agree with this, as the issue of reverse inference is widely discussed.

This paragraph briefly addresses the issue of task difficulty. We thank the reviewer for this suggestion, which allows us to address a potential reader concern, and improve the paper.

The new paragraph is the third one in the Discussion section and is repeated below for the reviewer’s convenience.

“The proposed role of the rDLPFC and rVLPFC as hubs of domain-general inhibitory control during stopping does not imply that these regions are exclusively dedicated to stopping. Indeed, it seems likely that these regions contribute to many cognitive functions. Rather, the current evidence suggests that when stopping an action or thought is required, these regions are recruited to cancel processing in target areas involved in representing to-be-suppressed content, thereby achieving the desired stopping outcome. Methodologically diverse evidence indicates that this contribution is causally necessary to successful inhibitory control and is not a mere epiphenomenon of doing difficult tasks. First, the current effective connectivity analyses indicate a robust top-down modulation of target regions by these putative prefrontal sources. This finding comports well with lesion and brain stimulation work in both humans and animals indicating that disrupting the function of rDLPFC and rVLPFC severely disrupts the capacity to stop, consistent with causal necessity (Aron et al., 2003; Chambers et al., 2006; Sasaki et al., 1989; Wessel et al., 2013). Second, although action and thought stopping are both difficult tasks, the network dynamically reconfigured its connectivity to target regions to suppress their function in a manner compatible with task goals. These features have the hallmarks of a control process configured to implement a particular regulatory function, rather than a generic response to task difficulty. RDLPC and rVLPFC are likely to work in concert with a broader network to achieve these goals, as our conjunction analyses suggest. The current work does not address the functional role of domain-general regions outside of the prefrontal cortex, the contributions of which should be examined in future work.”

Reviewer Comment 1.8

6. Line 498: the connectivity models overwhelmingly favoured a bidirectional relationship, but it is the top-down component of this relationship that the authors focus on. This seems like a selective emphasis. The bidirectional model means that there is no evidence that the top-down influence of prefrontal regions on target regions is any greater than the top-down influence of target regions on prefrontal cortex. I could easily imagine this very result being interpreted as evidence AGAINST a top-down relationship between prefrontal and target regions; instead, it is taken as clear evidence for a top-down influence of prefrontal on target regions. Again, the focus goes on the aspect of the results that is theory consistent.

Author Reply 1.8

The reviewer is correct that dynamic causal modelling demonstrated clear evidence for bidirectional interaction between Source regions in the PFC and target regions in the hippocampus or M1. DCM rules out models with exclusively top-down modulatory influence from the Source to the Target as accounts of these data. Indeed, such models fared very poorly. None of this, however, makes a difference to the degree of support our findings provide for our hypothesis.

Bottom-up connections from Target regions to Source regions are irrelevant to the predictions of the framework. The key prediction of dynamic targeting is that there exists a top-down influence from the Source regions on the Target regions (which is robustly supported), and that this influence shifts to different targets depending on the task goal. It is interesting, but inessential to the concept, that the target regions also communicate to the Source. Because the dynamic targeting concept is mute with respect to bottom-up influences, it carries no implications at all about the relative strength of top-down to bottom-up connections, and no evidence of any was found, in any case.

In the case of the hippocampus, it would certainly be surprising if connections from the hippocampus to the DLPFC were NOT found, as there are robust direct anatomical links following that trajectory. Anatomists would be surprised if we didn't find this (see the last author's paper with primate neuroanatomist, Helen Barbas for a review of primate anatomical findings relevant to mnemonic control, Anderson, Bunce, Barbas, 2016).

We thus respectfully disagree with the reviewer's comment in its entirety. No aspect of these findings is inconsistent with the theory, and the predicted top-down component is robustly supported, as discussed in the paper. We focus on the top-down component because that is what the paper is about and what is key to our dynamic targeting concept.

Reviewer Comment 1.9

7. The conclusion that right VLPFC and right DLPFC work together—and, effectively, are indistinguishable—is a bit unsatisfying. It does not seem likely that these regions are redundant. Obviously, the current study did not tease apart the contributions of these regions, but perhaps this just reflects a weakness of the current methods—e.g., DCM is perhaps not as sensitive as true causal methods (stimulation). I am not arguing that there is any evidence here for a difference between these regions, but there is only so much that can be inferred from a null result like this. To be fair, the functional equivalence of these regions is not the main take home from the paper but given that the goal is to reconcile thought suppression and motor stopping literatures—and these literatures have tended to highlight different regions (right DLPFC and right VLPFC,

respectively)—it seems a bit too tidy to conclude that, in fact, both regions contribute to both kinds of inhibition in exactly the same way.

Author Reply 1.9

We sympathise with the reviewer's desire for a more functionally differentiated account of the roles of DLPFC vs VLPFC. Indeed, based on the strength of the claims people have made about VLPFC *versus* DLPFC, one might have expected one of these regions to be a pivotal source with the other playing a support function (e.g., either salience detection or task-set maintenance, as stated in the paper). That is not, however, what the data showed – there was no such differentiation of function. It may be surprising and unsatisfactory, but we present the findings of the model as they occurred, agnostically.

It bears emphasis, however, that although inhibitory control theorists may have strong views about the particular computational functions of specific frontal regions, other areas of neuroscience emphasize the integrative behaviour of whole networks. Indeed, one might argue that a network-level inference is the dominant perspective today. Thus, whereas a reader may feel that the lack of differentiation unsatisfying, network theorists would feel vindicated.

Finally, a word of interpretative caution is in order, prompted by the reviewer's statement that "there is only so much that can be inferred from a null result like this". It is inappropriate to view our dynamic causal modelling tests through the lens of null hypothesis testing. Specifically, we used Bayesian inference to compare different models, not frequentist null-hypothesis testing. Some models had bidirectional interactions between the DLPFC and VLPFC, and others unidirectional interactions (and another, with no interaction). There was no evidence for an asymmetry in the interaction between these regions, as might be suggested by one of these regions playing a support role. The modelling provides very strong evidence for bidirectional interaction, and this is not simply a null result.

We do not rule out the possibility that future work may uncover functional differentiation between these regions. We accept the reviewer's point that brain stimulation methods might provide a way forward here. For now, we present to readers what the data tell us about the likely model architecture best characterizing these regions.

Author Action Taken 1.9

We took on board the reviewer's general hesitancy to accept our DCM modelling as the final word on whether there is functional differentiation between DLPFC and VLPFC. As a general rule, it is wise not to put all of one's eggs in one methodological basket. A different method, such as intracranial recording or brain stimulation, may reveal differentiation not in evidence here. As such, we amended our discussion of functional differentiation in these regions to register this prudent stance. We thank the reviewer for encouraging this, as we think this is a better message to send.

"We considered the possibility that only one of these two frontal regions is central to implementing top-down inhibitory control during stopping, with the other providing upstream inputs essential to initiate successful control. Our effective connectivity analysis probed alternative hypotheses about the way rDLPFC and rVLPFC interact during stopping. rDLPFC might implement the true inhibitory signal, receiving salience detection input from rVLPFC that up-regulates rDLPFC function, consistent with a possible role of the VLPFC in the ventral attention network (Corbetta and Shulman, 2002; Corbetta et

al., 2008). Alternatively, rVLPFC may implement inhibition, with rDLPFC preserving task set by sending driving inputs to the rVLPFC. Our findings indicate that both structures contributed in parallel to top-down inhibitory control and interacted bidirectionally during both action and thought stopping. Little evidence suggested a strong asymmetry in how rDLPFC and rVLPFC interacted, as should arise if one region simply served a role in salience detection or task-set maintenance. It remains possible, however, that rDLPFC and rVLPFC serve distinct functions that are not readily separable given the current manipulations and the level of temporal resolution available in fMRI data. Nevertheless, these findings suggest that rDLPFC and rVLPFC, at a minimum, act together to implement top-down inhibitory control during stopping. Although it might seem surprising that two spatially segregated prefrontal regions would act in concert to achieve this function, it seems less unusual considering their potential role in the cingulo-opercular network (CON). The majority of the regions identified in our inhibition conjunction analysis participate in this network, suggesting that it may play an important role in achieving stopping. Given the strong integrated activity of this network, elements of which are distributed throughout the brain (Cocuzza et al., 2020; Cole et al., 2013), this suggests future work should examine how rDLPFC and rVLPFC work together with other elements of this network to achieve successful stopping.”

Reviewer Comment 1.10

8. Line 412 and 413, there are two p values that are listed as “p = 1”—this seems to be some kind of error (the p values should not be exactly 1).

Author Reply 1.10

As stated in the Methods, we are correcting for multiple comparisons (4 ROIs). This can technically result in $p > 1$ which is then capped to $p = 1$. We realise that to avoid any confusion we should clearly state that these are adjusted p-values. Thank you for bringing our attention to this!

Author Action Taken 1.10

- In the MPVA Methods section we appended the last sentence as follows (the addition is highlighted in yellow):

“All tests were Bonferroni corrected for the number of ROIs, and adjusted p-values (p_{adj}) reported.”

- In the Results section we report the adjusted p-values as p_{adj}
- To remind the reader that we are correcting for multiple comparisons, we added this sentence at the end of Figure 5c caption:

“The stars represent the significance of classification accuracy being above 50% chance level (Bonferroni corrected for the number of ROIs).”

REVIEWER 2

Reviewer Comment 2.1

In their manuscript “Dynamic targeting enables domain-general inhibitory control over action and thought by the prefrontal cortex”, Apsvalka and colleagues present data from an fMRI investigation in N=24 that aimed to test whether domain-general regions in (pre)frontal cortex

were conjointly involved in the inhibition of action (operationalized in the stop-signal task) and associative memory (operationalized in the Think/NoThink paradigm). The authors report that indeed, two regions in the prefrontal cortex – specifically, in the right ventro- and dorso-lateral prefrontal cortices – are involved in inhibiting both representations. Moreover, the authors find that these regions work together in specifically targeting the downstream neural targets of the purported inhibitory process in either domain (M1 for action, hippocampus for associative memory). The empirical demonstration is exhaustive, and the question is explored in several complementary ways that result in a coherent picture.

Overall, this is a phenomenal paper. The research presented is very impressive and the authors went through great lengths to truly explore the underlying question from multiple complementary angles. Indeed, I would describe this work as the empirical pinnacle-to-date of a research idea that has received a lot of interest in the recent past (domain-generalizability of inhibitory control beyond motoric representations), with many high-quality papers already published by this group and others.

Author Reply 2.1

We thank the reviewer for their positive remarks about the work! We are delighted that you find the case to be compelling and exhaustive.

Reviewer Comment 2.2

The careful theoretical justification of the criteria used to evaluate domain-generalizability, alongside the affirmative empirical demonstration thereof in the data itself make for a highly compelling package. The manuscript is also extraordinarily well-written. I have very little to offer in the way of criticism, and indeed nothing that would be considered major. A few suggestions, largely regarding the theoretical framing, are listed in the following.

Author Reply 2.2

We have sought to address each of your comments and hope that we succeeded.

Reviewer Comment 2.3

1. In regard to the motivation of the hypothesis of a possibly domain-general inhibitory control system, I have always liked that fact that Logan and colleagues had already suggested this when they described the original horse race model of the stop-signal paradigm (indeed, one of the earliest papers was titled “On the ability to inhibit thought and action”). I personally often like to acknowledge this amazing foresight by Logan and colleagues, but I leave it up to the authors of the current manuscript if they want to incorporate this into the framing and motivation of the study.

Author Reply 2.3.

We agree that Gordon had and has amazing foresight. Indeed, Gordon was an early inspiration to our work on inhibitory control over memory as early as 1995, when we posited a shared inhibition mechanism underlying perceptual and mnemonic attention (Anderson & Spellman, 1995).

Author Action Taken 2.3

Upon examining our introductory paragraph, it became clear that we had not adequately cited Gordon early on in the topic of domain-general control. To rectify this, we added a special nod to Gordon as an early advocate. The text now reads:

“This proposal rests on the concept of inhibitory control, a putative domain-general control mechanism that has attracted much interest in psychology and neuroscience over the last two decades (Anderson et al., 2016; Aron et al., 2004, 2014; Banich and Depue, 2015; Bari and Robbins, 2013; Boucher et al., 2007; Diamond, 2013; Ersche et al., 2012; Eysenck et al., 2007; Joormann and Tanovic, 2015; Lipszyc and Schachar, 2010; for an early work, see Logan and Conwan, 1984).”

Reviewer Comment 2.4

2. In a similar vein – and with awareness that it may seem gauche to bring up our own research – I think that two of our papers were, if not the first, then at least two of the more influential neuroscientific papers to empirically demonstrate (Wessel et al., Nature Communications 2016) as well as theoretically and neuroanatomically embed (Wessel & Aron, Neuron 2017) the proposal that the neural system underlying action stopping may also underlie memory suppression. Despite my obvious bias, I think it would be fair to ask the authors to perhaps cite and discuss those papers.

They may also consider our recent work expanding this picture to attentional representations (Soh & Wessel, Cerebral Cortex, in press, PMID: 33140100), though I would not deem this essential, as it is not strictly about memory.

Author Reply 2.4

We agree that these are highly relevant papers that need to be included and apologise for the oversight on our part. Clearly, these should be cited.

Author Action Taken 2.4

We took the reviewer’s suggestion as an opportunity to dive into this line of work in more depth, and it is clear that this is exciting work. Thank you for the encouragement to do this. We were tempted to include a new paragraph in the discussion specifically to discuss the relationship between our proposal and the global stopping mechanism posited by the reviewer. In the end, however, despite thinking this would be very interesting to readers, we reluctantly decided to not do this because we have so many other required additions from other reviewers and little space to work with. Nevertheless, we were able to include mentions of this work in two places.

1. First, we included it in the introduction, towards the end of paragraph 2, we now include these two papers as part of the general set of proposals relating to domain-general inhibition:

“rVLPFC thus could promote top-down inhibitory control over actions, and possibly inhibitory control more broadly (Aron, 2007; Aron et al., 2004; Castiglione et al., 2019; Wessel and Aron, 2017; Wessel et al., 2016).”

2. We also included a new sentence in the general discussion in paragraph 2 (sentence 2) in which we draw specific attention to this work, without going into details.

“Based on these and related findings, we propose that anterior rDLPFC and rVLPFC constitute key hubs for a domain-general inhibitory control mechanism that can be dynamically targeted

to stop processing of diverse content represented throughout the brain. This proposal complements recent work positing a broad prefrontal inhibition mechanism that can interrupt both cognition and action (Soh and Wessel, 2021; Wessel and Aron, 2017; Wessel et al., 2016).”

Reviewer Comment 2.5

3. Finally, since the vLPFC has been found to be a key cog in the neural network identified here, Corbetta and Shulman’s circuit breaker proposal may warrant some discussion in this paper as well.

Author Reply 2.5.

Point well taken. Indeed, we are long-time fans of Corbetta and Shulman’s ideas, and regret not including references to their key papers.

Author Action Taken 2.5

We have included references to Corbetta and Shulman in two locations in which they would be most relevant.

1. First, In the DCM results section, when talking about different theoretical models of the relationship between VLPFC and DLPFC. Now we link one of our alternative models explicitly to Corbetta and Shulman:

“On the other hand, rDLPFC alone might show dynamic inhibitory targeting, consistent with the emphasis on the rDLPFC as the primary source of inhibitory control in research on thought suppression (Anderson and Hanslmayr, 2014; Anderson and Hulbert, 2021); rVLPFC may only be involved when attention is captured by salient stimuli (Corbetta and Shulman, 2002; Corbetta et al., 2008), such as the stop signal or intrusions, possibly exerting a modulatory effect on rDLPFC to upregulate its activity (rDLPFC alone model).”

2. Second, in the final discussion, when discussing the alternatives considered in our DCM comparisons, in the third paragraph:

“rDLPFC might implement the true inhibitory signal, receiving salience detection input from rVLPFC that up-regulates rDLPFC function, consistent with a possible role of the VLPFC in the ventral attention network (Corbetta and Shulman, 2002; Corbetta et al., 2008).”

Reviewer Comment 2.6

4. I very much liked the inclusion of the BEESTS model, especially since I noticed that the traditional SSRT estimate was quite unusually large. I would like some more detail on this analysis, specifically with regards to the quality of the model fit (BEESTS needs a substantial amount of trials per subject to work properly). Moreover, I’d be curious if the trigger failure parameter correlated with any of the variables of interest.

Author Reply 2.6

We are not sure how to assess the overall quality of the model fit with BEESTS. We were using BEESTS-WTF version 2.0. It provides an output of each subject’s goodness-of-fit plots. We have now included the BEEST output on our github:

https://github.com/dcdace/Domain-general/tree/master/data/SSRT_BEESTS_210216-235533

The trigger failure parameter correlated with both SSRT ($r = 0.658$, $p < 0.001$) and SIF ($r = -0.723$, $p < 0.001$). We have not included this information in our main text, but these plots are available in our supplementary analysis notebook sections 2.7 and 2.8 <http://bit.do/analysis-domain-general>

Reviewer Comment 2.7

Other than that, I have no substantial points of concern, and I congratulate the authors on this beautiful piece of work.

Author Reply 2.7

We thank the reviewer for a very helpful review.

REVIEWER 4

Reviewer Comment 4.1

Thank you for the opportunity to review this manuscript. I found the topic very interesting and the overall ideas to be presented very clearly. I think the paper has the potential to make important contributions to the field. I especially appreciated the author's willingness to lay out multiple steps for testing their hypotheses, and willingness to proactively address potential issues with data/additional tests.

Author Reply 4.1

We thank the reviewer for nice remarks about the paper. We are very happy that the reviewer appreciated our approach to the work.

Reviewer Comment 4.3

The analyses multi-method approach offers convergence across multiple analyses, which I see as a major strength.

Author Reply 4.3

We agree. We feel much more confident in the work as a result of taking a methodologically diverse approach, providing converging evidence, and we hope that readers do too.

Reviewer Comment 4.4

Below I offer some suggestions that I hope will be helpful for improving the manuscript, particularly with respect to the classification method – which is, in my view, the biggest weakness of the paper at present but can also be straightforwardly addressed.

Can the authors provide more details on the classifier? For example, how is chance 50% when the trials Think/No Think and No/NoGo were not balanced for data collection trials? The methods indicate that a random subset was drawn, but it's not clear how class imbalance was dealt with. Typically in a classification task – or when using machine learning principles – the class distribution in the training data can be changed (upsampled/downsampled, etc.), but only in the training set. It is typically not appropriate to change the distribution of the test set.

Author Reply 4.4

Each classifier was trained and tested to distinguish between two conditions. There are only two possible answers a classifier can give in each test case: condition 1, condition 2. That is the case for our domain-specific classifier where the classifier is distinguishing Stop from No-Think. There are only two possible answers a classifier can give: either Stop or No-Think. The guessing chance is 50%. The same is true for the domain-general classifier where the classifier is distinguishing Inhibit from Non-Inhibit. Thus, again only two possible answers, the guessing chance is 50%.

For all classifiers we only operate with the run-average condition values. Thus, for each subject we have 8 values for condition 1 and 8 values for condition 2 (see text in Methods “Both training and testing data consisted of two (conditions) by eight (runs) activation estimates for a set of voxels”).

The random subsets were drawn only on the number of voxels from the whole ROI and is a common practice to improve the reliability of accuracy for ROI-based analyses. As we write in the Methods “Specifically, for each ROI separately, we created up to 2000 unique subsets of randomly drawn 90% of ROI voxels”.

Author Action Taken 4.4

For some more clarity we updated the following sections in the MVPA Methods. The update is highlighted in yellow:

- “To increase the reliability of pattern classification accuracy, we used a random subset approach (Diedrichsen et al., 2013). Specifically, for each ROI separately, we created up to 2000 unique subsets of randomly drawn 90% of ROI voxels (for smaller ROIs, there were less than 2000 possible combinations). We then applied the LDA on each subset of voxels and averaged the subset results to obtain the final classification accuracy for each ROI.”
- “At the group level, for each ROI, we performed one-tailed t-tests to assess the statistical significance of classification accuracy being above the 50% chance level (each classifier was distinguishing between two conditions).”

Reviewer Comment 4.5

Critically, the authors say leave one out cross-validation. This is possibly great, but does it refer to leave one instance out or leave one person out? Leave one instance out offers the possibility of overfitting because of person-dependent instances in the training set. 8-folds are mentioned but it’s unclear how this maps on to leave one out, or perhaps I missing what is “left out.” This is important to address given the well-known issues with overfitting in such models.

Author Reply 4.5

Perhaps the reviewer overlooked that we had stated that “leave-one-run” out cross-validation was used, as the text formatting had put the ‘run’ in a new line, unfortunately.

Our Method states: “For each ROI subset, we performed leave-one-run out cross-validated LDA and averaged the classification accuracies across the eight cross-validation folds.” We hope this clarifies the issue.

Reviewer Comment 4.6

I also encourage the authors to consider the practical significance of their classifier accuracy a bit more. For example, a kappa of .07 (7% increase in classification accuracy; 57% accuracy) above chance would not be considered strong performance in other fields. In short, it is meaningful and is different from chance, but it's not doing that well prediction-wise in some cases.

Author Reply 4.6

We thank the reviewer for highlighting the need to address this issue.

There is a straightforward reason why overall classification accuracy is only 57% (now slightly higher, after minor revisions to ROIs...see replies to reviewer 3). A key idea that we built on in our section "*Adaptive forgetting can be predicted from action stopping representations*" is the "Conflict Reduction Benefit". This refers to the fact that, when a person suppresses the same thought multiple times, suppression becomes less necessary with each iteration, as the suppressed content becomes less accessible and control, less necessary. This effect has been observed repeatedly in the literature on retrieval suppression in various forms (see Anderson & Hulbert, 2021, Annual Review of Psychology).

The Conflict Reduction Benefit is the reason why the overall kappa is low. Consider the change in classification accuracy across blocks of the TNT task. In block 1, the ability to detect No-Think trials based on a Stop-signal classifier is 77% in the DLPFC. This declines progressively over blocks (77% down to 40% by block 8; see new figure in text), exactly as predicted by the conflict reduction benefit, and the slope of this decline predicts later forgetting, also as predicted.

What this essentially means is that during block 1, before any memory inhibition has accumulated, people are robustly engaging the domain-general stopping process; but by block 8, not so much. If you then look at overall classifier accuracy, without distinguishing blocks, you are left with 57%. Thus, the low classification is actually due to deflation in the accuracy, due to later blocks with less engagement of inhibition, which is theoretically predicted and interesting.

And of course, if you train a TNT classifier on all blocks and try to decode motor inhibition, accuracy will be lower than expected, here again, because of the contribution of the later blocks of TNT in which the conflict reduction benefit has accrued.

So, ironically, the lower kappa noted by the reviewer arises precisely because the domain-generality of the process is convincing—otherwise you would not get the predicted conflict reduction benefit. The initial block is a better reflection of the "pure" classification accuracy, with conflict reduction benefits mostly excluded (there are 2 repetitions per block, so it's not entirely excluded). The kappa is 27%, without the CRB, which is more practically significant.

Author Action Taken 4.6

To address the reviewer's point, we made two additions.

1. In the initial section reporting overall decoding accuracy in the PFC, we draw readers' attention to evidence for even stronger decoding reported in the upcoming section on adaptive forgetting. The statement is included below.

“This cross-modality decoding suggests that a domain-general inhibitory control process contributes to activity in these regions (see **Error! Reference source not found.c**; cross-modality prediction accuracy is even stronger in early task blocks—see next section).”

2. In the section on the conflict reduction benefit, we now also explicitly discuss how the pattern observed (declining prediction over blocks) likely deflates overall classification accuracy, and that a kappa of 57% is an underestimate of the generalizable component of across these tasks. This discussion can be found in the final paragraph of the section on Adaptive Forgetting, marked in yellow below.

“We examined how accurately an action stopping classifier distinguishes No-Think from Think conditions for the 8 fMRI runs (we note that there were three missing data points for the 8th run and one missing data point for the 7th run due to exclusion of some functional runs; see Methods). The rDLPFC showed a robust linear decline ($F_{7,157} = 9.61$, $p = 0.002$) in classification accuracy from the first ($M = 77\%$) to the eighth ($M = 40\%$) run, consistent with a conflict-reduction benefit (see **Error! Reference source not found.a**). The rVLPFC exhibited a marginal linear decline ($F_{1,157} = 2.88$, $p = 0.092$) in classification accuracy from the first ($M = 67\%$) to the eighth ($M = 29\%$) run (see **Error! Reference source not found.b**). Thus, in both the rDLPFC and rVLPFC, early blocks showed markedly better prediction accuracy than later ones, suggesting that domain-general processes are especially important during early attempts at thought stopping. Critically, for both rDLPFC ($r_{ss} = -0.676$, $p < 0.001$; **Error! Reference source not found.c**) and rVLPFC ($r_{ss} = -0.570$, $p = 0.004$; **Error! Reference source not found.d**), participants showing greater SIF exhibited a steeper classification accuracy decline. This suggests that adaptive forgetting had diminished demands on inhibitory control as blocks of thought suppression progressed. Notably, this cross-block decline should reduce the contribution of domain-general inhibition features to classifiers trained on all blocks together, deflating cross-domain prediction accuracy. If so, the cross-domain prediction accuracies reported for rDLPFC and rVLPFC in the preceding section underestimate the similarity of action and thought stopping. To further confirm that the conflict reduction benefit in the thought stopping task likely arises from a domain-general inhibition process, we related this decline to individual differences in motor inhibition speed. Consistent with the involvement of inhibition, the decline in classifier performance was associated to SSRT for both rDLPFC ($r_{ss} = 0.498$, $p = 0.013$; **Error! Reference source not found.e**) and rVLPFC ($r_{ss} = 0.416$, $p = 0.043$; **Error! Reference source not found.f**). Together, these findings support the view that suppressing unwanted thoughts engages a domain-general inhibition process indexed by action stopping and suggests that both rDLPFC and rVLPFC support this process.”

Reviewer Comment 4.7

Can the authors provide a bit more review of thought suppression and related regions? In general, the paper is clearly motivated. At the same time, it would be helpful to more explicitly contrast (up front) how previous studies have found specific inhibitory mechanisms (rDLPFC vs rVLPFC) and how this proposal incorporates and extends those findings in a domain general framework.

Author Reply 4.7

We thank the reviewer for this suggestion. In fact, upon reflection, we see the reviewer’s point. We had not quite made it explicit that we are seeking to integrate two disparate sets of claims.

Author Action Taken 4.7

To address the reviewer's point, we revised the end of paragraph 4 in the introduction to highlight the separate evolution of these domains of research, and our effort to bring them together in a single framework. Our edits are below.

"Reinforcing this domain-specific focus, research has posited that control originates from different prefrontal regions in these two domains suggesting separate control abilities: whereas the right anterior DLPFC has received attention in work on thought suppression (Anderson and Hanslmayr, 2014), the right VLPFC has been the focus in work on response inhibition (Aron et al., 2004, 2014), despite both regions often arising in both stopping tasks (Guo et al., 2018). To integrate research from these separate domains, we sought to determine which of these candidate sources of domain-general inhibitory control participate in stopping both actions and thoughts and which exhibit the key attributes of dynamic targeting."

Reviewer Comment 4.8

This is not a critique of the current paper, but I am curious if the authors' view this as domain general process is because people are inherently meta-aware/conscious of their stopping in both cases? So, is the ability to initiate stopping (thought/motor) general necessarily involve that awareness, or would stopping behaviors with a lack of awareness have the same patterns? Again, this is not a concern for the contribution of the current paper, but perhaps could be addressed in the discussion section.

Author Reply 4.8

This is a thought-provoking question. Until recently, we might have answered "probably" because, as the reviewer notes, awareness is strongly associated with these efforts at control. However, various pieces of data indicate otherwise. For example, numerous papers show that conscious awareness and intentionality are not necessary to engage stopping processes in response inhibition. van Gaal and colleagues have found that during a response inhibition task, if you include trials in which the task cue is masked (and participant therefore should view it as a go trial), participants will slow their response and occasionally even stop (van Gaal & Ridderinkhof, 2009; van Gaal et al. 2011; van Gaal et al. 2010; Chiu & Aron, 2014). Similarly, the prefrontal mechanisms associated with motor response inhibition are engaged under these masking conditions, showing that awareness is not needed for the engagement of this stopping system (Van Gaal et al. 2010).

Strikingly, this same logic was recently applied to the TNT task by Salvadaor et al. (2018). They demonstrated that—to our astonishment---that awareness of the task cue for the TNT is not needed to observe suppression-induced forgetting.

So, the answer appears to be, based on available data, that meta-awareness is not key.

Author Action Taken 4.8

We considered adding text to the discussion to raise this interesting possibility, but in the end decided against it due to space considerations. Given that our manuscript was already on the long side and that other additions were required of us, we thought that addressing this topic is something better handled in subsequent work. Indeed, the reviewer's comments have actually changed the shape of a planned experiment in which we were already planning to address highly related issues. We thank the reviewer for their contribution to this set of ideas.

Reviewer Comment 4.9

How might the dynamic targeting work for suppression in people with hippocampal damage? This is not a critique, but I'm wondering if the authors may offer some caveats to the domain general process proposed here. Are there exceptions to the dynamic targeting mechanisms?

Author Reply 4.9

We had not previously considered the effect of hippocampal damage on dynamic targeting. Basically, it would not be possible to suppress hippocampal activity if one does not have a hippocampus, of course. So, does that mean that somebody with hippocampal damage could not suppress intrusive thoughts?

Interesting question. On the one hand, such a patient should not have awareness that a thought is persistently intrusive because such persistence should only be recognized by somebody with intact memory function. On the other hand, somebody with amnesia surely does have thoughts that they find unpleasant, and may take immediate action to suppress them, just like everyone else. Can they do so? We believe so. Our prior work indicates that hippocampus is suppressed in parallel with cortical regions involved in representing the suppressed content. This later aspect should be intact in hippocampal amnesia patients and so should remain possible. Behaviourally, this could be detected by examining the impact of suppression on conceptual or perceptual priming (such impacts have been shown to occur...e.g. Gagnepain et al. 2014; and Wang et al. 2018 (consciousness and cognition).

Reviewer Comment 4.10

The figures were excellent; thanks to the authors for their thoughtfulness in graphical representation.

Author Reply 4.10

Thank you very much! We worked hard on them.

REFERENCES

- Anderson, M.C., Shanker, S., Turnbull, O., Brumerloh, B., & Fawcett, J. (2019). Right, but not left prefrontal cortex is necessary for the suppression of unwanted memories. *Proceedings of the Psychonomics Society*.
- Aron, A.R., Fletcher, P.C., Bullmore, E.T., Sahakian, B.J., and Robbins, T.W. (2003). Stop-signal inhibition disrupted by damage to right inferior frontal gyrus in humans. *Nat. Neurosci.* 6, 115–116.
- Chambers, C.D., Bellgrove, M.A., Stokes, M.G., Henderson, T.R., Garavan, H., Robertson, I.H., Morris, A.P., and Mattingley, J.B. (2006). Executive “brake failure” following deactivation of human frontal lobe. *J. Cogn. Neurosci.* 18, 444–455.
- Chiu, Y. C., & Aron, A. R. (2014). Unconsciously triggered response inhibition requires an executive setting. *Journal of Experimental Psychology: General*, 143(1), 56.
- Cipolotti, L., Spanò, B., Healy, C., Tudor-Sfetea, C., Chan, E., White, M., ... & Bozzali, M. (2016). Inhibition processes are dissociable and lateralized in human prefrontal cortex. *Neuropsychologia*, 93, 1-12.

- Huang, Y., Su, L., & Ma, Q. (2020). The Stroop effect: An activation likelihood estimation meta-analysis in healthy young adults. *Neuroscience letters*, 716, 134683.
- Hulbert, J.C., Henson, R.N., and Anderson, M.C. (2016). Inducing amnesia through systemic suppression. *Nat. Commun.* 7, 11003.
- Salvador, A., Berkovitch, L., Vinckier, F., Cohen, L., Naccache, L., Dehaene, S., & Gaillard, R. (2018). Unconscious memory suppression. *Cognition*, 180, 191-199.
- Sasaki, K., Gemba, H., and Tsujimoto, T. (1989). Suppression of visually initiated hand movement by stimulation of the prefrontal cortex in the monkey. *Brain Res.* 495, 100–107.
- Van Gaal, S., Lamme, V. A., Fahrenfort, J. J., & Ridderinkhof, K. R. (2011). Dissociable brain mechanisms underlying the conscious and unconscious control of behavior. *Journal of cognitive neuroscience*, 23(1), 91-105.
- Van Gaal, S., Ridderinkhof, K. R., Scholte, H. S., & Lamme, V. A. (2010). Unconscious activation of the prefrontal no-go network. *Journal of Neuroscience*, 30(11), 4143-4150.
- van Gaal, S., Ridderinkhof, K. R., van den Wildenberg, W. P., & Lamme, V. A. (2009). Dissociating consciousness from inhibitory control: evidence for unconsciously triggered response inhibition in the stop-signal task. *Journal of Experimental Psychology: Human Perception and Performance*, 35(4), 1129.
- Wessel, J.R., Conner, C.R., Aron, A.R., and Tandon, N. (2013). Chronometric Electrical Stimulation of Right Inferior Frontal Cortex Increases Motor Braking. *J. Neurosci.* 33, 19611–19619.
- Zhang, R., Geng, X., and Lee, T.M.C. (2017). Large-scale functional neural network correlates of response inhibition: an fMRI meta-analysis. *Brain Struct. Funct.* 222, 3973–3990.

Reviewers' comments:

Reviewer #1 (Remarks to the Author):

The authors were in some ways very responsive in that they provided very detailed arguments and consideration of the concerns that were raised. In other ways, they were surprisingly un-responsive in that there was very little willingness to respond with new data or analyses (in response to comments from the reviewers, collectively). However, I want to acknowledge that I was clearly the most negative of the initial reviewers. Indeed, there was strong excitement from other reviewers. While some of my concerns persist (in large part because they were not addressed with data), I do not wish to 'talk down' the other reviewers. Rather, I will re-articulate my lingering concerns but I have no issues with these concerns being offset by greater enthusiasm from other reviewers. Indeed, as noted in my initial review, there are several positive aspects of this paper (broad interest, an appealing combination of methods). The sticking point for me is the strength of inferences drawn from the data and what I view as a bit of 'spin' that is put on the data.

The following are related to my original comment (comment 1.3):

(a) The authors are correct that I misinterpreted the original classification results. I apologize for the mistake. However, I find the logic/presentation here confusing, which contributed to my initial mistake. As the authors surmised, I had thought the domain specific analysis was testing for Respond vs. Inhibit classification WITHIN each domain. In other words, I had interpreted "domain specific" to mean that the analysis was "within domain." Instead, this analysis is testing for classification of stop vs. no-think trials. Thus, successful classification of stop vs. no-think trials means there is something different in the pattern of bold responses for these two forms of inhibition. Personally, I think a within-domain vs. cross domain analysis is more intuitive and more common. Moreover, the tendency to interpret the current results as such is also enhanced by the fact that the domain specific and domain general data are plotted in a paired manner, essentially inviting the reader to compare those values. Thus, at a minimum, I would encourage that the domain specific and domain general data be reported in separate graphs. If the authors are arguing that it is not appropriate to compare these values, why plot them side-by-side (in paired format) in a graph?

(b) While I still think a within domain vs. across domain analyses would at least be worth looking at, I am certainly not saying it is wrong or inappropriate to test for classification of stop vs. no-think trials. Now that I understand what is being tested, however, it raises a different concern: why isn't domain-specific classification BETTER in the target regions than the prefrontal regions? Given the argument that prefrontal regions support a common form of inhibition across domains, it would seem that stopping and not thinking should be relatively MORE similar in prefrontal regions (because it is a common function) and less similar in the target regions (because, for these regions, it is not a common function). That would translate to higher domain-specific classification (stop vs. no-think) in target regions than in prefrontal regions. But that is not what is observed. Particularly for rVLPFC, domain-specific classification is much higher than in either target region (hippocampus or M1). So, the argument is that rVLPFC plays a domain general role in inhibition and it is apparently very important that there is generalization in rVLPFC from stop to no-think trials (which has to mean that they are associated with similar activity patterns), but ... it rVLPFC can also differentiate different forms of inhibition better than regions that play a domain specific role? This is very hard to reconcile. The authors have added an argument to the text that this performance in prefrontal regions may reflect sensitivity to the INPUTS that these regions receive. While this could be true, this just feels like a very slippery slope. I just can't get around the fact that in one analysis SIMILARITIES across conditions are taken as critical evidence that there is a domain general mechanism while in the other analysis, DIFFERENCES between these regions are NOT being taken as evidence against a domain general mechanism. I do agree with the authors about how and why prefrontal regions would need to support both functions (specific and general) but the issue to me is understanding how the same exact measure (similarity of BOLD activity patterns across conditions) can be interpreted in two different ways.

(c) The authors emphasize that a key point is that prefrontal regions showed above chance generalization (domain-general classification) whereas target regions did not. This does sound like a pretty important and clean distinction. But the concern I have is that the prefrontal regions really just look like they supported better classification OVERALL, compared to the target regions. (this is related to my point in my initial review about an interaction). What I am getting at is that some evidence of a double dissociation would be much more compelling. That is not shown. In the response letter, the authors include a fictional example (“Example against Necessity”) that they say would provide “excellent evidence” for their hypothesis despite not showing an interaction, but I do not agree with the logic here. In this fictional example, there is, again, a main effect of region (classification is worse in hippocampus than DLPFC). In theory, DLPFC and hippocampus could perform identical operations, but if there was greater noise in the hippocampus (which could be due to trivial properties like SNR, number of voxels, etc.) then you would get the pattern shown in this figure that the authors describe as “excellent evidence” for the hypothesis. Again, I just disagree with that reasoning. And this is precisely why double dissociations have so much inferential power.

(d) It is also notable that here (and this is a recurring problem), the authors present arguments but not data. Although I don’t agree with their argument about what constitutes “excellent evidence,” it is strange that, to the extent the authors see an opportunity for this straightforward analysis to yield excellent evidence, they don’t actually test it. They essentially say that this analysis is not worth doing because it could yield inconclusive results. But it is not statistical malpractice to run an analysis and ultimately conclude that the results are ambiguous. Yet, the authors dismiss the analysis entirely because the analysis MAY yield ambiguous results.

In comment 1.8, I had raised concern about the fact that the DCM analyses favored a bidirectional influence but there was a selective emphasis on the top-down effect (from PFC). I think this concern is more generally related to concerns raised by reviewer 3 (see below) about the confidence that we should have that DCM is providing information about causal influence. I suppose I am in a more skeptical camp and the bidirectional influence—with, as the reviewers acknowledge in their response, no claims about the relative strength of top-down vs. bottom-up influences—feels like weak evidence that the prefrontal regions are exerting a top-down influence. Of course, I agree with the authors that other evidence favors this interpretation. And I also acknowledge that it is not within the scope of this paper for the authors to justify DCM as a method. Thus, while I remain skeptical of the strength of inference that the DCM results afford, I also think it is fair to let readers interpret these results through the lens of their own beliefs about DCM.

Comments from other Reviewers:

Comments 3.4-3.6 (from Reviewer 3) were focused on the specificity of the DCM results and whether the frontal regions that were investigated were genuinely exerting a casual influence. The response from the authors was to defend the focus and to not include or report analyses that the reviewer suggested. While the authors do make some good points about hypothesis-driven analyses, DCM is not without its flaws and if similar DCM results were to be obtained (frontal-like effects) in many other brain regions, including regions that are NOT thought to have top-down influence, this would substantially weaken the arguments. Some strong control analyses would be very helpful, and the authors’ resistance to this is a bit vexing. The authors’ defense largely rests on the argument that we already know that these frontal regions play a causal role. (i.e., it’s not necessary to consider potential confounds because there is other evidence showing that PFC plays a causal role). But ... if it is, in fact, other results that provide the definitive proof of causal influence, and if the DCM results here are not designed to differentiate these frontal regions from other candidate regions, then what is the key insight afforded by the current DCM analyses? I am not trying to question whether there was some correlation between the PFC regions and the target regions, but causal arguments from fMRI invite greater skepticism. The defense provided by the authors feels like a concession that the DCM analysis is confirmatory and not testing alternative hypotheses. Personally, I would have found the DCM results more satisfying if the causal claims were just dropped and a simple correlation/connectivity analysis was reported (I would be less distracted by the issue of causality).

But this is a point where, again, I recognize I may be in a minority.

Reviewer 4 raised a question about the classification analyses and the response from the authors oriented me to something I had not realized: that the classification analyses are (as far as I can tell) not performed at the trial level but at the level of entire runs. Is this correct?? This is quite uncommon and should be emphasized (including in the figures). While I don't think the figure labels or description are technically wrong, I do think they are misleading and not ideal because the overwhelming majority of readers will assume that classification accuracy refers to the percentage of TRIALS that were successfully classified, not runs. In fact, while Reviewer 4 was not clear about some details of the classification analysis, I do not think that the reviewer even considered that the analyses were performed at the level of entire runs. The reviewer's question was about whether trials were balanced. Averaging trials within a run and then performing classification using the runs will, in my experience, generally improve classification accuracy when there is a real effect in the data. Thus, I think it's essential to at least emphasize (in multiple places) that a non-standard approach was used here.

Reviewer #2 (Remarks to the Author):

The authors have done a commendable job addressing my initial feedback and I have no remaining concerns.

Reviewer #3 (Remarks to the Author):

"Dynamic targeting enables domain-general inhibitory control over action and thought by the prefrontal cortex". Apšvalka et al., 2020. Nature Communications. NCOMMS-20-44422A-Z.

Summary & merits

In the resubmission of their study, Apšvalka et al. incorporate revisions that refine theory and discussion around their observations that two rIPFC domain-general source regions enact inhibitory control by dynamically interacting with domain-specific target regions. The feedback and concerns of four reviewers appears to be thoroughly considered and addressed by the authors, including a critical expansion upon theory that clarifies inferences based on the results. Namely, and I believe most impactfully, a delineation of mechanisms that would support the observations in Fig. 5 (especially panel C) in the scope of the dynamic targeting hypothesis. They explain that, while it may appear that rdIPFC and rvIPFC maintain domain-specific and domain-general representations, these source regions are still primarily exhibiting a domain-general inhibitory control function. This is based upon four processes: (1) domain-general 'stopping' processes/features; (2) domain-specific inputs to those sources; (3) domain-specific outputs; and (4) domain-specific stopping features. Further, (2)-(4) may be captured in the classification results for rIPFC, perhaps manifested by different multivariate patterns. Thus, the domain-general plus domain-specific components exhibited by rIPFC regions is accounted for by input/output layers conferring the domain specificity needed up- and down-stream. I appreciate that the authors also expand upon this in the Discussion section and note that future work can parse these mechanisms out further. These additions and clarifications greatly aid reader comprehension and intuition-building about the proposed mechanisms of dynamic targeting.

Comments:

On comment and reply 3.6: The detailed reply and action taken in the form of integrated/extensive edits to various sections of the paper are commendable and appreciated. One clarification I'd like to point out that does not have significant bearing on this paper as the revisions appear sufficient (particularly in light of the multiple lines of analysis, results, etc. already present in the paper) is that part of this comment was to suggest a control analysis, which could be either at the source end or the target end or both. For example (although this is just a simplistic example, there are certainly other ways to approach this): including a third source ROI that is not expected to confer inhibitory control to

target regions; one may hypothesize that this region would not exhibit the same domain-general representations and dynamic targeting capacity as the two rIPFC regions already included. A similar logic could be followed for the target regions, however, they do act as controls for each other so this is likely less of an outstanding question. This comment is included here as food for thought for future work as a straightforward way to curtail queries on ROI selection bias.

On comment and reply 3.13: As mentioned above, this revision is possibly the most pivotal in that it appears to address concerns across multiple reviewers. While the points and delineation of the four processes are well made, I wanted to point out that it is not abundantly clear how predictions #3 and #4 are different from one another (or c and d, respectively, as written in-manuscript). It may just be that 3 and 4 are related (domain-specific output features and domain specific stopping features, respectively) and are possibly driven by overlapping mechanisms, with the subtle difference that 3 is more about the interaction and 4 is more about the local process itself? However, I found myself filling in this gap and thought it might be helpful to add a sentence or two to explicitly state how these proposed processes are differentiated.

Reviewer #4 (Remarks to the Author):

The authors have done an excellent job of responding to the comments raised by myself and the other reviewers. They have expanded both the theoretical and empirical aspects of the study, which have greatly improved the manuscript. Thanks to the authors for a very thoughtful revision.

Thanks for clarifying the leave-one-run out. Indeed, I missed that. As a follow up, can the authors please add a sentence to indicate to readers that this has limitations in the sense that models can be overfitted based on learned aspects of individuals – i.e. these models are not generalizable to new people. As machine learning techniques become more popular in cognitive neuroscience, it might be helpful for readers to see both the benefits and possible issues laid out more clearly.

I also think there may be some confusion, particularly in the response letter, about the difference between kappa and accuracy. Kappa is % above chance and accuracy is % classified correctly out of all aspects. Related, the authors might consider adding some confusion matrices (or metrics of precision and recall values) to the supplementary materials, which could help readers understand if one class is being prioritized in the predictions.

Response to reviews:**NCOMMS-20-44422A-Z****REVIEWER 1****Reviewer Comment 1.1.**

The authors were in some ways very responsive in that they provided very detailed arguments and consideration of the concerns that were raised. In other ways, they were surprisingly un-responsive in that there was very little willingness to respond with new data or analyses (in response to comments from the reviewers, collectively). However, I want to acknowledge that I was clearly the most negative of the initial reviewers. Indeed, there was strong excitement from other reviewers. While some of my concerns persist (in large part because they were not addressed with data), I do not wish to 'talk down' the other reviewers. Rather, I will re-articulate my lingering concerns but I have no issues with these concerns being offset by greater enthusiasm from other reviewers. Indeed, as noted in my initial review, there are several positive aspects of this paper (broad interest, an appealing combination of methods). The sticking point for me is the strength of inferences drawn from the data and what I view as a bit of 'spin' that is put on the data.

Author Reply 1.1.

We are happy to address the reviewer's lingering concerns, and we are grateful for their openness to others' views.

Reviewer Comment 1.2.

The following are related to my original comment (comment 1.3):

(a) The authors are correct that I misinterpreted the original classification results. I apologize for the mistake. However, I find the logic/presentation here confusing, which contributed to my initial mistake. As the authors surmised, I had thought the domain specific analysis was testing for Respond vs. Inhibit classification WITHIN each domain. In other words, I had interpreted "domain specific" to mean that the analysis was "within domain." Instead, this analysis is testing for classification of stop vs. no-think trials. Thus, successful classification of stop vs. no-think trials means there is something different in the pattern of bold responses for these two forms of inhibition. Personally, I think a within-domain vs. cross domain analysis is more intuitive and more common. Moreover, the tendency to interpret the current results as such is also enhanced by the fact that the domain specific and domain general data are plotted in a paired manner, essentially inviting the reader to compare those values. Thus, at a minimum, I would encourage that the domain specific and domain general data be reported in separate graphs. If the authors are arguing that it is not appropriate to compare these values, why plot them side-by-side (in paired format) in a graph?

Author Reply 1.2.

We accept the reviewer's point about the organisation of our graph, and how this may inadvertently encourage people to compare our domain-specific and domain-general components, even though that was not our intention.

Author Action Taken 1.2.

We have replaced our original Figure 5C with the new one, pasted here. We separate the domain-specific and domain-general components on the left and right sides, respectively. This discourages direct comparison.

In addition, we did the “cost” analysis suggested (comparing within vs cross-domain classification across ROIs), which can be seen at the end of Author Reply 1.7. The analysis largely confirms the prediction that the reviewer attributed to us. We include this analysis in our online notebook (<http://bit.do/analysis-domain-general>). Because of our concerns about this analysis approach, we have not included it in the paper, and discuss its limited interpretation in the analysis notebook.

Reviewer Comment 1.3.

(b) While I still think a within domain vs. across domain analyses would at least be worth looking at, I am certainly not saying it is wrong or inappropriate to test for classification of stop vs. no-think trials.

Author Reply 1.3.

As noted above, the suggested analysis comparing within vs cross-domain classification across prefrontal and target ROIs can be found in Author Reply 1.7. The analysis confirms the prediction the reviewer attributed to us.

Reviewer Comment 1.4.

Now that I understand what is being tested, however, it raises a different concern: why isn't domain-specific classification BETTER in the target regions than the prefrontal regions? Given the argument that prefrontal regions support a common form of inhibition across domains, it would seem that stopping and not thinking should be relatively MORE similar in prefrontal regions (because it is a common function) and less similar in the target regions (because, for these regions, it is not a common function). That would translate to higher domain-specific classification (stop vs. no-think) in target regions than in prefrontal regions. But that is not what is observed. Particularly for rVLPFC, domain-specific classification is much higher than in either target region (hippocampus or M1). So, the argument is that rVLPFC plays a domain general role in inhibition and it is apparently very important that there is generalization in rVLPFC from stop to no-think trials (which has to mean that they are associated with similar activity patterns), but ... it

rVLPFC can also differentiate different forms of inhibition better than regions that play a domain specific role? This is very hard to reconcile.

Author Reply 1.4.

This is an interesting challenge. To paraphrase the issue, “if the theory says that DLPFC and VLPFC contain overlapping features for the two forms of stopping, but Hippocampus and M1 do not, doesn’t that mean that representations of these two tasks should be more distinct in the target regions? If so, doesn’t that mean that classifiers should reveal greater accuracy at distinguishing Stop from No-Think in the target regions than in PFC sources, when the opposite appears true in the data?” There are several points to make in response.

1. We suggest that it is ill-advised to build theoretical predictions around cross-ROI comparisons of accuracy. *Cross-area classifier accuracy comparisons* have interpretational difficulties stemming from extraneous factors that can affect overall accuracy, such as differences in (a) ROI size, (b) hemodynamic efficiency, and (c) signal quality. These issues are emphasised in tutorials on classification analysis (e.g., Haynes, 2015, an excellent Primer in Neuron). Therefore, we have grounded our theory’s prediction on outcomes within-ROIs, such as the presence or absence of cross-domain decoding in the PFC or in the Target regions.
2. To illustrate the problematic nature of cross-area comparisons, consider our reply to reviewer point 1.6, where we address the reviewer’s comments about “overall” classifier performance being better in VLPFC than in target regions. We showed that when one holds ROI size constant across all regions (one variable we can control), the apparently greater classifier accuracy in VLPFC than in Target regions vanishes, suggesting an artefact from the greater dimensionality of the VLPFC classification. It would be a mistake to build conclusions around the greater classification in PFC. The issue applies to the main concern of Reviewer comment 1.4.
3. In addition, the reviewer’s comment presumes that the features that distinguish Stop from No-Think are (a) the same across ROIs, and (b) of consistent reliability across ROIs. There are reasons NOT to assume this. For example, in the PFC, Stop/No-Think will be distinguished based on differences in input features (e.g., *No-Think trials, providing verbal input, coloured red or green; Stop trials, providing circles in different colours*) and output features concerning what brain structure the PFC has to target (M1 vs hippocampus) or even domain-specific stopping features. These input and output-related discriminating features would likely need to be *highly reliable* in the PFC, or else participants would not perform the correct task.

In contrast, in the Target structures (e.g., the hippocampus), output features may not reliably distinguish No-Think and Stop. For example, whereas during Think trials, the hippocampus will interact with the cortex to reinstate memories, during No-think trials, the whole objective of suppression is to prevent that output from happening; similarly, during Stop trials, there is no reason to expect any systemic, identifiable form of hippocampal output. Thus, discriminating on the basis of output features is unlikely to be reliable in the hippocampus (similar arguments apply to M1). Input features are subject to a similar concern.

4. No-Think and Stop trials can also be distinguished within target regions based on the impact of the inhibition process itself. We do not expect Stop trials to down-regulate hippocampal activity because the region should not be targeted by inhibitory control, so hippocampal activity during such trials is arguably “free to vary” noisily because it is task-irrelevant. On No-Think trials, in contrast, the whole point is to down-regulate hippocampal activity. The result will be differences between No-Think and Stop trials, but ones that are noisier, due to the undefined nature of hippocampal activity during Stop trials.

5. The moral of the story here is that it is not simply the NUMBER of features that distinguish Stop and No-Think trials that matter, but also their RELIABILITY. With differential reliability, one can easily observe more accurate discrimination of Stop and No-Think in the PFC than in the target regions, despite the presence of domain-general overlapping features in the PFC. Specifically, for a classifier to distinguish Stop from No-Think requires that only a modest subset of voxels exist that are highly reliable and diagnostic in distinguishing these conditions. For instance, if this subset was 100% reliable, classification accuracy will be high, even if most of the other voxels are routinely identical between the two conditions, or, alternatively, even if the other voxels are random noise. A smaller number of distinguishing features could exist in the PFC than in the two target regions (e.g., in the input or output layers of the network implementing control), but these features may simply be more reliable and diagnostic.
6. This is not simply a conjecture. We ran our classifiers on simulated data. In our simulation, as in our study, we had 24 subjects, the chosen simulated ROI had on average 24 voxels, and the performed classification was identical to our domain-specific classification as described in our Methods. We compared two versions. In version 1 (yellow line), the ROI had a subset of voxels which were perfectly reliable in predicting condition status. The remaining voxels were random noise. In version 2 (black line), the ROI also had a subset of voxels which were predictive of condition, but less reliably—noise was added; and the remaining voxels were random noise. We parametrically varied the proportion of voxels that are predictive of condition (from 1% to 100%). We tested whether an ROI with a small number of highly diagnostic voxels outperforms a second ROI with a very high fraction of condition-relevant voxels that were less reliable.

As you can see from the left figure, for example, an ROI with 20% of the voxels that were highly reliable (yellow line, third data point) (with the rest random noise), outperformed a second ROI with a much larger percentage of condition relevant voxels that were less reliably different (black line, which is lower for all % values). If, instead of using random noise voxels as the remaining voxels, you use overlapping voxel patterns, to represent systematic, but condition general activity, this difference is even greater (right panel).

This simulation illustrates why it is unwise to presume that domain-specific classification would be higher in target regions than in PFC regions, simply because there are no domain-general voxels in the target region. More broadly, it reinforces our general view that comparing classification accuracies across ROIs is fraught with interpretational problems. All the reviewer's objections to the data are centred around cross-ROI comparisons, which we have intentionally avoided with good justification.

7. For these reasons, we do not statistically compare the classification accuracies between the ROIs, as such comparisons are fraught. Rather, we approach the existence of the domain-general and domain-specific components primarily as a Yes/No question, which is sufficient for testing our hypotheses (in rare cases when we do examine quantitative differences in accuracy, it is always within ROI). From this standpoint, the fact that DLPFC and VLPFC contain both domain-specific and domain-general features, whereas the target regions only contain domain-specific features, is the main thing that matters.

In sum, it is straightforward to reconcile why VLPFC appears better at differentiating Stop and No-Think conditions than the target regions, despite domain-general voxels in the former. It's partially an artefact of ROI size. It's also a possibility that input/output features necessarily must be more reliable in the PFC to achieve proper task control.

Author Action Taken 1.4.

We considered adding a paragraph of discussion about this point to the paper or adding these analyses to the supplement. In the end, we elected not to do this because our paper already has many detailed analyses with multiple methods, and we do not wish to further lengthen it with analyses that readers are unlikely to put any weight on. However, if the reviewer feels strongly, we could include these simulations in the analysis notebook.

Reviewer Comment 1.5.

The authors have added an argument to the text that this performance in prefrontal regions may reflect sensitivity to the INPUTS that these regions receive. While this could be true, this just feels like a very slippery slope. I just can't get around the fact that in one analysis SIMILARITIES across conditions are taken as critical evidence that there is a domain general mechanism while in the other analysis, DIFFERENCES between these regions are NOT being taken as evidence against a domain general mechanism. I do agree with the authors about how and why prefrontal regions would need to support both functions (specific and general) but the issue to me is understanding how the same exact measure (similarity of BOLD activity patterns across conditions) can be interpreted in two different ways.

Author Reply 1.5.

It is helpful to remember that our conclusions about overlapping and distinct features between action and retrieval stopping arise from distinct classifiers, solving different problems. The behaviour of these is quantified in different DVs (not the same exact measure, as the reviewer states). It is also important to note that this DV is not global "similarity" in BOLD activity patterns, as the reviewer states (for similarity, RSA would have been a better measure), but rather the ability of a classifier to make a distinction between two patterns, which could in some cases rely on a subset of highly diagnostic voxels.

In classifier 1, we trained the classifier to distinguish Stopping from Going, and tested it on NT vs Think (or vice versa). The DV, in this case, is the probability that a NT trial would be classified as a Stop trial in the SST trained network, or vice versa. The fact that a classifier trained in one modality could correctly discern stopping in the second modality implies reliable commonalities in the patterns of activity generated by the two forms of stopping.

In classifier 2, we instead trained a classifier to distinguish Stopping from No-Think in one part of the data and examined whether it could continue to do so in the other part of the data (the activity patterns of Go and Think are not included here at all unlike in the previous, domain-general classifier). Here, the DV was

the probability with which a Stop trial would be successfully classified as a Stop trial in the Stop vs NT trained network (or a NT trial would be classified as a NT trial etc...). The ability to do so indicates that there are reliable enough differences between the tasks that a classifier can be trained to exploit them.

To see why this is not problematic, imagine a simple ROI with 24 voxels. In this ROI, let 12 of the 24 voxels be highly overlapping across tasks, with 6 (25%) reserved to represent domain-specific patterns on a common set of voxels (e.g., input or output patterns) and 6 are random noise. Classifier 1 will be optimised, over training trials, to detect the existence of the 12 overlapping voxels. Classifier 2 will be optimised to detect the difference between the 6 domain-specific voxels; the overlapping 12 voxels will be uninformative to classifier 2 as they simply don't discriminate between No-Think and Stop conditions. The classifier will learn to discriminate unique patterns in the input or output layers placing greater weight on these features. A classifier's capacity to do this is in no way incompatible with the existence of the domain-general component present in the overlapping 12 features.

In other words, domain-general and domain-specific features are not mutually exclusive. They are not negatively interdependent in the way the reviewer envisions. For example, if the noise in the input or output layer of the example changed, it would influence the evidence for domain-specific features correspondingly in Classifier 2 (we did something like this in Author Reply 1.4, for example). But Classifier 1, which relies on the other features, would be unaffected.

We are glad that the reviewer agrees with our inference that the PFC contains both domain-general and domain-specific features. We hope that these illustrations have helped to clarify why there is no inconsistency in the way we are treating the data.

Reviewer Comment 1.6.

(c) The authors emphasize that a key point is that prefrontal regions showed above chance generalization (domain-general classification) whereas target regions did not. This does sound like a pretty important and clean distinction. But the concern I have is that the prefrontal regions really just look like they supported better classification OVERALL, compared to the target regions. (this is related to my point in my initial review about an interaction). What I am getting at is that some evidence of a double dissociation would be much more compelling. That is not shown. In the response letter, the authors include a fictional example ("Example against Necessity") that they say would provide "excellent evidence" for their hypothesis despite not showing an interaction, but I do not agree with the logic here. In this fictional example, there is, again, a main effect of region (classification is worse in hippocampus than DLPFC). In theory, DLPFC and hippocampus could perform identical operations, but if there was greater noise in the hippocampus (which could be due to trivial properties like SNR, number of voxels, etc.) then you would get the pattern shown in this figure that the authors describe as "excellent evidence" for the hypothesis. Again, I just disagree with that reasoning. And this is precisely why double dissociations have so much inferential power.

Author Reply 1.6.

We are happy to address the reviewer's further concern that our data can be explained by the putative "overall" lower classification performance in the target regions (the Hippocampus and M1) than in the PFC regions (DLPFC and VLPFC).

The reviewer's main concern is illustrated in the left-hand panel of the figure below. The reviewer is concerned that the main reason that the domain-general classification for the Hippocampus and M1 (the pink dots in the right two columns) are at chance is because the overall performance of these regions is lower than the overall classification performance in the DLPFC and VLPFC (the left two columns).

Several observations are important to make in relation to this concern.

1. Our hypothesis predicts that the domain-general classification for the Hippocampus and M1 (the pink dots in the right two columns) *should be* at chance, whereas for the rDLPFC and rVLPFC (the pink dots in the left two columns) the classification should be above chance, as was observed. The reviewer is concerned that this difference may be an artefact of ROIs for the Hippocampus and M1 having lower performance “overall” than the PFC ROIs, perhaps because data quality issues limit the performance of any classifier. Although we do not encourage comparison of classification accuracy across ROIs (for the reasons above), if one were to do it, the only theory-neutral way of gauging whether the Hippocampal or M1 ROIs generally yield worse classification than the PFC classifiers is to evaluate the domain-specific classification analysis (green dots). If performance is also lower for Hippocampus and M1 than in PFC regions for these classifiers, it might suggest that the target region ROIs just perform poorly (due to any number of reasons, including data quality).
2. One could look at the green dots on the right two columns of the left panel and argue that they are lower than the green dots in the left two columns. However, this impression is primarily driven by the higher domain-specific (green dot) classification for the rVLPFC, which is indeed reliably higher in accuracy than the other ROIs; the rDLPFC (green dot) domain-specific classification accuracy is similar to classification in the target regions, and, in fact, does not reliably differ from them. Thus, there is no independent evidence to support the reviewer’s concern about higher performance levels in the rDLPFC, although the concern could be raised about the rVLPFC
3. However, the higher classification level for the rVLPFC (green dot) domain-specific classification is an artefact of trivial differences in ROI properties, in relation to the other regions. The rVLPFC ROI is bigger than the other ROIs. When we recomputed the analyses accounting for the ROI size differences (in our random subset selection keeping each subset constant across the 4 ROIs), this apparent superiority of the rVLPFC classifier vanishes. As can be seen in the righthand panel, the green dot (domain-specific) level is virtually constant across ROIs, with the main effect of classification performance being non-significant for this analysis, $F_{3, 69} = 1.703$, $p = 0.174$. In contrast, the main effect across the 4 ROIs, in the between-domain classification remains significant, reflecting the predicted chance performance of the cross-domain classifiers in the target regions, $F_{3,69} = 4.976$, $p = 0.003$. [* Note: we preserved the original classification analysis that was not matched for ROI size as the main one in the manuscript because (a) ROI sizes were determined a priori on the basis of anatomical boundaries (hippocampus) or analytic results (e.g.,

meta-analytic conjunction), and (b) we wanted to ensure that the same ROIs were used in all analyses, independent of method; this “size-matched” control analysis was done to illustrate that the VLPFC accuracy advantage is not likely to be meaningful].

4. Given that the reviewer’s concern rests on putatively higher classification accuracies in PFC ROIs than target ROIs, the fact that there is no statistically reliable evidence for these differences in the domain-specific classification condition should be greatly reassuring. Even without controlling for ROI size, only the rVLPFC showed any indication of higher classification, an effect that is an artefact of ROI size.
5. The reviewer raises an interesting alternative hypothesis—that PFC and target regions may be performing “identical operations” that are merely obscured by poorer quality ROIs in the Hippocampus and M1. On a priori grounds, this doesn’t have an obvious theoretical basis, it is right to consider it, as a “straw man”. On what grounds might anyone suspect that a classifier trained on Motor Cortex data to distinguish motor stopping from going would be able to predict the difference between episodic retrieval and retrieval stopping? We suggest that the current data set doesn’t provide strong support to the hypothesis. We, therefore, focus on the tests and the results related to the theory put forward in the paper.

Author Action Taken 1.6.

For readers who may be interested, we report the foregoing novel control analyses in our Supplementary Material and include all these classification results in our online Supplementary analysis notebook. We have, moreover, included a remark in the paper indicating that the somewhat higher classification level for rVLPFC is an artefact of ROI size, and does not survive when this is controlled, steering them to the supplementary material for further details.

“The superior discrimination of action and thought stopping in the rVLPFC compared to rDLPFC was reliable; however, control analyses matching ROI size eliminated this advantage, suggesting that it was an artefact of the larger ROI used for rVLPFC (see Figure S4).” (page 18, top)

We also include reference to this analysis further down on page 18, where we highlight that the comparable classification accuracy across ROIs in our control analysis argues against differences in performance level across ROIs.

“Notably, although comparisons of classification accuracies across ROIs should be interpreted with caution (Haynes, 2015), the ability of the classifier to distinguish No-Think from Stop trials did not vary across our four ROIs (rDLPFC, rVLPFC, hippocampus, M1) when ROI size was controlled (see Figure S4). Thus, all ROIs supported comparable classification performance in domain-specific classification, making it unlikely that the null classification results in the between-domain classifier in the hippocampus and M1 simply reflect poor signal quality in those target regions.”

Reviewer Comment 1.7.

(d) It is also notable that here (and this is a recurring problem), the authors present arguments but not data. Although I don’t agree with their argument about what constitutes “excellent evidence,” it is strange that, to the extent the authors see an opportunity for this straightforward analysis to yield excellent evidence, they don’t actually test it. They essentially say that this analysis is not worth doing because it could yield inconclusive results. But it is not statistical malpractice to run an analysis and ultimately conclude that the results are ambiguous. Yet, the authors dismiss the analysis entirely because the analysis MAY yield ambiguous results.

Author Reply 1.7.

We accept this general point. The data MIGHT NOT be subject to the ambiguities we are concerned about in our data. It could be, for example, that the contribution of input/output features to classification is constant across ROIs rather than variable; that diagnosticity of process-specific features is actually constant across ROIs and not variable, etc. Thus, the reviewer's suggested analyses could yield potentially interesting data in many cases, if the strong assumptions they must entail come true.

The problem is that if we report such additional analysis in our paper, we become responsible for the stated prediction in future studies by us, and others, even if we have little confidence in the stability of this prediction---we become responsible for future attempts to replicate what we do not believe to be reliable.

So, while it is possible for us to undertake the analyses the reviewer recommended, we do not feel comfortable reporting many of them, and they do not test predictions related to our hypotheses. An author should resist claiming a prediction (such as the reviewer's favoured prediction) that they don't stand behind.

To be specific--As discussed in Author Reply 1.3 of the previous response letter, the reviewer's logic of comparing the "cost" to classification performance of going from within to between domain classification (within-domain – cross-domain accuracy) across different ROIs (PFC vs Hippocampus or M1) is problematic on several grounds. We recapitulate the problems here.

1. Classifiers operating within a domain (e.g., No-Think versus Think) can learn to distinguish the two trial types on the basis of EITHER domain-general inhibition features OR task-specific features relating to input or output dynamics. For example, the classifier could learn to distinguish NT from Think trials based on perceptual input characteristics, such as red versus green colouring. Running this within-domain classifier on another task domain (Stopping vs Going in motor action) may yield a drop-off in classification accuracy (i.e., a cost) EITHER because intrinsic stopping processes are different in the two domains OR because the perceptual or motor (input or output) requirements differ for this task (which uses coloured circles as stimuli and demands manual output via button pressing). The cost mixes both.
2. Doing this same comparison across two ROIs (e.g., DLPFC vs M1) makes the unrealistically strong assumption that the impact of input/output features on each ROI is held constant, and that the only basis for observing a difference in cost across ROIs has to do with the presence or absence of domain-general features. It is simply not realistic to assume this—the inputs to motor cortex area M1 and its outputs cannot be assumed to be the same as they are in DLPFC.
3. For the foregoing reasons, the "cost" measure that the reviewer proposes does not yield clear predictions. One can observe bigger, smaller or the same size costs in M1 or the Hippocampus as we do in the PFC, depending on differences in the relative contributions of domain-general features and input/output features to classifier performance.
4. Added to these concerns, there could easily be differences in reliability of features across ROIs that contribute to further differences between cost measures in these ROIs. The impact of feature reliability is vividly illustrated in Author Reply 1.6.
5. For these reasons, EVEN IF the output of the reviewer's preferred analysis fully conformed to the reviewer's expectations, we would not wish to report it because we would then be in a position of endorsing an analysis that we feel is not consistently diagnostic.

Nonetheless, we have undertaken the reviewer's suggested "cost" analysis and report it in our online analysis notebook for readers, along with a discussion of its caveats. We illustrate the results of this below,

using ROIs matched for size (to make it somewhat more acceptable to compare the accuracies across ROIs). As is evident from this analysis, the cost of moving from the Within-Domain to the Domain-General classifier is significantly greater in Target ROIs than it is in the PFC source ROIs, as reflected in the General/Within x PFC/Target interaction ($F_{1,23} = 15.84$, $p = 0.0006$), precisely as the reviewer says it ought to be, if our theory is correct. (Significant main effect of Type: $p < 0.001$; non-significant main effect of ROI: $p = 0.164$)

Although the reviewer argues that these results are supportive of our hypothesis, we do not wish to present this evidence in the main paper for the reasons above. We focus instead on the fundamental predictions and tests we believe are most defensible: above chance cross-domain decoding in PFC regions, and chance cross-domain decoding in target regions.

Author Action Taken 1.7.

As noted above, we have added the foregoing analysis to the analysis notebook available online for readers so inclined to examine it. We respectfully request that the editor not insist that we report analyses in the manuscript that we don't stand behind, although we have done them and include them in our online analysis notebook, with appropriate caveats.

Reviewer Comment 1.8.

In comment 1.8, I had raised concern about the fact that the DCM analyses favoured a bidirectional influence but there was a selective emphasis on the top-down effect (from PFC). I think this concern is more generally related to concerns raised by reviewer 3 (see below) about the confidence that we should have that DCM is providing information about causal influence. I suppose I am in a more skeptical camp and the bidirectional influence—with, as the reviewers acknowledge in their response, no claims about the relative strength of top-down vs. bottom-up influences—feels like weak evidence that the prefrontal regions are exerting a top-down influence.

Of course, I agree with the authors that other evidence favours this interpretation. And I also acknowledge that it is not within the scope of this paper for the authors to justify DCM as a method. Thus, while I remain skeptical of the strength of inference that the DCM results afford, I also think it is fair to let readers interpret these results through the lens of their own beliefs about DCM.

Author Reply 1.8.

We thank the reviewer for their willingness to let readers decide for themselves about the DCM method. We agree that not everyone is persuaded by effective connectivity analyses. For what it is worth, work has been done to validate the causal inferences allowed by DCM. For example, as DCM has evolved over time, work with simulated data (where the ground truth of causal directions is known *a priori*) has shown that DCM effectively identifies the presence and direction of causality (multiple papers on this, with different variants of the method). Moreover, there have been empirical efforts to demonstrate, using real data in which the direction of causality is known, that effective connectivity analysis identifies the presence and direction of causality (e.g., work by Michael Cole and colleagues); and that DCM model selection is highly reliable over repeated sessions; and that the fMRI-DCM inferences are supported by neurophysiological interventions including TMS (eg. Boudrias and Ward). We don't expect to change the reviewer's sceptical stance, but it is important to recognise that objective efforts have been made to validate the inferences that the methods allow.

Reviewer Comment 1.9.

Comments from other Reviewers:

Comments 3.4-3.6 (from Reviewer 3) were focused on the specificity of the DCM results and whether the frontal regions that were investigated were genuinely exerting a causal influence. The response from the authors was to defend the focus and to not include or report analyses that the reviewer suggested. While the authors do make some good points about hypothesis-driven analyses, DCM is not without its flaws and if similar DCM results were to be obtained (frontal-like effects) in many other brain regions, including regions that are NOT thought to have top-down influence, this would substantially weaken the arguments. Some strong control analyses would be very helpful, and the authors' resistance to this is a bit vexing. The authors' defense largely rests on the argument that we already know that these frontal regions play a causal role. (i.e., it's not necessary to consider potential confounds because there is other evidence showing that PFC plays a causal role). But... if it is, in fact, other results that provide the definitive proof of causal influence, and if the DCM results here are not designed to differentiate these frontal regions from other candidate regions, then what is the key insight afforded by the current DCM analyses? I am not trying to question whether there was some correlation between the PFC regions and the target regions, but causal arguments from fMRI invite greater skepticism. The defense provided by the authors feels like a concession that the DCM analysis is confirmatory and not testing alternative hypotheses. Personally, I would have found the DCM results more satisfying if the causal claims were just dropped and a simple correlation/connectivity analysis was reported (I would be less distracted by the issue of causality). But this is a point where, again, I recognize I may be in a minority.

Author Reply 1.9.

The paper uses complementary analytical methods. The convergence or consensus in their results is a striking feature of the paper that many readers will appreciate, without being distracted by claims of primacy of one method over another. Some are more confident in the statistical simplicity of correlations, while others are frustrated by the lack of evidence for directionality in correlations, when tools exist to formally test alternate causal models.

To further clarify the role of Dynamic Causal Modelling:

1. The reviewer is correct that we noted in our original response letter that other data already exist to show the causal role of right lateral PFC in motor response inhibition and in retrieval stopping.

Moreover, we have previously shown using DCM that the right DLPFC modulates hippocampal activity during retrieval suppression. Thus, there is *a priori* reason to believe in the causal status of these regions, justifying our focus on them. But that prior evidence does not prove dynamic targeting, or dynamic connectivity.

2. The objectives of the present paper are to test whether (i) the very same regions that implement motor stopping also implement retrieval stopping, and (ii) the connectivity of these regions to their respective target structures dynamically shifts with task goals, yielding inhibition in the affected target region and behavioural indices of inhibition. In other words, the goal is to test whether the 5 properties of dynamic inhibitory targeting apply to these regions. Two of the properties—the existence of a causal influence, and the domain-specific targeting of that influence – are best tested by effective connectivity analysis (more than functional connectivity measures like correlations). DCM contributes to the evaluation of this dynamic targeting concept and its attribution to the rDLPFC and rVLPFC.
3. It is true that other regions were observed that showed domain-general activity in both our within-subjects conjunction and our meta-analytic conjunction. We report these for completeness, to highlight their importance for future work. Indeed, we report them in some detail in anticipation of future the considerations of large-scale brain networks and their role in achieving inhibitory control (such tests require technological innovations and are not yet practicable). However, the purpose of our project was not to establish the roles of these other regions. Indeed, our design is ill suited to test the roles of these other regions: we don't have specific hypotheses about these other regions, nor do we have manipulations that could tease apart their relative roles. Our focus was from the start on the prefrontal cortex. This *a priori* approach complements exploratory network discovery approaches.

Although we had not planned DCM source analyses using regions outside the right lateral prefrontal cortex, we have done so. This meets the reviewer's request that we look at regions that we did not suspect to be sources of inhibition.

To examine whether another pair of regions might also behave like the rDLPFC and rVLPFC, we replaced the rDLPFC and rVLPFC with two different areas that we had not planned to examine. Our goal was to ask whether, using these new "Source" ROIs, there was a clear winning model indicating that the Hippocampus and M1 were modulated by them during Retrieval or Action Stopping.

To choose a region, from amongst those identified in our meta-analytic conjunction analysis, we sought ROIs that exhibited evidence of cross-task decoding. If our replacement regions were both activated by Action and Retrieval Stopping and showed evidence of domain-general stopping representations, it would be a strong test of whether the DLPFC and VLPFC are unique in their capacity to exert top-down modulation. Thus, we performed a cross-task decoding analysis on all regions. Only two other regions (apart from rDLPFC and rVLPFC) showed cross-domain decoding: right and left IPL (classification test results for all ROIs are summarised in Supplementary material).

Using these nodes *in lieu* of rDLPFC and rVLPFC, we tested the analogous space of 73 models to see if there was compelling evidence for a model in which this pair of regions behaved like our prefrontal regions. The resulting exceedance probabilities are displayed below.

For reference, it is conventional within the DCM community that exceedance probabilities below .8 are of less interest. For a model to be considered a strong account of the data, values $>.9$ are sought. The lefthand panel is our submitted rDLPFC/rVLPFC model; and the right reflects the models with right and left IPL instead of rDLPFC/rVLPFC. The 73 models in each analysis are along the x-axis; their exceedance probability is on the y-axis.

The left panel shows the clear winning model in the rDLPFC/rVLPFC analysis (exceedance probability of .92). In contrast, the IPL analysis (right panel) does not yield a model that provides a compelling account of the data (the highest exceedance probability is .25). This suggests that dynamic inhibitory targeting is unlikely to originate from IPL, and by extension, cannot be inferred simply from the presence of cross-task decoding.

Although these data favour our emphasis on right lateral PFC and our attribution of a causal role of those regions, we emphasise that our intention was not to map out the entire network in this project but to assess the causal status of prefrontal regions. It is beyond the scope of this paper to try to test the full architecture of inhibitory control. Rather, we focus on the lateral PFC to demonstrate proof-of-concept of the dynamic targeting attributes.

Author Action Taken 1.9.

We agree with the reviewer that this analysis plays a worthwhile role as a control to show that not all regions activated by Action and Retrieval Stopping are necessarily involved in top-down modulation with dynamic targeting. We have inserted new text in the final results paragraph on page 24, reporting this analysis and steering readers to the Supplements for details. (see paragraph below for convenience).

“The preferential modulations of hippocampus or M1, depending on whether thoughts or actions are to be suppressed, confirm our key hypothesis that top-down modulation by rDLPFC and rVLPFC is dynamically targeted depending on participants’ task goals. However, a winning model with goal-dependent top-down connectivity to the hippocampus and M1 might be identified for any brain region robustly activated by both action and retrieval stopping, and not just the rDLPFC and rVLPFC. To test this possibility, we modified our DCM analysis by replacing the rDLPFC and rVLPFC nodes with two other regions from our meta-analytic conjunction analysis as sources of control. To choose regions, we

performed our domain-general classification analysis on all ten meta-analytic conjunction regions (see Table 1b). Apart from rDLPFC and rVLPFC, only the right and left inferior parietal lobule (IPL) exhibited significant domain-general component (see Figure S5). Using the right and left IPL as sources of control, DCM did not reveal a model with clear evidence for top-down modulation of hippocampus and M1 (see Figure S6). Thus, to be activated by both stopping tasks and to show cross-task decoding is not sufficient to infer goal-dependent inhibitory modulation of connectivity. Instead, our results suggest that rDLPFC and rVLPFC may be particularly important origins of this targeted signal. Together, the results of the DCM analysis suggest that, when stopping a prepotent response, rDLPFC and rVLPFC, interact with each other and are both selectively coupled with M1 when stopping actions and selectively coupled with the hippocampus when stopping thoughts: in other words, both regions manifest dynamic targeting.”

Reviewer Comment 1.10.

Reviewer 4 raised a question about the classification analyses and the response from the authors oriented me to something I had not realized: that the classification analyses are (as far as I can tell) not performed at the trial level but at the level of entire runs. Is this correct?? This is quite uncommon and should be emphasized (including in the figures). While I don't think the figure labels or description are technically wrong, I do think they are misleading and not ideal because the overwhelming majority of readers will assume that classification accuracy refers to the percentage of TRIALS that were successfully classified, not runs. In fact, while Reviewer 4 was not clear about some details of the classification analysis, I do not think that the reviewer even considered that the analyses were performed at the level of entire runs. The reviewer's question was about whether trials were balanced. Averaging trials within a run and then performing classification using the runs will, in my experience, generally improve classification accuracy when there is a real effect in the data. Thus, I think it's essential to at least emphasize (in multiple places) that a non-standard approach was used here.

Author Reply 1.10.

The reviewer is correct that our classification analyses were performed at the level of runs and not individual trials. We reported this methodological choice transparently in our original methods section.

Contrary to the reviewer's statement, run-wise classification (or other forms of 'temporal compression') is standard, and at least as common as trial-wise classification (Haynes, 2015; Valente et al., 2021). For designs like ours (non-blocked event-related, within-subject classification), run-wise is a better approach as it provides a clearer class identity and temporal independence and improves the SNR (Mourão-Miranda et al., 2006; Etzel et al., 2009, 2011; Allefeld and Haynes, 2014; Yang et al., 2014). To illustrate this point, consider this quote from Haynes' (2015) "Primer" on pattern-based approaches, published in *Neuron*, which states that (a) run-wise classification is normal, and (b) run-wise classification is often preferable.

“A second important choice is the level of temporal aggregation (Figure 4). The individual samples entering a classifier analysis can stem from single fMRI volumes, single trials, single blocks, entire scanning runs, or even from single subjects (see Figure 4 for details). It is even possible to use entire spatiotemporal patterns for classification (Mourão-Miranda et al., 2007). The level of temporal aggregation is important when it comes to ensuring the statistical independence between training and test data sets in each of the cross-validation folds (Figure 4; see also Mumford et al., 2014). ...**The safest solution is to perform a cross-validation across independent fMRI measurement periods (runs) for both trial-based and block-based experiments (Mumford et al., 2014)**” (Haynes, 2015, page 261-262).”

This accepted methodological recommendation is reflected in the widespread use of run-based classification, contrary to the reviewer's claim. For example, a quick search identified multiple studies from high profile researchers and high impact outlets (e.g., Science) implementing run-wise classification: sequence-tapping tasks (6 runs; 8 runs; 8 runs), (Wiestler and Diedrichsen, 2013, Elife; Kornysheva and Diedrichsen, 2014, Elife; Wiestler et al., 2014, Journal of Neuroscience); decision-making tasks (8 and 5 runs) (Schuck et al., 2016, Neuron; Schuck and Niv, 2019, Science); cue-approach training (6 runs) (Bakkour et al., 2017; Neuroimage); Go/No-Go task (6 runs) (Fedota et al., 2014, Neuropsychologia). Of course, one could name numerous studies using trial-wise analysis. But the point is that the right choice depends on the design and aims of the study.

In our case, the better choice was run-wise analysis.

Regarding Reviewer 1's comment about Reviewer 4, the latter initially thought we were doing cross-subject classification, which would also create an uneven number of exemplars per condition. However, as should be clear from **Reviewer Comment 4.2** in this response letter, this misunderstanding has been clarified, and the reviewer understands and accepts our run-based analysis.

Author Action Taken 1.10.

We added an additional clarification in the Methods (page 43) to emphasise that we are performing a run-wise classification:

"We performed a run-wise classification. For designs like ours (non-blocked event-related, within-subject classification), compared to trial-wise classification, run-wise classification is proven to be a better approach. It provides a clearer class identity and temporal independence and improves the signal-to-noise (Allefeld and Haynes, 2014; Etzel et al., 2009, 2011; Mourão-Miranda et al., 2006; Yang et al., 2014)."

REVIEWER 2

Reviewer Comment 2.1.

The authors have done a commendable job addressing my initial feedback and I have no remaining concerns.

Author Reply 2.1

We thank the reviewer for their nice remarks, and the very useful comments along the way.

REVIEWER 3

Reviewer Comment 3.1.

"Dynamic targeting enables domain-general inhibitory control over action and thought by the prefrontal cortex". Apšvalka et al., 2020. Nature Communications. NCOMMS-20-44422A-Z.

Summary & merits

In the resubmission of their study, Apšvalka et al. incorporate revisions that refine theory and discussion around their observations that two rIPFC domain-general source regions enact inhibitory control by dynamically interacting with domain-specific target regions. The feedback and concerns of four reviewers appears to be thoroughly considered and addressed by the authors, including a critical expansion upon theory that clarifies inferences based on the results. Namely, and I believe most impactfully, a delineation

of mechanisms that would support the observations in Fig. 5 (especially panel C) in the scope of the dynamic targeting hypothesis. They explain that, while it may appear that rdIPFC and rvIPFC maintain domain-specific and domain-general representations, these source regions are still primarily exhibiting a domain-general inhibitory control function. This is based upon four processes: (1) domain-general ‘stopping’ processes/features; (2) domain-specific inputs to those sources; (3) domain-specific outputs; and (4) domain-specific stopping features. Further, (2)-(4) may be captured in the classification results for rIPFC, perhaps manifested by different multivariate patterns. Thus, the domain-general plus domain-specific components exhibited by rIPFC regions is accounted for by input/output layers conferring the domain specificity needed up- and down-stream. I appreciate that the authors also expand upon this in the Discussion section and note that future work can parse these mechanisms out further. These additions and clarifications greatly aid reader comprehension and intuition-building about the proposed mechanisms of dynamic targeting.

Author Reply 3.1.

We thank the reviewer for these kind remarks and are very happy that our expansion of the theory helps strengthen the case and build comprehension.

Reviewer Comment 3.2.

Comment:

On comment and reply 3.6: The detailed reply and action taken in the form of integrated/extensive edits to various sections of the paper are commendable and appreciated. One clarification I’d like to point out that does not have significant bearing on this paper as the revisions appear sufficient (particularly in light of the multiple lines of analysis, results, etc. already present in the paper) is that part of this comment was to suggest a control analysis, which could be either at the source end or the target end or both. For example (although this is just a simplistic example, there are certainly other ways to approach this): including a third source ROI that is not expected to confer inhibitory control to target regions; one may hypothesize that this region would not exhibit the same domain-general representations and dynamic targeting capacity as the two rIPFC regions already included. A similar logic could be followed for the target regions, however, they do act as controls for each other so this is likely less of an outstanding question. This comment is included here as food for thought for future work as a straightforward way to curtail queries on ROI selection bias.

Author Reply 3.2.

We thank the reviewer for this useful suggestion, which we will take on board in future studies.

Although the reviewer did not request that we do any additional analyses for this comment, we emphasize that the analysis we reported in Author Reply 1.9 in this response letter serves the same role as a “control analysis” that the reviewer recommends for the future. When we performed the DCM analysis using parietal regions instead of frontal regions as the source of inhibitory control, evidence for targeted top-down modulation of hippocampus and M1 disappeared. This shows that it is possible for two regions that are part of the domain-general univariate activation pattern to NOT provide evidence for top-down modulation, validating our focus on PFC.

We now report this analysis in a supplement and mention it in a paragraph on page 24, printed below for convenience.

“The preferential modulations of hippocampus or M1, depending on whether thoughts or actions are to be suppressed, confirm our key hypothesis that top-down modulation by rDLPFC and rVLPFC is dynamically targeted depending on participants’ task goals. However, a winning model with goal-dependent top-down connectivity to the hippocampus and M1 might be identified for any brain region robustly activated by both action and retrieval stopping, and not just the rDLPFC and rVLPFC. To test this possibility, we modified our DCM analysis by replacing the rDLPFC and rVLPFC nodes with two other regions from our meta-analytic conjunction analysis as sources of control. To choose regions, we performed our domain-general classification analysis on all ten meta-analytic conjunction regions (see Table 1b). Apart from rDLPFC and rVLPFC, only the right and left inferior parietal lobule (IPL) exhibited significant domain-general component (see Figure S5). Using the right and left IPL as sources of control, DCM did not reveal a model with clear evidence for top-down modulation of hippocampus and M1 (see Figure S6). Thus, to be activated by both stopping tasks and to show cross-task decoding is not sufficient to infer goal-dependent inhibitory modulation of connectivity. Instead, our results suggest that rDLPFC and rVLPFC may be particularly important origins of this targeted signal. Together, the results of the DCM analysis suggest that, when stopping a prepotent response, rDLPFC and rVLPFC, interact with each other and are both selectively coupled with M1 when stopping actions and selectively coupled with the hippocampus when stopping thoughts: in other words, both regions manifest dynamic targeting.”

Reviewer Comment 3.3.

On comment and reply 3.13: As mentioned above, this revision is possibly the most pivotal in that it appears to address concerns across multiple reviewers. While the points and delineation of the four processes are well made, I wanted to point out that it is not abundantly clear how predictions #3 and #4 are different from one another (or c and d, respectively, as written in-manuscript). It may just be that 3 and 4 are related (domain-specific output features and domain specific stopping features, respectively) and are possibly driven by overlapping mechanisms, with the subtle difference that 3 is more about the interaction and 4 is more about the local process itself? However, I found myself filling in this gap and thought it might be helpful to add a sentence or two to explicitly state how these proposed processes are differentiated.

Author Reply 3.3.

The reviewer has understood the intended distinction correctly. Domain-specific output features specifically implement the interaction of the PFC with the intended domain-specific target region; domain-specific stopping features, by contrast, reflect hypothetical special-purpose computations within the stopping mechanism itself (within the local PFC region implementing stopping) needed to perform stopping within a particular domain. In a multi-layer distributed network, domain-specific output features would arise in the output layer of the network and influence communication with other regions. But domain-specific stopping features would arise in the network’s hidden layers. We do not specify, in the theory, what those domain-specific stopping computations might be, but we thought it was important to allow for their potential contribution.

Author Action Taken 3.3.

We agree that this distinction is subtle, and that more support could be provided for the reader in understanding it. To address this recommendation, we included the following additional clarifying sentence on page 17.

“Domain-specific stopping features differ from domain-specific output features in that the latter govern the interaction of the stopping mechanism with posterior-cortical or subcortical target regions, whereas the former reflect computations specific to a stopping domain that are local to the prefrontal source region.”

REVIEWER 4

Reviewer Comment 4.1.

The authors have done an excellent job of responding to the comments raised by myself and the other reviewers. They have expanded both the theoretical and empirical aspects of the study, which have greatly improved the manuscript. Thanks to the authors for a very thoughtful revision.

Author Reply 4.1.

We thank the reviewer for their kind remarks about our revision!

Reviewer Comment 4.2.

Thanks for clarifying the leave-one-run out. Indeed, I missed that. As a follow up, can the authors please add a sentence to indicate to readers that this has limitations in the sense that models can be overfitted based on learned aspects of individuals – i.e. these models are not generalizable to new people. As machine learning techniques become more popular in cognitive neuroscience, it might be helpful for readers to see both the benefits and possible issues laid out more clearly.

Author Reply 4.2.

The reviewer is correct that our within-subjects classifiers are specific to the individual subjects used to train and test them. These classifiers were never intended to generalize to other participants, only to make the point that, for a given participant, a common mechanism exists that supports both action and thought stopping, as reflected in cross-domain classification.

Author Action Taken 4.2.

We added additional clarifications in the Results and Methods.

In the Results, on page 17, we updated the following sentence:

“To identify the predicted cross-modality similarities, **within each subject**, we trained a classifier to distinguish Inhibit and Respond conditions in one modality and tested the ability to distinguish Inhibit and Respond conditions in the other modality.”

In the Methods, on pages 42 and 43, we updated these three sentences:

“We used linear discriminant analysis (LDA) **and within-subject classification** to classify voxel activity patterns within the same four ROIs that we used for the DCM analysis”

“For the domain-general component, we performed a **within-subject** cross-task classification. “

“For the domain-specific component, we trained and tested the **within-subject** LDA classifier to distinguish No-Think from Stop conditions.”

And added this additional sentence:

“Note that with the within-subject classification, the trained classifiers cannot be generalised to other subjects.”

Reviewer Comment 4.3.

I also think there may be some confusion, particularly in the response letter, about the difference between kappa and accuracy. Kappa is % above chance and accuracy is % classified correctly out of all aspects. Related, the authors might consider adding some confusion matrices (or metrics of precision and recall values) to the supplementary materials, which could help readers understand if one class is being prioritized in the predictions.

Author Reply 4.3.

We agree, and are happy to add the requested confusion matrices in the Supplementary materials.

Author Action Taken 4.3.

We have added confusion matrices to the Supplementary materials Figure S6 and Figure S7.

Figure S1. Domain-general classification confusion matrices for each ROI.

Figure S2. Domain-specific classification confusion matrices for each ROI.

REFERENCES

- Allefeld C, Haynes J-D (2014) Searchlight-based multi-voxel pattern analysis of fMRI by cross-validated MANOVA. *NeuroImage* 89:345–357 doi:10.1016/j.neuroimage.2013.11.043.
- Bakkour A, Lewis-Peacock JA, Poldrack RA, Schonberg T (2017) Neural mechanisms of cue-approach training. *NeuroImage* 151:92–104 doi:10.1016/j.neuroimage.2016.09.059.
- Etzel JA, Gazzola V, Keysers C (2009) An introduction to anatomical ROI-based fMRI classification analysis. *Brain Research* 1282:114–125 doi:10.1016/j.brainres.2009.05.090.
- Etzel JA, Valchev N, Keysers C (2011) The impact of certain methodological choices on multivariate analysis

- of fMRI data with support vector machines. *NeuroImage* 54:1159–1167 doi:10.1016/j.neuroimage.2010.08.050.
- Fedota JR, Hardee JE, Pérez-Edgar K, Thompson JC (2014) Representation of response alternatives in human presupplementary motor area: Multi-voxel pattern analysis in a go/no-go task. *Neuropsychologia* 56:110–118 doi:10.1016/j.neuropsychologia.2013.12.022.
- Haynes J-D (2015) A Primer on Pattern-Based Approaches to fMRI: Principles, Pitfalls, and Perspectives. *Neuron* 87:257–270 doi:10.1016/j.neuron.2015.05.025.
- Kornysheva K, Diedrichsen J (2014) Human premotor areas parse sequences into their spatial and temporal features. *eLife* 3:e03043 doi:10.7554/eLife.03043.
- Mourão-Miranda J, Reynaud E, McGlone F, Calvert G, Brammer M (2006) The impact of temporal compression and space selection on SVM analysis of single-subject and multi-subject fMRI data. *NeuroImage* 33:1055–1065 doi:10.1016/j.neuroimage.2006.08.016.
- Schuck NW, Cai MB, Wilson RC, Niv Y (2016) Human Orbitofrontal Cortex Represents a Cognitive Map of State Space. *Neuron* 91:1402–1412 doi:10.1016/j.neuron.2016.08.019.
- Schuck NW, Niv Y (2019) Sequential Reply of nonspatial task states in the human hippocampus. *Science* 364:eaaw5181 doi:10.1126/science.aaw5181.
- Valente G, Castellanos AL, Hausfeld L, De Martino F, Formisano E (2021) Cross-validation and permutations in MVPA: validity of permutation strategies and power of cross-validation schemes. *NeuroImage*:118145 doi:10.1016/j.neuroimage.2021.118145.
- Wiestler T, Diedrichsen J (2013) Skill learning strengthens cortical representations of motor sequences. *eLife* 2:e00801 doi:10.7554/eLife.00801.
- Wiestler T, Waters-Metenier S, Diedrichsen J (2014) Effector-independent motor sequence representations exist in extrinsic and intrinsic reference frames. *The Journal of neuroscience : the official journal of the Society for Neuroscience* 34:5054–5064 doi:10.1523/JNEUROSCI.5363-13.2014.
- Yang Z, Huang Z, Gonzalez-Castillo J, Dai R, Northoff G, Bandettini P (2014) Using fMRI to decode true thoughts independent of intention to conceal. *NeuroImage* 99:80–92 doi:10.1016/j.neuroimage.2014.05.034.

Reviewers' comments:

Reviewer #1 (Remarks to the Author):

The authors have addressed my remaining concerns. I appreciate the thoroughness of their responses (to my comments and those from other authors). I believe the manuscript has been strengthened and is now suitable for publication.